# Systemically injected oxygen within rapidly dissolving microbubbles improves the outcomes of severe hypoxaemia in swine

Julia Garcia Mancebo[1,2,9], Kristen Sack[1,9], Jay Hartford[1], Saffron Dominguez[1], Michelle Balcarcel-Monzon[1], Elizabeth Chartier[1], Tien Nguyen[1], Alexis R. Cole[1], Francesca Sperotto [1,2], David M. Harrild[1,2], Brian D. Polizzotti[1,2], Allen D. Everett[3], Alan B. Packard [4,5], Jason Dearling[4,5], Arthur G. Nedder[6], Simon Warfield [4,5], Edward Yang[4,5], Hart G. W. Lidov[7,8], John N. Kheir [1,2]✉ & Yifeng Peng [1,2]✉

Acute respiratory failure can cause profound hypoxaemia that leads to organ injury or death within minutes. When conventional interventions are ineffective, the intravenous administration of oxygen can rescue patients from severe hypoxaemia, but at the risk of microvascular obstruction and of toxicity of the carrier material. Here we describe polymeric microbubbles as carriers of high volumes of oxygen (350–500 ml of oxygen per litre of foam) that are stable in storage yet quickly dissolve following intravenous injection, reverting to their soluble and excretable molecular constituents. In swine with profound hypoxaemia owing to acute and temporary (12 min) upper-airway obstruction, the microbubble-mediated delivery of oxygen led to: the maintenance of critical oxygenation, lowered burdens of cardiac arrest, improved survival, and substantially improved neurologic and kidney function in surviving animals. Our findings underscore the importance of maintaining a critical threshold of oxygenation and the promise of injectable oxygen as a viable therapy in acute and temporary hypoxaemic crises.

Hypoxaemia, or low blood oxygen saturation, can occur in the setting of lung disease or airway obstruction, which causes blood to circulate through the lungs and then to the body without being fully reoxygenated. In hospitalized patients, severe, episodic hypoxaemia can result from endotracheal tube occlusion (for example, secretions), progressive lung injury and a number of other causes. Such episodes may be addressed with airway clearance, lung recruitment, increased ventilatory support, and when needed, inhaled nitric oxide or extracorporeal membrane oxygenation. However, hypoxaemia is sometimes temporarily so severe and refractory to these manoeuvres that myocardial contractility fails, resulting in pulseless electrical activity or even asystole; this clinically manifests as cardiac arrest. Of note, 15–40% of in-hospital cardiac arrests (IHCA) are caused by respiratory insufficiency[1]. Such patients exhibit a hypoxic insult compounded by an ischaemic insult (that is, cessation of cardiac activity and blood flow), resulting in severe ischaemic injury to the brain, kidneys and other

[1]Department of Cardiology, Boston Children's Hospital, Boston, MA, USA. [2]Department of Pediatrics, Harvard Medical School, Boston, MA, USA. [3]Department of Pediatrics, Blalock-Taussig-Thomas Congenital Heart Center, Johns Hopkins University, Baltimore, MD, USA. [4]Department of Radiology, Boston Children's Hospital, Boston, MA, USA. [5]Department of Radiology, Harvard Medical School, Boston, MA, USA. [6]Animal Resources at Children's Hospital, Boston Children's Hospital, Boston, MA, USA. [7]Department of Pathology, Boston Children's Hospital, Boston, MA, USA. [8]Department of Pathology, Harvard Medical School, Boston, MA, USA. [9]These authors contributed equally: Julia Garcia Mancebo, Kristen Sack. ✉e-mail: John.Kheir@childrens.harvard.edu; Yifeng.Peng@childrens.harvard.edu

organs. Survival to hospital discharge following IHCA approximates 20%, with ~1 in 3 patients suffering from substantial neurologic impairment[2]. It is well recognized that survival following IHCA is enhanced by manoeuvres that optimize oxygen delivery during IHCA (such as high-quality cardiopulmonary resuscitation (CPR))[3], achieving early return of spontaneous circulation (ROSC)[4] and rapid cannulation to extracorporeal membrane oxygenation during CPR[5]. Successful resuscitation from IHCA requires the rapid identification and reversal of the underlying cause[6]; however, reversing refractory hypoxaemia remains an unmet challenge.

The intravenous administration of oxygen ($IVO_2$) via an injectable gas carrier offers a complementary mechanism to rapidly increase blood oxygen saturation[7,8]. $IVO_2$ is distinct from blood substitutes made of perfluorocarbon or haemoglobin analogues, which are optimized as circulating gas carriers in the setting of a functional lung unit rather than administering oxygen to the bloodstream[9–12]. Similarly, the transfusion of oxygenated blood itself is an impractical treatment for hypoxaemia because the relatively low gas fraction of blood would require administration of a volume that would quickly overwhelm the circulatory system and cause lung injury and heart failure[13]. Thus, alternative materials have been sought to develop gas carriers that improve oxygen carrying capacity in fluids[14].

Several groups have described microbubbles (MBs) coated with phospholipids or polymer shells to deliver oxygen to acutely reverse hypoxaemia in animal models[15–20]. However, translating $IVO_2$ from biomaterials research to a clinically viable therapeutic faces many hurdles. This is primarily because the intravenous administration of even small volumes of gas in an emergency setting requires that concerns of microvascular obstruction, material-related toxicities and product viability be addressed[21–23]. It is therefore also unknown whether the administration of $IVO_2$ in severe hypoxaemia would be 'clinically' beneficial. In this work, we pursue a rational approach to design polymeric microbubbles (PMBs) as gas carriers for translating $IVO_2$. PMBs are designed to instantaneously dissolve in physiologic media without a diffusion sink, their shells reverting to soluble, excretable molecular constituents, thus allowing rapid delivery of high volumes of oxygen without observable acute safety or toxicity risks. We show that injection of oxygen via PMBs substantially improves survival and neurologic outcomes in a realistic animal model of severe hypoxaemic cardiac arrest. These findings illustrate the therapeutic importance of maintaining a critical threshold of oxygenation during acute, profound hypoxaemic events, highlighting the unique potential of $IVO_2$ as a viable therapy in this setting.

## Results

### Gas-carrier design

We consider several factors to be critical to the general design of a pharmaceutically acceptable gas carrier. First, the carriers must rapidly release oxygen upon contact with blood. Second, any particulate that is injected must rapidly dissolve following administration to avoid vascular obstruction. Third, carrier fluids must be minimized to avoid fluid overload. Fourth, the carrier materials must have low toxicity and be efficiently cleared to minimize long-term side effects.

Initially, lipid-coated oxygen microbubbles (LOMs) appeared to be a promising and intuitive solution to $IVO_2$, as they are highly biocompatible, have high gas fractions and resolve into micelles following gas dissolution in the presence of an oxygen sink[15]. However, lipid shells inhibit complete bubble dissolution via surface reorganization and gas exchange[24–26], and infused LOMs persist in circulation and cause obstruction via gas embolism even at low doses in healthy animals[21], which forbids their use as a resuscitation therapeutic in emergency settings (see Supplemental Fig. 7 in ref. 20). To enhance gas dissolution, we recently created pH-responsive MBs that are stable in storage but rapidly dissolve in blood, triggered by physiologic pH[20]. This was achieved by a method of interfacial nanoprecipitation (IFNP) to form a

solid shell made of a pH-responsive polymer surrounding a gas core[27]. We showed that IFNP MBs significantly decreased the acute risks of vascular obstruction when compared with LOMs and other microcarriers[20]. Unfortunately, IFNP requires the use of hydrophobic polymers that must be water insoluble to enable nanoprecipitation in aqueous solution after they are predissolved in an organic co-solvent. As a result, following IFNP MB dissolution in blood, the shell constituents persisted as large nanoparticles and insoluble aggregates (>100 nm) which subsequently caused substantial material toxicity and tissue damage probably due to adverse hydrophobic interactions and organ deposition (Supplementary Fig. 1). Thus, we hypothesized that fabricating MB shells using low-molecular weight (MW) and more-hydrophilic polymers that revert to soluble molecular constituents at physiologic pH would minimize material toxicity by decreasing non-specific biologic interactions and facilitating clearance[27–29]. However, our previous approach of IFNP[20] could not be used to construct MB gas carriers using water-soluble low-MW polymers.

To circumvent this limitation, we developed a broadly applicable method of pH-induced interfacial crosslinking to create stable, pH-responsive PMBs using low-MW and water-soluble polymers in aqueous solution without the use of an organic co-solvent. At blood pH, PMB shells revert to their molecularly soluble components (Fig. 1a), a US Food and Drug Administration (FDA)-endorsed strategy to minimize the safety risks of injectable nanomaterials[30]. To demonstrate this approach, we first selected a low-MW dextran (6 kDa) as starting material and optimized its chemical modification with acetyl and carboxyl moieties (LmD) (Supplementary Fig. 2) to create a pH-responsive amphiphile that is soluble at physiologic pH but insoluble in an acidic environment (Fig. 1a). To prepare PMBs, the LmD polymer was dissolved in a pH-adjusted dextrose solution (pH 6.5) and homogenized at the air–water (a/w) interface, while the mixture was slowly acidified using dilute hydrochloric acid (target pH ~3.5). Homogenization created bubble templates to promote absorption of LmD at the a/w interface due to its amphiphilic character, while LmD was crosslinked via the acid-induced protonation of carboxylic groups. This process resulted in the formation of a thin-shell (<50 nm) MB around the gas core (Fig. 1b). Interfacial crosslinking was driven by intermolecular hydrogen bonding and other Van der Waals interactions. The infrared absorption of C = O of the carboxylic acid groups in polymer shells (Fig. 1c and Supplementary Fig. 3) revealed that a majority of them were in bonded states (1,727 $cm^{-1}$, 1,734 $cm^{-1}$) along with a smaller fraction of unbonded ones (1,740 $cm^{-1}$)[31]. The yield of PMBs significantly increased with dextrose concentration (Fig. 1d), probably due to its promotion of polymer aggregation and greater density at the a/w interface at lower pHs (Supplementary Fig. 4). The PMB shell revealed a hydrogel-like property and osmotic balance with the surrounding fluid; washing PMBs manufactured in 30% dextrose with pure water caused swelling of the shell and water influx with replacement of the gas core (Fig. 1e); nonetheless, the aseptically fabricated PMBs with 30% dextrose can be washed and stored in 10% dextrose (D10), a standard clinical IV fluid. Packing density (and therefore gas carrying capacity) of the foam was optimized by varying homogenization speed, with an optimal gas fraction of 62 ± 2% (volume of gas/volume of foam) and a mean particle diameter of ~5 μm (Fig. 1f,g). Further details on PMB fabrication are provided in Supplementary Figs. 5–7. Similar to previously described polymeric microparticles[16,20], the thin shells of PMBs are highly gas permeable, allowing air-filled PMBs (aPMBs) to be readily converted to oxygen-filled PMBs (oPMBs) through passive purging of the headspace with oxygen. oPMBs can be stored in a closed container at room temperature and are stable for months (Fig. 1h). The long-term stability of oPMBs was evaluated via an accelerated stability test at various temperatures. As shown in Supplementary Figs. 8–14, oPMBs stored in glass syringes were stable up to 45 °C for up to 30 days without changes in foam volume or size distribution, while loss of oPMBs was observed at 60 °C. On the basis of these results, we expect the shelf life of oPMBs

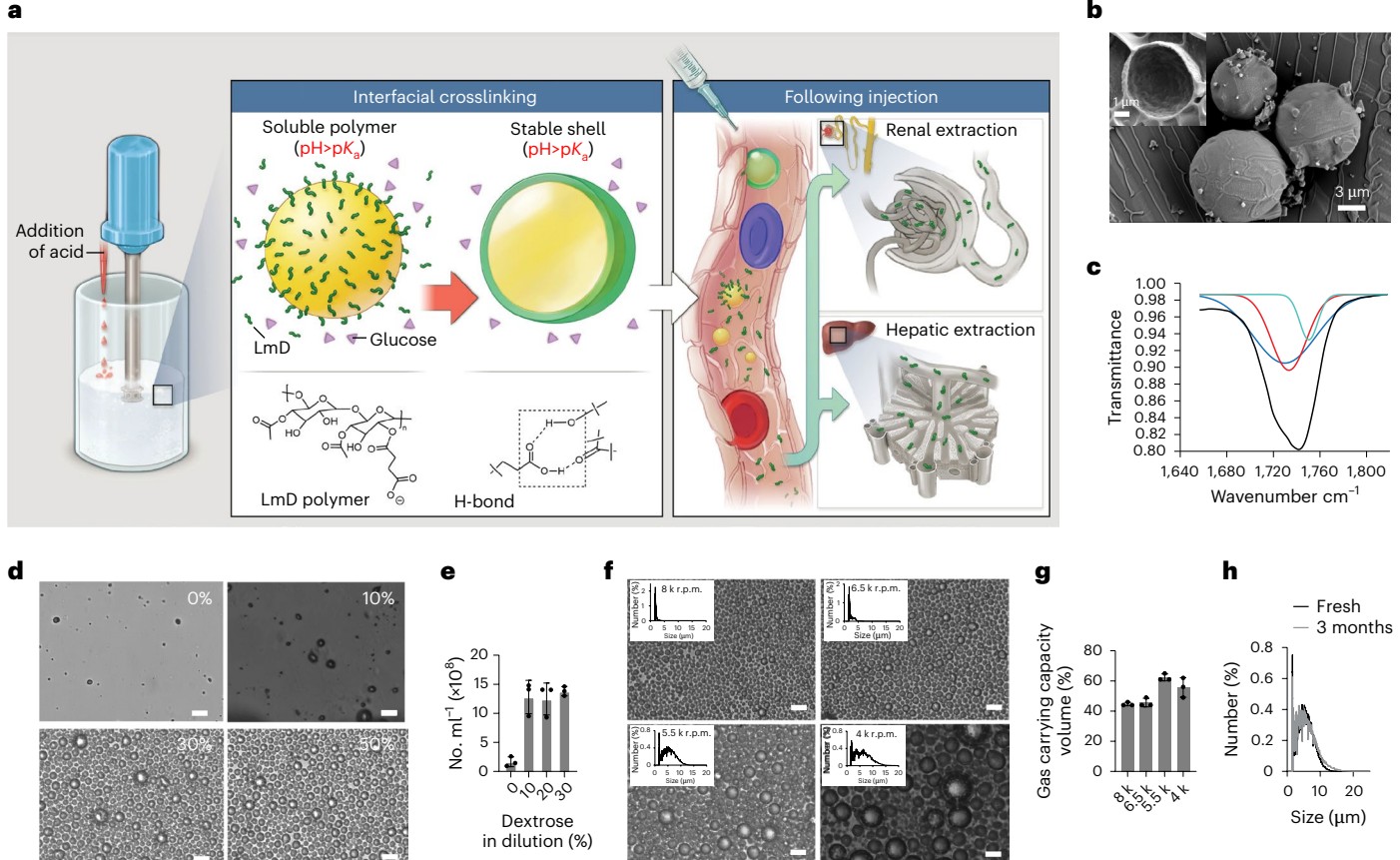

**Fig. 1 | Design and fabrication of LmD PMB gas carriers for IVO$_2$ therapy.**
**a**, LmD PMBs are manufactured through a process of homogenization and simultaneous titration of acid for interfacial crosslinking of the LmD polymer, which is solidified through hydrogen bonding. Following contact with blood, the PMBs rapidly dissolve to deliver gas and their shells immediately revert into their low-MW and soluble molecular constituents, which are excreted via urine and hepatic clearance. **b**, Cryo-scanning electron microscopy imaging of LmD PMBs depicts their thin shell and smooth surface. **c**, The infrared absorption peak of carbonyl groups (black curve) in LmD PMBs indicates that they exist in various H-bound states (deconvoluted Gaussian peaks by taking second derivative; blue and red, bound; green, unbound). **d**, Light microscopy of PMB

solutions homogenized in varying dextrose concentrations illustrates that PMB concentration (that is, yield) increases with increasing dextrose additives. Scale bars, 10 μm. **e**, The effect of osmolarity of carrier fluids on LmD PMBs that were fabricated under 30% dextrose. LmD PMBs originally made with 30% dextrose were not stable in water but were stable in D10 and solutions with higher dextrose concentrations. Data are means ± s.d., biological replicates. **f**, Size distributions (insets) and microscopy of LmD PMBs fabricated under various homogenization speeds under 30% dextrose. Scale bars, 10 μm. **g**, The gas carrying capacity of LmD PMB foams fabricated under various speeds. Data are means ± s.d., biological replicates. **h**, The aseptically fabricated LmD PMBs stored in D10 at room temperature did not change size distribution after 3 months.

to be at least 6 months at room temperature and at least 1 year under refrigerated conditions. The sterility of each manufactured lot was maintained and monitored before in vivo use. To show the material tunability of this system, we also demonstrated the application of this approach to prepare PMBs from a low-MW carboxylated hydroxyethyl starch polymer (Supplementary Fig. 15). The remainder of this work describes our evaluation with LmD PMBs.

**Dissolution of PMBs ex vivo**

The intravenous administration of a gas requires the rapid dissolution of gas carriers to avoid vascular obstruction. Subsequently, the carrier materials must be biocompatible and rapidly cleared. As noted earlier, the LmD polymer itself exhibits pH-responsive behaviour in solution due to the presence of carboxylic acid groups (p$K_a$ ~ 4.8) and is molecularly soluble above pH 5 (Fig. 2a and Supplementary Fig. 16). In this work, the mechanism of dissolution of PMBs following injection hinges on the pH-based deprotonation of carboxylic groups, which increases the solubility and hydration of polymers that compose the shell. This causes water influx into the shell, increasing surface tension and destabilizing the gas core, promoting its dissolution. The deprotonated shell simultaneously reverts to small and soluble components.

Notably, unlike lipid-coated bubbles, which require a gas concentration gradient (that is, sink) to dissolve[24–26], PMBs dissolve at a physiologic pH within seconds even in the absence of a sink (Supplementary Video 1). To examine whether PMB shells fully dissolve and revert to soluble LmD constituents, we added PMBs into phosphate buffered saline (PBS) solution of varying pH under stirring and confirmed dynamic light scattering (DLS) size measurements 2 min following admixture (earlier timepoints were not obtained by DLS due to the sampling limitation of the instrument). Between pH 9.0 and 5.0, PMB shells were all fully dissolved within 2 min, reverting to soluble polymers of similar sizes as those in LmD solutions prepared from solid states (Fig. 2a). Solubilized LmD polymer has a mean hydrodynamic radius less than 10 nm and an estimated MW of ~12 kDa (determined by nuclear magnetic resonance in Supplementary Fig. 2), well below the MW cut-off for glomerular filtration (30–45 kDa)[32,33]. In contrast, dissolution of previous IFNP MBs, which consist of hydrophobic polymers of higher MW (>60 kDa), led to formation of large and insoluble nanoparticles (>100 nm) that visibly precipitate over time.

To better investigate the pH-dependent dissolution kinetics, we examined PMBs mixed with PBS while continuously applying ultrasound. Similar to various polymeric-shelled MBs[34,35], the gas core of

PMBs creates acoustic backscatter and produces contrast in proportion to the presence of gas bubbles within the field of view. The decrement in contrast intensity (that is, bubble dissolution) was shown to be pH dependent: at pH 9.0, 7.2 and 6.5, PMBs were no longer visible within 2–3 s, while dissolution of the gas core was prolonged at pH < 6 (Fig. 2b,c). (Of note, although the LmD shell is less soluble at pH 4.8 and 3.8, we noticed that the gas core of PMBs slowly becomes fluid filled as shown by the slow decrease in echo intensity, probably because the salts in PBS affected the swelling of the hydrogel-like shells). While it is known that ultrasound may contribute to loss of MBs due to inertial cavitation, Fig. 2b suggests that pH is the dominant factor affecting dissolution rate. To further account for acoustic destruction, we performed the same experiments while applying ultrasound only at selected terminal timepoints, finding similar dissolution rates (Fig. 2d,e).

To further examine dissolution of the shell (separate from that of the gas core), we performed UV-Vis spectroscopy. In this construct, it is expected that an increase in absorbance from baseline could be caused by either undissolved gas cores or large polymeric aggregates (that is, undissolved shell or aggregated constituents). From pH 9.0 to 6.0, UV absorbance reached baseline within 2–3 s, similar to the kinetics in acoustic studies, indicating both that the gas core had dissolved, and the shell had reverted to its soluble constituents in that time (Fig. 2f,g). Between pH 5.5 and 5.0, the return to baseline was much longer than in the acoustic study, suggesting that within this pH range the gas core dissolves first, and the shell requires more time to revert to soluble polymers. These findings are consistent with the pH-triggered dissolution mechanism of PMBs that is essentially an acid–base reaction, the rate of which is proportional to the concentration of hydroxyl ions in the solution and limited by diffusion. This mechanism also explains some discrepancies in the dissolution kinetics seen at lower pHs. For example, in contrast to UV-Vis, DLS showed that PMBs were fully dissolved at 2 min at pH 5.5 and 5.0, probably due to lack of sufficient mixing in UV-Vis experiments. However, at pH > 6.0, the dissolution kinetics measured from various methods were all in good agreement. Collectively, these results validate our proposed design for the new gas carrier and established that both the gas core and the shell of PMBs rapidly dissolve at pH levels (7.5–6.5) that are relevant to intravenous injection, as the blood pH rarely drops below 6.5 even in extreme instances.

## Effects of PMBs on acute haemodynamics

Previous injectable gas carriers that did not exhibit a triggered dissolution mechanism caused pulmonary vascular obstruction due to bubble persistence following injection (Fig. 3a,b). Any intravenously injected fluid will immediately travel from the injection site to the right atrium and ventricle, from which it is then ejected into the pulmonary circulation before returning to the left heart to enter the arterial system. The pulmonary capillaries are the smallest blood vessels and thus have the highest susceptibility to occlusion[36]; if particulate matter or gas embolism from gas carriers obstructs pulmonary capillaries, pulmonary vascular resistance (PVR) increases[37]. To provoke vascular obstruction in this model, we used air-filled aPMBs rather than oPMBs, since the driving gradient for oxygen egress exceeds that of nitrogen due to the high concentration of deoxyhaemoglobin in the venous system. Following a baseline period, rats in the test group ($n$ = 4, weight 503 ± 52 g) received 5 repeated injections of 5 ml 50% volume gas/volume foam in D10 (~2.5 ml gas per injection) every 3 min (total 12.5 ml of air) followed by a 60-min observation period. The control group ($n$ = 5, 518 ± 40 g) received an equal volume of D10. The injected gas content was ~1.7 ml kg$^{-1}$ min$^{-1}$, representing the equivalent of 50–100% of basal oxygen consumption of a human[38]. In both groups (Fig. 3c,d), PVR decreased during the injection period relative to baseline, and there were no differences between groups (−64.5 ± 40.1 mmHg ml$^{-1}$ kg$^{-1}$ min$^{-1}$ for PMB versus −45.2 ± 27.2 mmHg ml$^{-1}$ kg$^{-1}$ min$^{-1}$ for the control, $P$ = 0.41), probably representing preload recruitable stroke work and increased cardiac index in both groups (Extended Data Fig. 1). Relatedly, mean arterial blood pressure (MABP) increased similarly in both groups during the injection period (Fig. 3e,f) (53.0 ± 20.6% for PMB versus 57.6 ± 21.7% above baseline, $P$ = 0.69). Taken together, this acute haemodynamic profile suggests an absence of pulmonary obstruction following serial, rapid PMB injections, and is distinct from the profile seen with rapid injection of LOMs and gas carriers composed of poly(lactic-co-glycolic acid)[16,21,26].

To verify the dissolution of PMBs in vivo, we performed transthoracic echocardiography following injection of PMBs versus LOMs (Fig. 3g and Supplementary Video 2). aPMBs or oPMBs (70% foam) were continuously infused at various rates up to 12 ml gas kg$^{-1}$ min$^{-1}$; oxygen-filled LOMs (50% foam, equal gas volume) were infused only at the lowest rate (4 ml gas kg$^{-1}$ min$^{-1}$), as higher rates caused haemodynamic collapse. Injection of LOMs immediately opacified all four heart chambers (Fig. 3h and Supplementary Video 3), demonstrating transpulmonary passage of undissolved LOMs. In contrast, aPMBs and oPMBs were not visible in the LV even when injected at a 3× higher rate (Fig. 3i–l, Supplementary Figs. 17 and 18 and Supplementary Videos 4 and 5). Compared with LOMs, opacification of the right ventricle was also less pronounced following injection of aPMBs and oPMBs injected at an equivalent rate, providing evidence for their rapid in vivo dissolution. Right heart opacification disappeared within seconds following the end of PMB injections, whereas circulating LOMs were still visible 10 min after injection, highlighting their in vivo persistence[26]. Taken together, these data suggest that PMBs dissolve rapidly following even rapid injection and do not cause vascular obstruction even in the absence of a sink, while delivering a high gas payload.

## Efficacy of PMBs in hypoxaemia-related cardiac arrest in swine

Having demonstrated the acute safety of PMBs, we assessed their effect on a clinically realistic, extreme model of hypoxaemic respiratory

---

**Fig. 2 | LmD PMBs rapidly dissolve at physiologic pH. a**, DLS measurement of size following mixing of LmD PMBs in PBS solution for 2 min at varying pH. LmD PMBs fully dissolve above pH 5 and revert to their soluble components with a mean size <10 nm, similar to those of LmD solutions prepared from solid states. In contrast, the previous generation of IFNP MBs (made from more hydrophobic polymers) led to formation of much larger nanoparticles. **b**, Phantom sonography of aPMBs in aerated PBS shows the pH-dependent dissolution rate, evidenced by the disappearance of contrast intensity produced by the gas core under continuous ultrasound. The black dashed line indicates the timepoint when aPMBs were administered. In the absence of a gas sink, PMBs rapidly dissolved above pH 6 within seconds. Data (means ± s.e.m.) presented as change in contrast/bright area from baseline. (Of note, although the LmD shell is less soluble at pH 4.8 and 3.8, we noticed that the gas core of PMBs slowly becomes fluid filled as shown by the slow decrease in echo intensity, probably because the salts in PBS affected the swelling of the hydrogel-like shells.) **c**, Representative images from the phantom sonography study showing the dissolution profile of aPMBs at different timepoints at various pHs. BL, baseline. **d,e**, To account for any destructive effect that ultrasound itself has on PMBs, the experiment was repeated while only applying ultrasound at the expected dissolution time from **b**, showing similar dissolution times even without the application of continuous ultrasound (**d**). **e**, Data (means ± s.e.m.) presented as change in area of contrast/brightness from baseline, analysed using Student's $t$-test. Cont., continuously applied ultrasound; Inter., intermittently applied ultrasound; NS, not significant. **f,g**, Dissolution of the shell and gas core was then studied using UV-Vis absorbance spectroscopy. Similar to the characterization using ultrasound (which detects only dissolution of the gas core), UV-Vis returns to baseline within seconds at pH above 6 (**g**), suggesting that the gas core has dissolved and that the shell has broken down into its constituent components. At more acidic pH, return of UV-Vis absorbance to baseline took 10 min or longer (**f**). Contrasting this with sonographic experiments (**b**) in which ultrasound scatter returned to baseline within 90–260 s, these findings suggest that following dissolution of the gas core, the remaining shell constituents take additional time to dissolve and revert to soluble components. All repeated measurements are biological replicates.

failure and IHCA (Fig. 4a). Briefly, Yorkshire swine were anaesthetized and instrumented, including tracheal intubation and placement of arterial and venous catheters. Following a period of observation under IV sedation and neuromuscular blockade while breathing 21% oxygen, the swine underwent 12 min of apnoea/asphyxia. Animals experiencing cardiac arrest (defined as systolic blood pressure <40 mmHg for

5 s or longer)[39] received high-quality, chest compression CPR and rhythm-directed resuscitative interventions, including medications and defibrillation according to current standards[6]. At minutes 6, 8 and 10 of injury, swine were randomized to receive either IVO$_2$ (combined total, 400 ml of 35% vol O$_2$/vol foam oPMBs, containing ~140 ml oxygen, $n = 8$) or an equal volume of oxygenated D10 ($n = 10$). Given that the

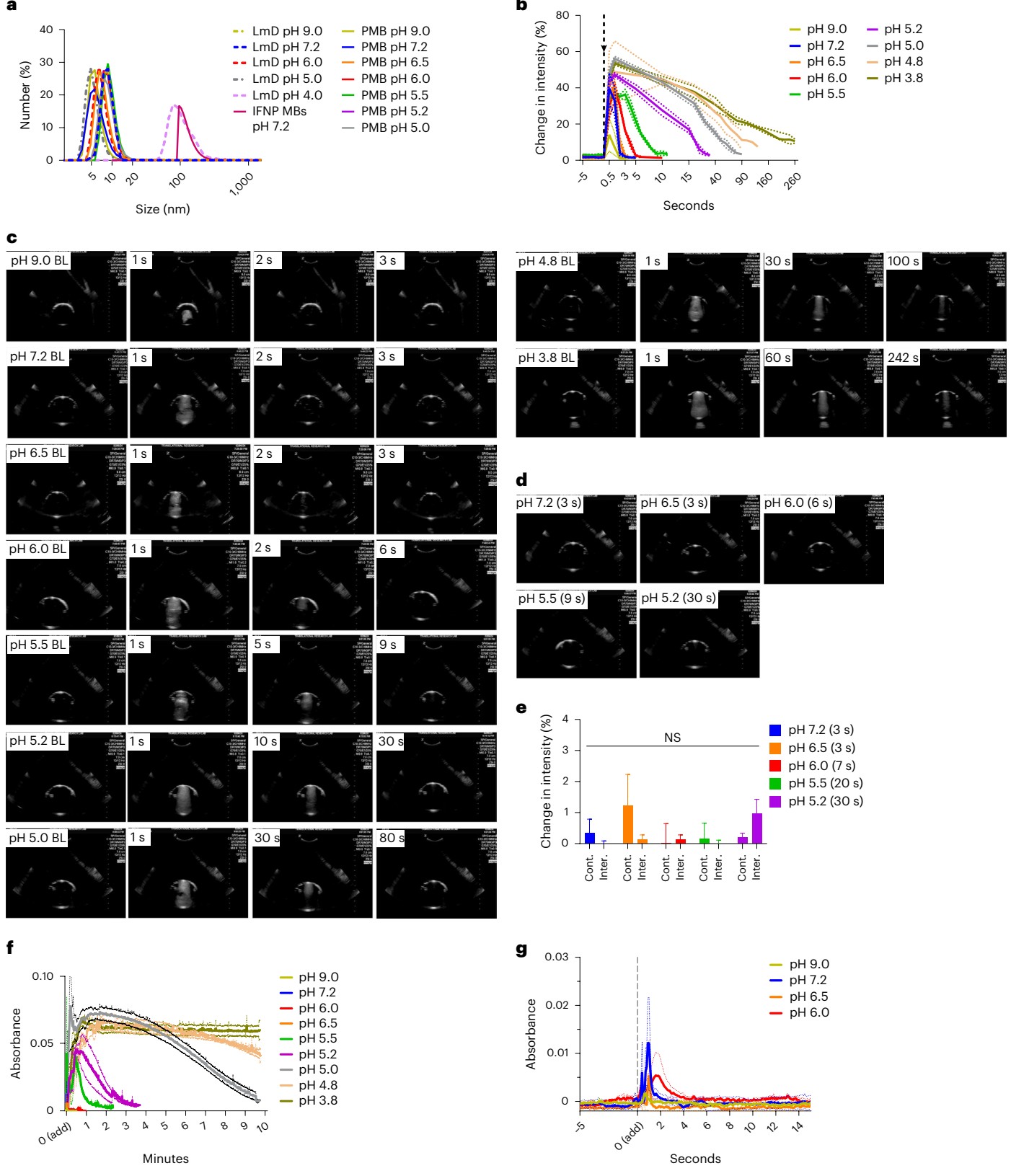

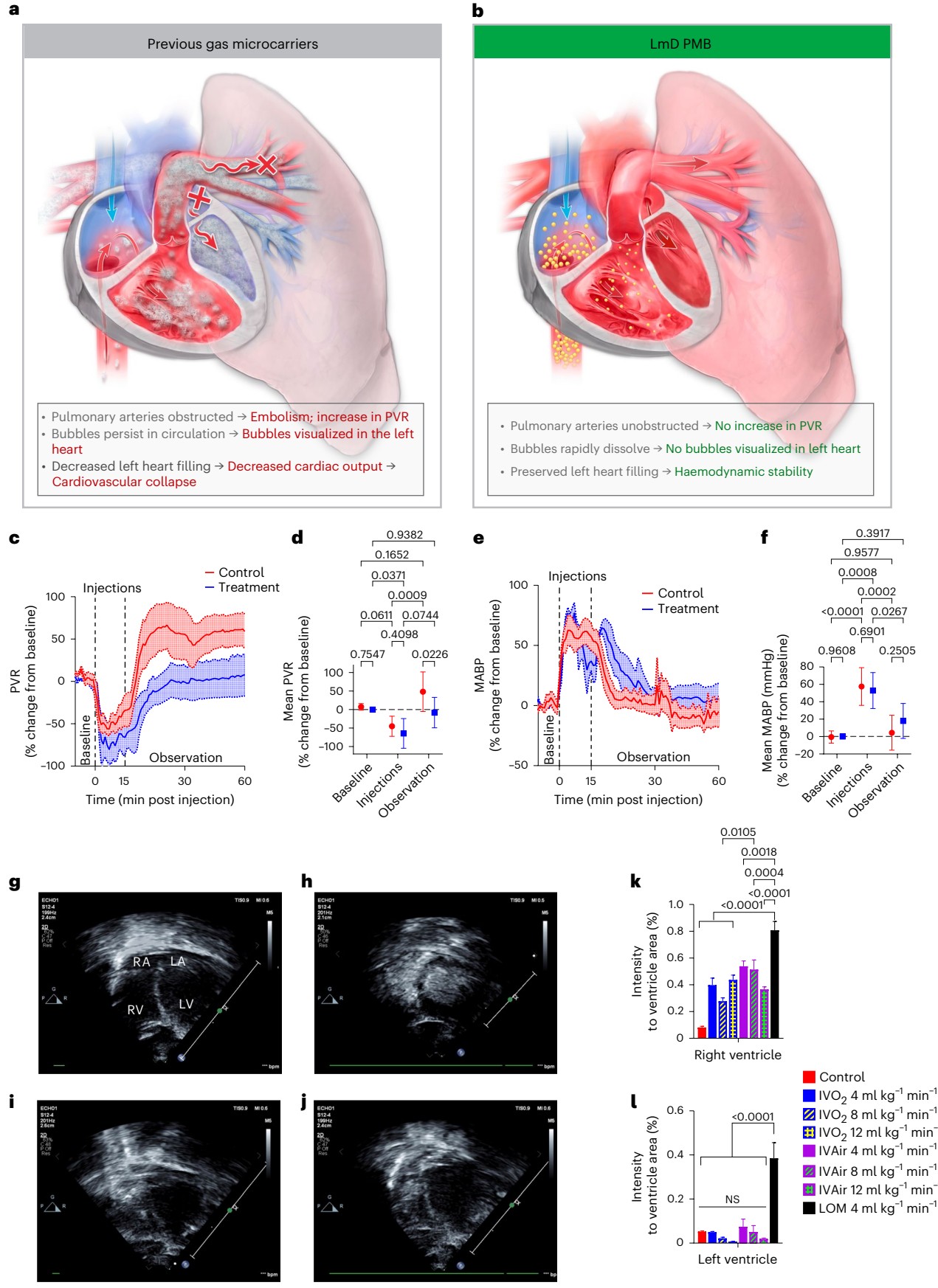

**Fig. 3 | The pH-triggered, rapid dissolution mechanism of LmD PMBs avoids vascular obstruction and haemodynamic instability. a**, The rapid dissolution of PMBs is critical to their in vivo safety. Previously described gas carriers did not dissolve rapidly or coalesced following injection, leading to pulmonary vascular obstruction. **b**, In contrast, PMBs dissolve so rapidly following injection that they exist mainly as soluble molecular constituents by their first contact with the pulmonary circulation. **c**–**f**, Haemodynamic safety study. $n = 4$ for treatment, $n = 5$ for D10 control. Continuous measurements collected as biological replicates. PVR (**c**) was not significantly different from baseline following 5 injections of 5 ml 80% aPMBs (50% vol air/vol foam) over 1 min each. Data (means ± s.e.m.) presented as mean percent change from baseline. Mean PVR during each experimental period (**d**) did not change during or following infusions of aPMBs during either the injection or observation period. Data are means ± s.e.m., $P$ values calculated using two-way ANOVA. MABP (**e**) increased during injection of PMBs and returned to baseline thereafter. Data (means ± s.e.m.) presented as mean percent change from baseline. Mean MABP during each experimental period (**f**) did not change during or following

infusions of aPMBs during either the injection or observation period. Data are means ± s.e.m., $P$ values calculated using two-way ANOVA. **g**–**j**, Representative transthoracic echocardiography images during infusion of PMBs and LOMs through the left parasternal window for a four-chamber view. **g**, Control animal, four-chamber view with right atrium (RA), left atrium (LA), right ventricle (RV) and left ventricle (LV). **h**, High opacification in both left and right ventricles of an animal injected with LOMs at a rate of 4 ml kg$^{-1}$ min$^{-1}$. **i**, Moderate opacification in the right ventricle of an animal injected with IVO$_2$ of oPMBs at a rate of 4 ml kg$^{-1}$ min$^{-1}$, with no visible signal noted in the left chambers. **j**, Moderate opacification in the right ventricle of an animal injected with intravenous aPMBs (IVAir) at a rate of 4 ml kg$^{-1}$ min$^{-1}$, with no visible signal noted in the left chambers. **k**,**l**, Percentage of opacified areas in the right (**k**) and left (**l**) ventricles relative to respective ventricle area was quantified during administration of IVO$_2$, IVAir and LOM at the flow rates shown, each during a 1-min infusion. Data are means ± s.e.m., comparisons using one-way ANOVA with Tukey's multiple comparisons test.

mean measured resting oxygen consumption (VO$_2$) during the baseline period in this experiment was 73.2 ml min$^{-1}$, this volume represents the provision of ~30% of resting oxygen consumption for the last 6 min of asphyxia. After 12 min, the airway was opened and ventilation restored. CPR was continued for up to 30 min or until ROSC. Surviving animals were then maintained for 4 days in an intensive care unit (ICU) environment, including mechanical ventilation, extubation readiness testing, inotropic support, and seizure monitoring and treatment according to a standardized protocol. On day 4, brain magnetic resonance imaging (MRI) was performed, followed by euthanasia and pathological analysis.

**Effect of oPMBs on resuscitation metrics.** There were no differences in baseline characteristics between groups, including age (43 days oPMB versus 49 days control, $P = 0.08$) or weight (12.1 ± 1.1 versus 12.0 ± 1.3 kg, $P = 0.953$). At 6 min of asphyxia/apnoea, a similar number of animals experienced cardiac arrest (Supplementary Fig. 19). CPR quality was excellent in both groups with no significant differences in compression rate, compression fraction or compression depth (Supplementary Fig. 20). Arterial oxygen saturation (SaO$_2$; measured by co-oximetry on blood gas every odd minute) reached undetectable levels (<3%) at 6 min, and was significantly higher in IVO$_2$-treated swine than in those receiving control at 9 min (24 ± 14% versus 4 ± 2%, $P = 0.012$), 11 min (23 ± 7% versus 3 ± 1%, $P = 0.017$) and 13 min (96 ± 8%

versus 77 ± 36%, $P = 0.024$; Fig. 4b). Similarly, the partial pressure of oxygen in arterial blood (PaO$_2$) was significantly higher in IVO$_2$-treated swine at 7, 9 and 11 min (Supplementary Fig. 21). Arterial carbon dioxide tension was higher at 7, 9, 11 and 13 min in IVO$_2$-treated swine ($P < 0.05$; Fig. 4c), presumably due to preserved cellular metabolism and CO$_2$ production (although this may also be partially explained by the Haldane effect). Although all swine experienced cardiac arrest, IVO$_2$ restored circulation during asphyxia in a number of IVO$_2$-treated swine (Fig. 4d), such that the duration of CPR (Fig. 4e) and resuscitative doses of epinephrine indicated in the protocol (Fig. 4f) were significantly lower in IVO$_2$-treated swine; IVO$_2$-treated swine also had significant improvements in MABP during the treatment period (Fig. 4g). Following relief of airway obstruction, swine in the IVO$_2$-treated group were more likely to achieve ROSC (100% versus 30%, $P = 0.003$) and overall survival (88% versus 30%, $P = 0.007$) (Fig. 4h).

**Effect of oPMBs on organ injury.** Only 3 out of 10 swine in the control group achieved ROSC, all of which experienced severe neurologic injury: none extubated successfully, and all experienced refractory status epilepticus and diabetes insipidus (a phenomenon indicative of profound brain injury) (Supplementary Fig. 22). Of the 8 achieving ROSC in the IVO$_2$-treated group, 5 were successfully extubated within 24 h and were able to ambulate and eat and drink independently by

**Fig. 4 | IVO$_2$ via LmD oPMBs improves survival and meaningful outcomes in a swine model of severe hypoxaemic respiratory failure. a**, Study timeline. IVO$_2$ treatment (oPMBs) ($n = 8$) or control (D10) solution ($n = 10$) was administered at minutes 6, 8 and 10. ETT, endotracheal tube. **b**, IVO$_2$ rapidly and significantly increased arterial oxyhaemoglobin saturation (SaO$_2$) during the asphyxial period (grey shading). **c**, Arterial carbon dioxide tension (pCO$_2$) during asphyxia was significantly higher in IVO$_2$-treated swine. **b**,**c**, Groups compared using two-way ANOVA with Sidak's multiple comparisons, with only significant $P$ values shown. **d**, IVO$_2$ treatment increased the fraction of animals free of cardiac arrest and CPR during and post asphyxia. Treatment period, blue shading. **e**,**f**, IVO$_2$ treatment significantly decreased CPR time (**e**) and the required dose of epinephrine (**f**) used during resuscitation. Groups compared using Student's $t$-test. **g**, IVO$_2$ treatment improved MABP during resuscitation. Groups compared using two-way ANOVA with Sidak's multiple comparisons, with only significant $P$ values shown. **h**, IVO$_2$ treatment significantly improved ROSC at 30 min (log-rank test $P = 0.003$; Gehan–Breslow–Wilcoxon test, $P = 0.003$) and 84-h survival (log-rank test $P = 0.013$; Gehan–Breslow–Wilcoxon test, $P = 0.007$). **i**, IVO$_2$ treatment significantly improved the SNDS in surviving swine. ACA, asphyxial cardiac arrest; POD, post-operative day. Groups compared using two-way ANOVA with Sidak's multiple comparisons. **j**, GFAP, a marker of astrocyte injury, was significantly elevated at day 3 in the control group, whereas no difference from baseline was observed in the treatment group. Groups compared using two-way ANOVA with Sidak's multiple comparisons test. **k**, Representative weighted T2 MR image at 84 h post asphyxia depicts total grey

matter and white matter diffusion restriction (supratentorial/infratentorial) with T2 prolongation throughout the cortex. **l**, Representative image of an IVO$_2$-treated swine reveals faint T2 prolongation in the basal ganglia. **m**,**n**, Three-dimensional representation of the median injury from brain MRI in control (**m**) versus treated (**n**) animals. Areas of enhancement on axial and coronal T2 and diffusion coefficient images were manually processed on a voxel-per-voxel basis. **o**, Volume of abnormal enhancement on T2 and diffusion coefficient images was significantly lower in IVO$_2$-treated swine than in surviving control swine. Comparison using Student's $t$-test. **p**, Representative gross photos from the control group showed swollen, friable brain tissue with severe maceration of the ventral surface, and their pathological sections showed an overall dusky colour, blurring of the grey–white junction and intraventricular discoloration; in contrast, representative photos from the IVO$_2$-treated group revealed well-preserved brain tissue with few apparent abnormalities. **q**, Histologic injury score was statistically significantly lower in the basal ganglia structures in IVO$_2$-treated swine than in controls. Scoring: 0, no damage; 1, rare hypereosinophilic neurons; 2, clusters of hypereosinophilic neurons; 3, >50% of neurons are hypereosinophilic; 4, >90% of neurons are hypereosinophilic; 5, cavitated infarction. Groups compared using two-way ANOVA with Sidak's multiple comparisons test. DG, dentate gyrus; PL, pyramidal layer. **r**,**s**, BUN (**r**) and creatinine (**s**) were significantly higher in the control group on day 3 than in IVO$_2$-treated swine. Note that **i**–**s** reflect data collected only in surviving swine, which omits 7 of the 10 swine in the control group that did not survive. For all figures, data are means ± s.d.; measurements are biological replicates.

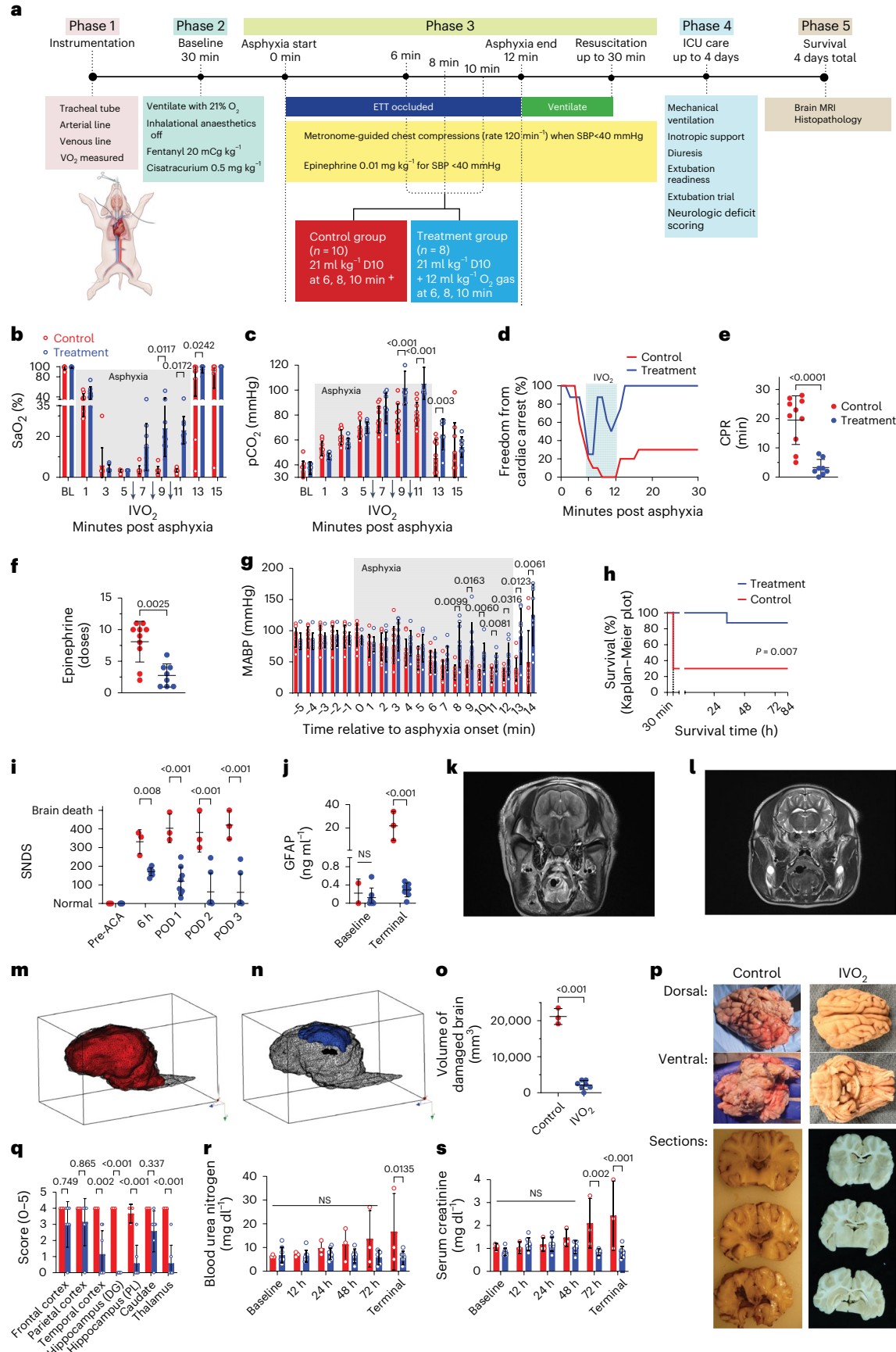

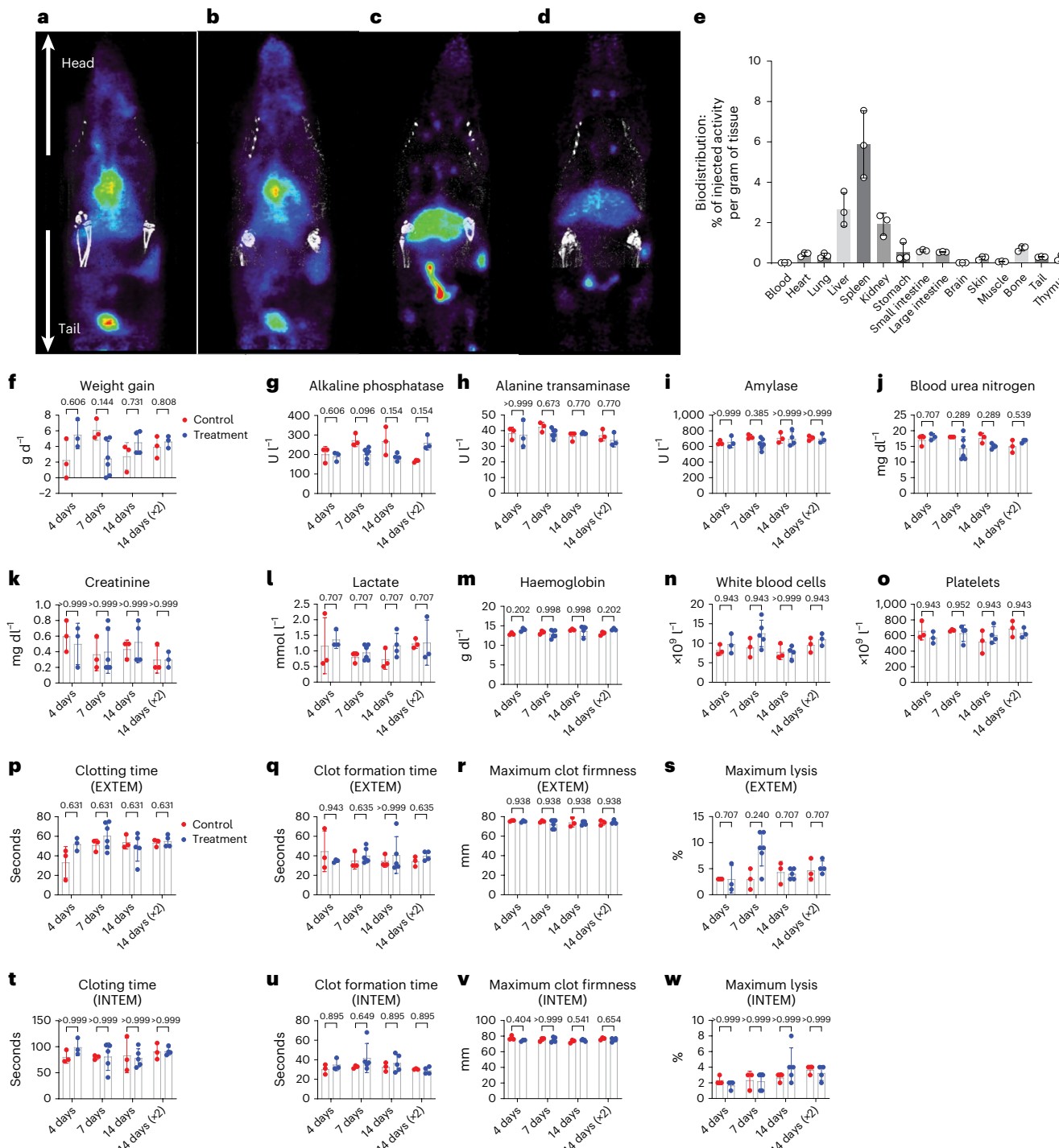

**Fig. 5 | Biodistribution and safety study of LmD PMBs in rodents.**
**a**–**d**, Representative PET/CT images of rats receiving [89]Zr-labelled polymers over time. **a**, Immediately after infusion, radioactivity was observed in the upper abdomen, liver and kidneys, with a significant portion being excreted via bladder. **b**, Continuous excretion via urine and hepatic clearance at 24 h. **c**, Bowel excretion continues via hepatic clearance at 48 h, with maximum accumulation in stools. **d**, Low radioactivity level on day 7, with continuous excretion via faeces. **e**, Biodistribution on day 7, with the residual polymer presented as injected activity per gram of a particular organ (n = 3; measurements are biological replicates). **f**–**o**, Major clinical markers for organ injury and toxicity were normal in animals receiving LmD PMBs compared with control group (n = 3–6 per group) in rodent safety study. Dosage 1, 32 ml of 70% oPMBs (40% v/v oxygen) per kg, equivalent to the efficacy dose, with endpoints at three timepoints (4, 7 and 14 days), and control groups receiving equal volume of D10. Dosage ×2,

doubling of dosage 1, administered 30 min apart, a total of 64 ml of 70% oPMBs (40% v/v oxygen) per kg, 14-day single timepoint, with control group receiving equal volume of D10. Data are means ± s.d., compared using multiple Mann–Whitney tests. Weight gain (**f**). Liver function tests: (**g**) alkaline phosphatase, (**h**) alanine transaminase, (**i**) amylase. Renal function tests: (**j**) BUN, (**k**) creatinine. (**l**) Lactate. Complete blood count: (**m**) haemoglobins, (**n**) white blood cells, (**o**) platelets. All values of animals receiving PMBs of both dosages at all timepoints were within normal ranges and showed no significant difference from control groups. **p**–**w**, Coagulation analysis by ROTEM in external coagulation pathway (known as EXTEM) and intrinsic coagulation pathway (known as INTEM) showed that infusion of PMBs did not adversely affect clotting: (**p,t**) clotting time, (**q,u**) clot formation time, (**r,v**) maximum clot firmness, (**s,w**) maximum lysis. Data are means ± s.d., compared using multiple Mann–Whitney tests with q value shown. All measurements are biological replicates.

3 days post injury (Supplementary Fig. 22). Even among only surviving swine, Swine Neurologic Deficit Scores (SNDS) were significantly lower at post-injury days 1–3 in $IVO_2$ treated swine (Fig. 4i). Glial fibrillary acidic protein (GFAP), a brain astrocytic protein released in proportion to brain cellular injury[40], was nearly two orders of magnitude higher at 4 days in surviving control swine than in those receiving $IVO_2$ ($0.3 \pm 0.2$ in $IVO_2$ versus $21.8 \pm 12.4$ ng ml$^{-1}$ in controls, $P < 0.001$) (Fig. 4j). None of the 3 surviving control animals exhibited any detectable intracranial blood flow by magnetic resonance angiography, and all had evidence of a generalized oedema and uncal and tonsillar herniation (Fig. 4k). When assessed using a manual segmentation of the diffusion imaging, $IVO_2$-treated swine exhibited significantly lower volumes of white matter injury (2,269 (IQR 1,560–3,533) mm$^3$) compared with controls (20,784 (19,154–23,556) mm$^3$, $P < 0.001$) (Fig. 4k–o). The degree of brain injury in control animals was notable by gross examination of the brain tissue (Fig. 4p and Supplementary Fig. 23). Histological sections showed that injury in the control group was widespread in all areas, while injury in the treated animals was significantly attenuated, with limited focal injury in the cerebral cortex and basal ganglia (Fig. 4p,q and Supplementary Fig. 24). The hypoxic-ischaemic injury score in the control group was significantly higher than in the treated group (overall injury score $27.7 \pm 0.6$ versus $11.0 \pm 4.4$, $P < 0.001$; Fig. 4q and Supplementary Fig. 25). Further, $IVO_2$-treated swine had significantly less renal injury based on blood urea nitrogen (BUN) (Fig. 4r), creatinine (Fig. 4s) and histologic analysis (Supplementary Fig. 26). There were no significant differences in the lab or histologic manifestations of injury in other organs (Extended Data Fig. 2 and Supplementary Figs. 27–29). These results demonstrate that oPMBs effectively reverse hypoxaemia and, through a moderate increase in blood oxygenation, significantly decrease the burden of cardiac arrest, decrease mortality and diminish hypoxic-ischaemic injury in acute, severe hypoxaemia.

## Safety study

The treatment of swine with oPMBs was well tolerated and the animals showed no clinical sign of adverse effects. To probe the biodistribution and pharmacokinetics of the PMB constituents, we performed positron emission tomography with computed tomography (PET/CT) imaging for 7 days following injection of $^{89}Zr$-labelled LmD polymers in healthy rodents. Following a single tail-vein injection (400 mg kg$^{-1}$, equivalent dose in swine study), the vast majority of LmD polymer was excreted in the urine within 24 h, and the remainder underwent hepatic clearance (Fig. 5a–d). By day 7, low levels of LmD were found in the spleen ($5.9 \pm 1.5\%$ injected dose per gram), liver ($2.7 \pm 0.7\%$) and kidneys ($1.9 \pm 0.5\%$). Based on empirical calculations[41], >75% of injected polymer had been cleared, with the majority of the remainder visualized in the bowel lumen (that is, in the process of being excreted) (Fig. 5e). These results support our design hypothesis that the use of low-MW and more-hydrophilic polymers greatly facilitates clearance. To further assess the potential adverse effect of PMBs, we conducted an up to 14-day safety study in healthy rodents following a tail-vein injection of a single dose of oPMBs (32 ml kg$^{-1}$, 70% foam, equivalent to the efficacy dose used above, $n = 12$ oPMB, $n = 9$ D10 control) sacrificed at 4-, 7-, 14-day timepoints, and a double dose administered 30 min apart (a total of 64 ml kg$^{-1}$, $n = 3$) sacrificed at 14-day timepoint; control animals ($n = 3$) received equal volumes of D10. Throughout the observation period, all animals survived and exhibited normal behaviour, and had normal urine output and weight gain (Fig. 5f). There were no differences between groups, at any timepoint, in blood gas, chemistry, complete blood count or hepatic function testing (Fig. 5g–o and Supplementary Fig. 30); there were no signs of clinically observable immune toxicities or platelet dysfunction. Further, rotational thromboelastometry (ROTEM) showed that PMBs did not affect either intrinsic or extrinsic coagulation pathways, with normal coagulation time, clot formation time, maximum clot firmness and maximum lysis at all timepoints (Fig. 5p–w). Histological analysis of major organs in

both groups showed similar and normal morphologies, except that spleen macrophage vacuole formation was observed in treated animals. This was probably due to splenic polymer uptake, which, interestingly, was not observed in swine (Supplementary Fig. 31). These results together support that PMBs revert to soluble low-MW components after IV administration and are well tolerated in clinically relevant doses.

## Discussion

We have described a pharmaceutically viable design of gas carrier to enable the clinical translation of $IVO_2$ therapy. PMBs exhibit high gas carrying capacity, acceptable shelf stability, and manufacturability at sufficient scale and control to ultimately enable a clinical trial. At the gas concentration of the foams described (35–50 ml $O_2$ dl$^{-1}$ foam), the provision of 100 ml oxygen gas requires the co-administration of 100–185 ml of additional fluid. Unlike previous injectable gas carriers (for example, LOMs), the pH-triggered mechanism of PMBs actively enhances their dissolution and the delivery of the gas payload even in the absence of the diffusion gradient, hence minimizing the risk of vascular obstruction even when administered at a high dose or under conditions of low blood flow. This feature is essential to their use in settings of critical illness and emergencies in which cardiac output and blood flow may vary moment to moment, settings in which other gas carriers will cause gas embolism or particle jamming, pulmonary vascular obstruction and cardiovascular collapse[21,26]. Following dissolution, PMB shells rapidly revert to low-MW, soluble components—a strategy that decreases adverse non-specific hydrophobic interactions, a major contributor of nanotoxicity. Our prototype LmD PMBs show high tolerance and undergo renal and hepatic clearance—a substantial improvement in safety from earlier generations (Supplementary Fig. 1). We also showed that the polymeric structures of shell materials are tunable. The combination of these features results in a general drug design that may enable the clinical intravenous injection of oxygen.

Acute, severe hypoxaemia is a common and life-threatening event among critically ill patients and represents an enormous clinical challenge[42–44]. Among others, it may occur in the setting of tracheal intubation[45], mechanical ventilation (for example, due to secretions)[46] or airway bleeding[47], and can lead to cardiac arrest when not immediately addressed[48]. When hypoxia progresses beyond a critical threshold, mitochondrial reduction occurs, cells become energy deprived and morphologically damaged, and cardiovascular collapse ensues[49]. Outcomes in such patients are dismal[50,51]. We have shown that $IVO_2$ interrupts this lethal cascade for short periods of time until normoxia can be restored by conventional means. Preventing cardiovascular collapse in this setting is paramount for the prevention of neurologic injury. The brain is extremely oxygen avid[52] and exquisitely sensitive to interruptions in oxygen supply, becoming isoelectric after 15–30 s without blood flow[53] and sustaining irreversible injury within minutes[54,55]. In our study, the initial, expected response to very severe hypoxia was a pronounced increase in blood pressure and heart rate (representing a sympathetic nervous system response), which degraded into hypotension and pulseless electrical activity as oxygen substrates became depleted. Intravenous oxygen is synergistically beneficial to circulation in this setting: (1) local hyperoxia in the pulmonary arteries causes pulmonary vasodilation, lowering impedance to blood flow; (2) provision of oxygen for energy generation restores systemic vascular resistance (both arteriolar and venous) and myocardial function, raising perfusion pressure and blood flow; and (3) the residua of the polymer shell expands intravascular volume and myocardial preload, augmenting cardiac output. Together, these effects maintain or quickly restore circulation in the setting of hypoxia-related cardiac arrest. Further, we showed that the dose of oxygen required to accomplish these effects was a small fraction of baseline consumption in health. In critical illness states such as severe hypoxia, oxygen consumption becomes supply limited[56], such that the provision of a given dose of oxygen may have a more prolonged or pronounced effect on cellular metabolism. Further,

a given dose of oxygen may also have a more pronounced effect when used in clinical settings of lung injury. Normally, oxygen flows from the alveolus into the blood, but in this model of airway obstruction, oxygen tension of the pulmonary artery exceeded that of the alveolus, such that oxygen initially diffused backwards, equilibrating with the functional residual capacity of the lung (that is, increasing the volume of distribution of the gas payload); in this sense, this was an exaggerated model of ventilation-perfusion inequality, a central pathology in patients with clinical lung disease. We expect that this phenomenon of back diffusion would be attenuated, and the dose response therefore more pronounced, in patients with more heterogeneous ventilation-perfusion inequality or with an oxygen diffusion gradient. Taken together, these findings highlight the unique pharmacological advantages of $IVO_2$ as a promising new treatment for the rapid reversal of life-threatening hypoxaemia in emergency settings, including prehospital, ICU and operating room environments. We acknowledge that while we have demonstrated benefit in a model of asphyxial cardiac arrest, whether this benefit is replicated in other models of severe hypoxaemia (for example, lung injury) needs to be experimentally determined using different animal models.

In clinical practice, we envision that $IVO_2$ would be available on-demand in environments caring for critically ill patients at risk for hypoxaemia. Because arterial oxygen saturation is often continuously monitored by photoplethysmography in patients at risk for hypoxaemia, $IVO_2$ could become a new treatment for refractory hypoxaemia, that is, refractory to current standards of care, including airway clearance (for example, suctioning), lung recruitment (for example, hand ventilation) and the use of other critical manoeuvres. Its dosing could be titrated in the same way that pre-arrest bolus doses of epinephrine are titrated to treat refractory hypotension[57], since both blood pressure and arterial oxygen saturation (by plethysmography) are routinely measured as vital signs in hospitalized patients. As in the swine model we described, pre-specified doses of $IVO_2$ could eventually be added to the resuscitation algorithm for patients being treated for in-hospital cardiac arrest caused by known or presumed hypoxaemia. Because such patients typically receive IV resuscitative treatments within 3–5 min[58], $IVO_2$ may restore early spontaneous circulation and significantly improve outcomes in such patients. The precise dose and timing of $IVO_2$ treatment in each of these settings merit further investigation and optimization.

The major limitation of this technology is the volume of the fluid which must be administered to deliver the oxygen payload. In this work, the gas fraction of the foam ranged between 35 and 50%, which can probably be optimized further as it has been with similar products[15,59]. However, given the high rate of resting oxygen consumption (for example, a healthy adult consumes ~250 ml min⁻¹), oxygen provision using this technology is unlikely to provide more than several hundred millilitres of oxygen, particularly in critically ill patients with heart disease in whom even small quantities of administered volume may aggravate venous hypertension and pulmonary oedema. Thus, $IVO_2$ should be considered as a strategy for IV bolus administration of oxygen in critical situations, to reverse a critical physiology (such as a cardiac arrest or a pulmonary hypertensive crisis) or to bridge such patients to definitive treatment (such as extracorporeal membrane oxygenation support). The oxygen deficit that accrues over even a few critical minutes can be the difference between a favourable outcome and manifesting of neurologic injury for a lifetime, or be the difference between survival and death[5]. From an experimental perspective, another important limitation to note is that some members of the research team in the cardiac arrest experiment could not be blinded to treatment group due to the acute and rapid ostensible differences in physiologic response of the patient to the treatment. An additional practical challenge for blinding is that the PMB solution has a distinct white/opaque colour that is easily identifiable. Although we could have taped all the syringes and lines, it was challenging to completely mask the drug, particularly

when priming the injection ports or changing the syringes during treatment, since the drippings from the syringes or residues at the injection port were immediately visible. However, this limitation was balanced by actively measuring the quality of CPR (which was not different between groups) and by strictly following a treatment protocol throughout all phases of the experiment (including during the resuscitation and survival periods). Finally, the number of animals in the safety study was somewhat low, such that we were underpowered to detect minor differences between groups.

## Outlook

In future work, the molecular optimization of PMBs may further expedite the metabolism of shell constituents following injection. This may diminish macrophage vacuolization in the spleen of some animals in the safety study, probably due to polymer precipitates in an acidic lysosome (although this was not seen in any swine treated with the same polymer). An important feature of the current approach is that the molecular structures of the polymeric constituents can be tuned (for example, by decreasing MW, enhancing hydrophilicity or introducing enzymatic degradability) to further improve drug safety and minimize potential long-term adverse effects. In addition, PMBs could deliver various medical gases through intravenous and other routes (for example, enteral)[60,61], such as oxygen delivery for cancer therapy[62], neuroprotectant delivery (for example, hydrogen)[63,64], or the delivery of ultrasound and MRI contrast gases (for example, hyperpolarized xenon)[65].

## Methods

### Polymer synthesis

Dextran (MW 6 kD, 20 g) and 4-dimethylaminopyridine (64 g) were added to a round-bottom flask under nitrogen and then dissolved in 200 ml of anhydrous dimethylsulfoxide via oil bath at 55 °C. Separately, 5.6 g of succinic anhydride and 20 ml of acetic anhydride were dissolved in 60 ml of anhydrous dimethylsulfoxide and transferred to an additional funnel connected to the reaction flask. The anhydride solution was dropwise added to the reaction mixture over a period of 40 min under rigorous mixing using magnetic stirring. The reaction was maintained at 55 °C under nitrogen for 12 h before workup. The reaction mixture was slowly precipitated in a 4 l beaker that contains 3 l 6% acetic acid aqueous solution under rigorous stirring. The resultant precipitate was collected by centrifuging and subsequently washed with ionized water three times. The final product was collected by freeze drying as white powders.

### Manufacturing of oxygen-filled oPMBs

All fabrication procedures were conducted inside a biosafety workstation ISO class 5 equipped with vertical laminar flow (AirClean System). All equipment was UV sterilized and solutions were presterilized by autoclaving. The stock solution for homogenization was prepared as 11 mg ml⁻¹ in 30 wt% dextrose solution via addition of 0.6 ml 1 N sodium hydroxide per gram of polymer. Afterwards, the LmD solution was placed in an ice bath and a high-power UV lamp was submerged to sterilize the solution for 2 h under magnetic stirring before homogenization. LmD stock solution (90 ml) was transferred into a 1 l beaker via a graduated cylinder, and the homogenization probe (L5M-A Laboratory Mixer, Silverson) was submerged into the solution to stay at the a/w interface. The beaker was placed in a water bath at 33 °C and the solution was allowed to equilibrate to the same temperature for 5 min. Afterwards, the LmD solution was homogenized at 5,500 r.p.m.; upon homogenization, the solution immediately turned into a viscous white foam. The beaker position was occasionally adjusted manually to maintain maximum and even mixing. The polymer solution was first mixed for 2 min, followed by addition of 0.24 ml of dilute hydrochloric acid (HCl) (0.6%) by pipetting. The homogenization process continued with the addition of the same amount of HCl every minute

for another 8 min (a total of 9 additions of acid). The homogenization continued for another minute and terminated at the 11th minute. Upon the last addition of acid (~10 min), a notable decrease in foam viscosity was observed, suggesting that the solution pH had transitioned below p$K_a$ to protonate majority of the carboxylic acid groups, and the polymer crosslinking led to a phase transition. The resulting foam was left undisturbed in the water bath for another 10 min, then 150 ml of 10% dextrose solution was poured into the foam to help transfer the foam mixture from the beaker to a 500 ml conical-shaped flask. Multiple batches of LmD foams were combined and collected into conical-shaped flasks. They were allowed to sit overnight and LmD PMBs floated to the top to form a cake-like cream layer, while polymeric debris accumulated at the bottom of the flask. Then the bottom fluid of foam as well as polymer debris were siphoned using a roller pump and a long stainless-steel needle, and then fresh sterile D10 solution was added to the thick foam layer to redisperse them in solution via gentle shaking of the flask. This process was repeated 3 times, and foam layers from various batches were further combined and concentrated to the desired final concentrations. To oxygenate the PMBs, the PMB foams were placed in a flask with a silicon septum, and the headspace was then purged by flowing humidified oxygen via a 0.25 μm sterile filter for 12 h with occasional shaking of the flask. A small aliquot of the solution was drawn to a syringe via a sterile needle to measure the pO$_2$ of the solution. The fully oxygenated PMBs were then transferred and constituted at desired concentrations in 60 ml syringes before intravenous administration for animal experiments. Sterility tests were conducted throughout the entire fabrication process. Both before and after the PMB fabrication, the sterility of each LmD solution, each fabrication batch as well as combined foams were tracked and tested by plating onto blood agar plates which were continuously monitored for potential bacterial growth. Only lots that exhibited no colony growth after incubation for 7 days were used for animal experiments.

All animal experiments conducted in this work were approved and conducted according to Boston Children's Hospital Institutional Animal Care and Use Committee policy.

### Rodent haemodynamic safety studies
Male Sprague-Dawley rats (weight 503 ± 52 g, Charles River Laboratories) were anaesthetized with intraperitoneal injection of ketamine (45–75 mg kg$^{-1}$) and xylazine (5–10 mg kg$^{-1}$), followed by orotracheal intubation. Then, anaesthesia was maintained by inhalational isoflurane (1–2%). Animals were mechanically ventilated (SAR-1000, CWE) with a tidal volume 5–8 ml kg$^{-1}$ and respiratory rate 40–45 breaths per min on 30% oxygen. Instrumentation included a 24G angiocatheter in the tail vein for microparticle infusion, two femoral artery cannulae placed by cutdown (one with a 24G angiocatheter used for blood collection and the other with a pressure catheter for haemodynamic monitoring) (Millar Mikro-Tip Pressure Catheter Transducer, model SPR-671, 1.4F), and a femoral venous catheter (3 French) placed by cutdown and advanced to the right atrium to monitor central venous pressure. Subsequently, a median sternotomy was performed and pressure–volume catheters (Millar Mikro-Tip Pressure-Volume Catheter Transducer, 9 mm spacing, model SPR-847, 1.4F) were placed into the right and the left ventricles for pulmonary artery pressure and cardiac output monitoring, respectively. Temperature was controlled by a rectal temperature probe connected to a heating pad for a central temperature target of 37 °C. All these instruments were calibrated and continuously recorded using PowerLab/LabChart software (LabChart Pro 8 software, ADInstruments). After a 15-min baseline period, animals were injected with either intravenous oxygen formulation (IVO$_2$) (80% foam) or control solution (10% dextrose). Each injection was a total of 5 ml volume over 1 min, followed by a 1 ml Plasma-Lyte flush and then a 2-min stabilization period. Each injection was repeated a total of 5 times. Afterwards, there was a 60-min observation period with continuous haemodynamic monitoring. Animals were euthanized at the end of the study by cardiac explantation. MABP, left-ventricular end-diastolic pressure (LVEDP), cardiac index (CI = stroke volume × heart rate kg$^{-1}$) and pulmonary vascular resistance (PVR = (mean pulmonary arterial pressure − LVEDP)/CI) were exported as 1-min averages. Baseline was calculated as the average of the 15-min baseline in all animals included in the study, then data were presented as percentage change from baseline. Comparison between groups was performed by analysing averages over the 3 periods using analysis of variance (ANOVA).

### Rodent echocardiography safety study
Male Sprague-Dawley rats (weight 400–500 g, Charles River Laboratories) were instrumented similarly to our previous study, including orotracheal intubation, mechanical ventilation (FiO2 0.3), femoral vein catheterization and femoral artery haemodynamic continuous monitoring. Anaesthesia was maintained with inhaled isoflurane (1–2%). Transthoracic echocardiography (Philips EPIQ ultrasound machines, Philips Healthcare) was performed by a certified paediatric cardiologist according to previous studies[66] in the left parasternal window for a four-chamber view. Our intravenous formulation filled with either oxygen (IVO$_2$) or air gas (IVAir) and a previous generation of lipid oxygen microparticles (LOM) were compared to a control echocardiography where no infusion was performed. Injections at three different rates (4 ml kg$^{-1}$ min$^{-1}$, 8 ml kg$^{-1}$ min$^{-1}$ and 12 ml kg$^{-1}$ min$^{-1}$) were analysed using ImageJ for intensity quantification. A selection of 10 frames per study was randomly selected and compared to control images. Left and right ventricle cavity areas were selected in the analyser and the intensity quantified relative to the ventricle surface area. Results were compared using ANOVA.

### Efficacy study in an asphyxia model in swine
Female Yorkshire swine (10.1–12.5 kg, Parson's Farms) were housed individually with a 12-h dark/light cycle with free access to food and water. At least a week of acclimatization was maintained before the experiment.

**Experimental protocol.** Animals were anaesthetized by intramuscular injection of tiletamine and xylazine and orally intubated via direct laryngoscopy. Following endotracheal intubation, animals were connected to a mechanical ventilator and sedated with 1–3% isoflurane, titrated to effect. Ventilation was managed on volume control, with tidal volumes of 8–10 ml kg$^{-1}$, PEEP 5, rate of 12–15 bpm and 21% FiO$_2$ (Draeger Apollo). Minute ventilation was titrated to achieve an end tidal carbon dioxide concentration of 40 mmHg. Core temperature was maintained at 38 °C with a heating blanket. Oxygen consumption was monitored during the baseline period (GE E-CAIOVx Respiratory Module). Instrumentation included a femoral arterial catheter for blood sampling (3 French, 5 cm) and a femoral venous catheter for treatment/control infusion (3 French, 5 cm). We also placed an oximetric catheter (PediaSat, 4.5 Fr, 5 cm, Edwards Lifesciences) in the contralateral femoral artery (for continuous haemodynamic monitoring) and in the right internal jugular vein. All catheters were placed by surgical cutdown using the Seldinger technique under direct visualization of the vessels. Catheters were transduced and calibrated according to manufacturer instructions before use. Swine were monitored continuously with telemetry, pulse oximetry, and cerebral and somatic near-infrared reflectance spectrometry (NIRS, Somanetics). CPR quality was monitored using real-time feedback (CPR Electrodes, M Series, Zoll Medical) for real-time feedback and defibrillation. Following instrumentation, anaesthesia was transitioned to intravenous fentanyl (20 mcg kg$^{-1}$ per dose) and cisatracurium (0.5 mg kg$^{-1}$ per dose), and isoflurane was discontinued for the remainder of the experiment. Inhalational anaesthesia was discontinued 30 min before asphyxia. Anaesthetic level was monitored by haemodynamic response and movement to painful stimuli, and bolus doses repeated as needed. Following a 30-min baseline period, baseline blood samples were obtained for complete

blood count (CBC), biochemistry, venous and arterial blood gases and serum for neuro biomarkers.

Following the baseline observation period, the endotracheal tube was clamp occluded and the ventilator disconnected (Time = 0). Arterial blood gases were measured every 2 min (starting on Time = 1) during the first 30 min. Cardiac arrest was defined as a systolic blood pressure (SBP) less than 40 mmHg for 5 s or longer. When in cardiac arrest, animals received metronome-guided, high-quality chest compressions and intravenous medications as outlined by the American Heart Association Advanced Cardiac Life Support (ACLS) algorithm, including epinephrine (0.01 mg kg$^{-1}$) every 2 min, lidocaine (1 mg kg$^{-1}$), atropine (0.01 mg kg$^{-1}$), amiodarone (5 mg kg$^{-1}$), calcium gluconate (50 mg kg$^{-1}$) and sodium bicarbonate (as needed, 1 mEq kg$^{-1}$). Animals were randomly allocated to an intervention, either intravenous oxygen or control. Intravenous oxygen consisted of 400 ml of 60% PMBs (~140 ml oxygen and 260 ml D10). Control consisted of 260 ml D10 solution at 13 ml kg$^{-1}$ per dose. At minutes 6, 8 and 10, the allocated intervention was administered via the femoral central venous catheter. Each of the three doses was administered over 2 min.

At time = 12 min, the tracheal tube clamp was removed, and ventilation was restored with 100% oxygen and the ventilator settings above (Maquet Servo-i). CPR was continued until Time = 30 min or until ROSC, defined as SBP > 50 mmHg.

**Survival period and critical care metrics.** Surviving animals were then maintained in an intensive care environment for 4 days. Animals were mechanically ventilated and assessed for extubation readiness every 12 h with a spontaneous breathing test, as well as neurological and haemodynamic status. Inotropic support with continuous infusions of dopamine (3–10 mcg kg$^{-1}$ min$^{-1}$), epinephrine (0.02–0.2 mcg kg$^{-1}$ min$^{-1}$), norepinephrine (0.02–0.5 mcg kg$^{-1}$ min$^{-1}$) and vasopressin (0.0005–0.002 U kg$^{-1}$ min$^{-1}$) was used as needed to maintain SBP > 70 mmHg. Sedation and analgesia were maintained while intubated, with continuous infusions of propofol (1–3 mg kg$^{-1}$ h$^{-1}$) and fentanyl (5–10 mcg kg$^{-1}$ h$^{-1}$) as needed for animal comfort. Complete neurologic examination using a previously validated tool, the Swine Neurological Deficit Score[67] was performed at baseline, $T$ = 6 h and daily thereafter. This score includes an evaluation of cranial nerves, respiration, motor and sensory function, level of consciousness and behaviour. In this score, 0 is considered normal (no deficits) and 500 is equivalent to brain death.

A determination of extubation readiness was made on the following conditions: no seizures in the past 2 h, not on inotropic support, respiratory rate <40 breaths per min and SpO$_2$ > 92% on pressure support only ventilation (PS 6, PEEP 5, FiO$_2$ < 50%) for 2 h. If the animal was deemed to be clinically ready to extubate, all lines were removed except for the internal jugular catheter, which was maintained for blood sampling and drug administration. A fentanyl patch (25 mcg) was placed for analgesia. Once extubated, the animal was transferred to a kennel for observation and monitoring with continuous pulse oximetry.

Seizures were treated as follows. Airway and haemodynamic support were provided as clinically indicated. If a seizure lasted for >2 min, a dose of IV lorazepam was administered (0.1 mg kg$^{-1}$) every 5 min up to 3 doses. Thereafter, persistent seizures were treated with IV phenytoin (20 mg PE kg$^{-1}$), repeated once after 15 min if seizures persisted. Thereafter, persistent seizures were treated with IV phenobarbital (20 mg kg$^{-1}$), repeated once after 15 min if seizures persisted. Swine that were persistently seizing thereafter were treated with an infusion of midazolam, with a starting dose of 0.1 mg kg$^{-1}$ h$^{-1}$, with an hourly uptitration of 0.1 mg kg$^{-1}$ h$^{-1}$ to a maximum dose of 0.7 mg kg$^{-1}$ h$^{-1}$. Inotropic support was provided as required for hypotension.

Diabetes insipidus was treated as follows. If urine output exceeded 3 ml kg$^{-1}$ h$^{-1}$, serum sodium was checked every 2 h. When sodium exceeded 150 mmol l$^{-1}$ and urine output (UOP) exceeded 5 ml kg$^{-1}$ h$^{-1}$ for 2 h, vasopressin infusion was initiated at 1 mU kg$^{-1}$ h$^{-1}$ and uptitrated hourly to 10 mU kg$^{-1}$ h$^{-1}$ until UOP decreased to <5 ml kg$^{-1}$ h$^{-1}$. During the survival period, the following monitoring labs were performed at 30 min, at 1, 2, 3, 4, 6, 12 and 24 h, and then twice daily during the survival period: blood gas, CBC, biochemistry and GFAP in serum. GFAP was assessed using an electrochemiluminescent immunoassay (Meso Scale Diagnostics) with a detection range of 0.001–40.0 ng ml$^{-1}$.

**Brain MRI.** On day 4, animals were reintubated if they had been previously extubated, and anaesthetized for brain MRI (3 Tesla Skyra model scanner, 64-channel head and neck coil, Siemens). High-resolution images including weighted T1, T2, magnetic resonance angiography (MRA), diffusion weighted imaging (DWI), susceptibility weighted imaging (SWI), magnetic resonance spectroscopy (MSR) and fluid attenuated inversion recovery (FLAIR) were obtained. These images were analysed by a board-certified neuroradiologist blinded to treatment allocation. Then, areas of enhancement on axial and coronal T2 and diffusion coefficient images were manually processed on a voxel-per-voxel basis and outlined (itk-SNP software application, Penn Image Computing and Science Laboratory, University of Pennsylvania, and Scientific Computing and Imaging Institute, University of Utah) by the Department of Radiology at Boston Children's Hospital; all researchers were also blinded to treatment allocation. From these values, total volumes of cranial injury were calculated using software normalized to brain volume. The total volume of injury was compared between groups using $t$-test.

**Neurohistology analysis.** Following brain MRI, both internal jugular veins and carotids were cannulated by cutdown for brain perfusion. The brain was perfused by administrating 2 l of normal saline via both carotid arteries followed by 4 l of paraformaldehyde for fixation. During this procedure, the animal was sedated with inhaled isoflurane until death was confirmed by the absence of vital signs. A complete autopsy was performed including all vital organs and brain. All organs were formalin fixed before pathological analysis. The skull of swine was opened and the head was fixed in 4% paraformaldehyde for 72 h, following which brain tissue was extracted and placed in formalin for analysis. Brains were processed, sliced, embedded in paraffin and stained for haematoxylin and eosin, and slides were examined by fluorescence with a Rhodamine filter. Hypoxic-ischaemic injuries were studied in 7 different regions: frontal, parietal and temporal cortex, dentate gyrus and pyramidal layer of hippocampus, caudate and thalamus, by a board-certified pathologist who was blinded to treatment allocation. The lesions were graded according to a previously defined scale as follows: 0 = no injury, 1 = rare hypereosinophilic (HE) neurons, 2 = clusters of HE neurons, 3 = over 50% of HE neurons, 4 = over 90% HE neurons, 5 = cavitated infarction. This analysis only included animals that survived during the 4-day observation period. Heart, lungs, kidneys and liver were subjectively reviewed by a pathologist blinded to treatment group.

**PET/CT imaging and biodistribution study**

LmD polymer (400 mg) was dissolved in 10 ml water by adjusting pH to 6 in a round-bottom flask, to which TCO-PEG3-amine (8 mg, 0.021 mmol, Click Chemistry Tool), 1-ethyl-3-(3-dimethylaminopropyl) carbodiimide (5.76 mg, 0.03 mmol) and $N$-hydroxysuccinimide (2.3 mg, 0.02 mmol) were added to react at room temperature. After 12 h, the product was precipitated by adding dilute hydrochloride acid (0.1 M), centrifuged and repeatedly washed with water. The final product LmD-TCO was collected as solid powder. Separately, tetrazine-NHS ester (0.12 mmol, 38 mg) and DFO-NH2/mesylate (69 mg, 95%, 0.1 mmol) were reacted in 1 ml of dimethylformamide in the presence of 30 μl of triethylamine for 12 h at room temperature, after which the product tetrazine-DFO was precipitated with excess of acetone and obtained as a pink powder after drying. Chemicals and reagents were obtained from Sigma-Aldrich unless otherwise specified. To minimize

metal contamination, glassware was washed with 2 M nitric acid (Fisher Scientific, certified ACS plus grade) and rinsed with ultrapure water (>15 MΩ resistivity) (US Filter/Siemens Water Technologies) before use. All solutions were prepared using ultrapure water. Metal contaminants were removed from buffer solutions using a Chelex-100 resin column (Bio-Rad). Metal-naive pipette tips were purchased from Rainin Instrument and were used to prepare all samples. Plasticware was washed with ultrapure water and then equilibrated with buffers where appropriate. Zirconium-89 was purchased from Washington University in St Louis (St Louis, Missouri) and was supplied in 1 M oxalic acid. In a representative labelling reaction, a 40 µl aliquot of the stock $^{89}$Zr solution (in 1 M oxalic acid) was neutralized with 18 µl 2 M $Na_2CO_3$, 100 µl PBS was added and the pH was adjusted to 7 with glacial acetic acid. Then 0.1 ml 1 mg ml$^{-1}$ tetrazine-DFO in dimethylsulfoxide was added into the tube and the solution was incubated at room temperature for 1 h. Afterwards, 1 ml of LmD-TCO (20 mg ml$^{-1}$) in PBS solution (pH adjusted to 7.2) was added and allowed to react with the $^{89}$Zr-complexed tetrazine-DFO for 5 min. The reaction was monitored by instant thin layer chromatography; $^{89}$Zr-LmD complexes remained at the origin and unbound $^{89}$Zr migrated with the solvent front. To remove unbound $^{89}$Zr, the labelled LmD polymers were precipitated by adding 0.1 ml of acetic acid and the labelled product was collected by centrifugation. The labelled polymer was further purified by repeatedly washing with water and centrifuging. The final polymers were then redissolved in PBS buffer by adding 1 N sodium hydroxide to adjust the pH to 7.2 at a concentration of 5–10 mg ml$^{-1}$.

Small-animal PET/CT image data were obtained using a Bruker Albira multimodality (PET/SPECT/CT) small-animal imaging system (Bruker). Phosphor storage screens were scanned using a Fujifilm BAS-5000 (FUJIFILM Life Science). Radioactivity in the tissue samples was assayed with a Packard Cobra II automated gamma counter. Once the polymer was radiolabelled, the injection solution was prepared by mixing 5–10 mg labelled polymer (radioactivity of 120–140 µCi) with unlabelled (cold) polymers in a total of 6 - 7 ml of PBS solution. Male Sprague-Dawley rats (125–150 g) were anaesthetized with 2% isoflurane via nose cone and the prepared polymer solution administered via tail-vein injection (400 mg kg$^{-1}$). Afterwards, PET/CT imaging was performed immediately after injection, at 3, 24 and 48 h, and then every 48 h until day 7. On day 7 after injection, animals were euthanized by cardiac explantation under anaesthesia (3% isoflurane). Tissue samples were immediately collected and weighed, and then assayed using gamma counter to determine $^{89}$Zr activity. The percent of injected dose per tissue gram was calculated by comparison to samples of known $^{89}$Zr concentration.

### Rodent 14-day safety study

Male Sprague-Dawley rats (400–500 g) were anaesthetized with 1 to 2% isoflurane, inhaled via nose cone. A tail-vein catheter was placed, flushed with Plasma-Lyte, and animals were then randomized to either receive intravenous oxygen solution (treatment, 70% foam) or 10% dextrose solution (control) at a single dose of 32 ml kg$^{-1}$ over 10 min followed by a 1 ml Plasma-Lyte flush, or a double dose (total of 64 ml kg$^{-1}$, 30 min between doses, $n$ = 3–6). After recovering from anaesthesia, animals were placed in metabolic cages for 6 h for urine and stool output recording. In metabolic cages, animals had access to food but not to water. This process was repeated once daily. Animals' behaviour was monitored daily, including activity, response to pain, response to light and sound, respiration pattern and comfort. At 3, 7 or 14 days, animals were randomly anaesthetized with isoflurane, a catheter was placed in the femoral artery for blood collection (including blood gas analysed using ABL 90 Flex Plus; Radiometer America), CBC (analysed using VetScan; HM5 Hematology Analyzer, Zoetis), basic chemistry panel (analysed using VetScan; VS2 Chemistry Analyzer, Zoetis) and coagulation (analysed using rotational thromboelastometry; ROTEM Delta 3678), and then euthanized by heart explantation, collecting vital organs. Heart, lungs, kidneys and liver were placed in formalin and then sent to the Department of Research Histopathology at Harvard Medical School for blind analysis. Organs were processed, sliced, embedded in paraffin and stained for haematoxylin and eosin.

### Statistical analysis

The primary outcome of this study was the SNDS at 3 days in surviving swine. On the basis of our preliminary data, we expected an SNDS score of 420 with s.d. of 100 in surviving swine in the control group. The inclusion of 3 surviving animals per group allowed us to detect a difference of 250 in the SNDS at 3 days with 90% power and alpha of 0.05, which would be very clinically significant. Continuous variables were described as mean and s.d., and categorical were described as numbers and percentages. Differences between groups in arterial saturation, arterial carbon dioxide tension, haemodynamics, histopathology scores, chemistry and haematologic parameters were evaluated over time by using a two-way repeated measures ANOVA with an interaction term between treatment group and time; time-dependent differences between groups were evaluated using a post hoc analysis (Sidak's multiple comparisons method). Any missing values were imputed as median values across the group for that timepoint. Single timepoint values, such as baseline characteristics, resuscitation outcomes, GFAP changes and differences in cerebral infarct volumes were compared between groups using Student's $t$-test, Mann–Whitney $U$ test or exact Fisher test, as appropriate. Statistical analysis and graphing were performed using Prism v.9.00 software (GraphPad). For all tests, a two-sided $P < 0.05$ was considered statistically significant.

Additional information on materials and characterizations, and a table that summarizes the dose of IVO$_2$ in each animal experiment (Supplementary Table 1) are included in Supplementary Information.

### Reporting summary

Further information on research design is available in the Nature Portfolio Reporting Summary linked to this article.

## Data availability

The main data supporting the results in this study are available within the paper and its Supplementary Information. Additional microscopic and ultrasound images, experimental notes and raw (animal) patient data generated during the study are available from the corresponding authors on reasonable request. Source data are provided with this paper.

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

## Acknowledgements

We thank S. Mackay for assistance with large animal experiments. Cryo-SEM imaging was performed at the Center for Nanoscale Systems with assistance from A. Graham, supported by the NSF's National Nanotechnology Infrastructure Network. We thank B. Tang for assistance with UV-Vis measurement and R. Berg for suggestions for the manuscript. J.N.K., B.D.P. and Y.P. disclose support for the research described in this study from the National Institutes of Health (grant 5R01HL141818) and the US Department of Defense (grant W81XWH-19-1-0237). This work was supported by the Office of the Assistant Secretary of Defense for Health Affairs through the Peer Reviewed Medical Research Program. Opinions, interpretations, conclusions and recommendations are those of the author and are not necessarily endorsed by the Department of Defense. Y.P. also thanks the support by an OSP pilot award from Boston Children's Hospital.

## Author contributions

J.N.K., B.D.P. and Y.P. conceptualized the project. J.N.K. and Y.P. developed the methodology. All authors conducted investigations. J.N.K. and B.D.P. acquired funding. J.N.K. and Y.P. administered and supervised the project. J.G.M., J.N.K. and Y.P. wrote the original draft. All authors reviewed and edited the paper.

## Competing interests

B.D.P., J.N.K. and Y.P. are inventors on a patent (US11147890B2, United States, 2018) related to the technology described in this article. The other authors declare no competing interests.

## Additional information

**Extended data** is available for this paper at https://doi.org/10.1038/s41551-024-01266-8.

**Correspondence and requests for materials** should be addressed to John N. Kheir or Yifeng Peng.

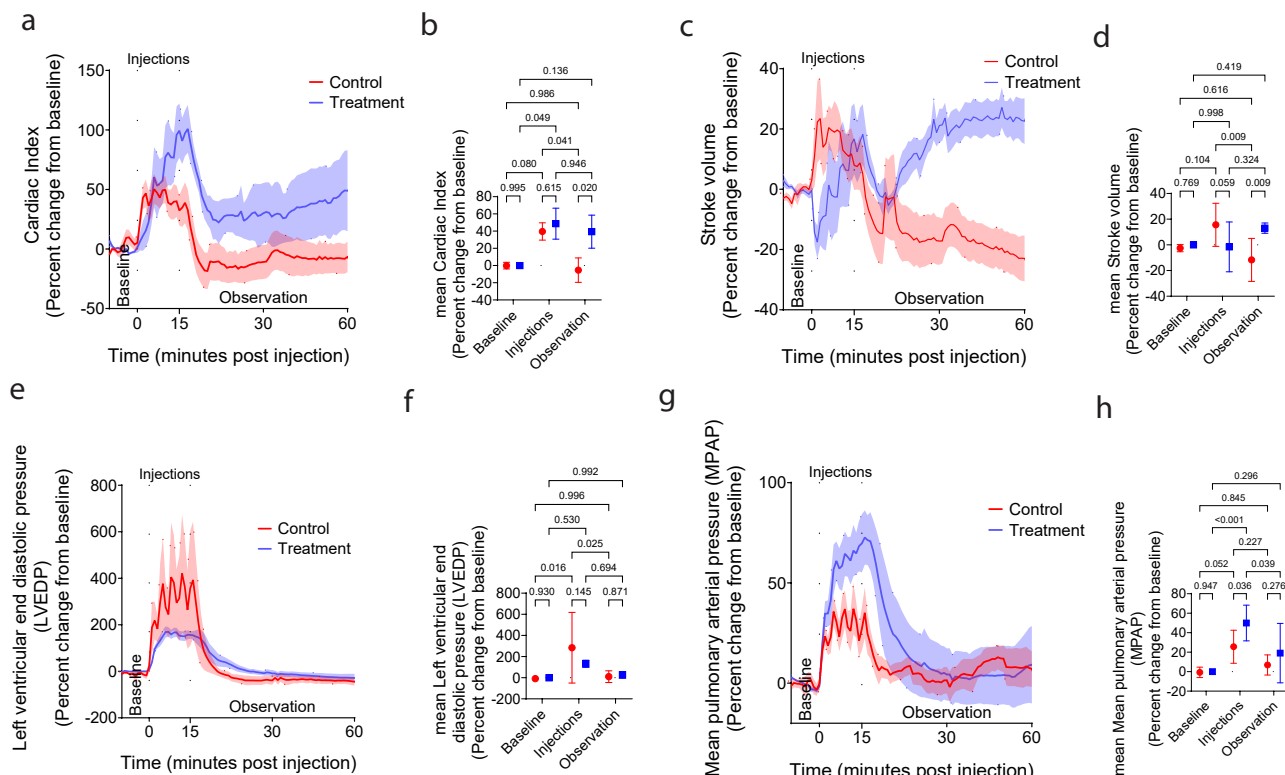

**Extended Data Fig. 1 | Haemodynamic response in acute rodent safety study.** Changes in cardiac index (CI) (**a**) and the mean CI values during each experimental period (**b**), changes in stroke volume (SV) (**c**) and the mean SV values during each experimental period (**d**), changes in left-ventricular end-diastolic pressure (LVEDP) (**e**) and the mean LVEDP values during each experimental period (**f**), as well as changes in mean pulmonary arterial pressure (MPAP) (**g**) and the mean MPAP values (**h**) during each experimental period were presented as mean and SEM, and analyzed by two-way ANOVA.

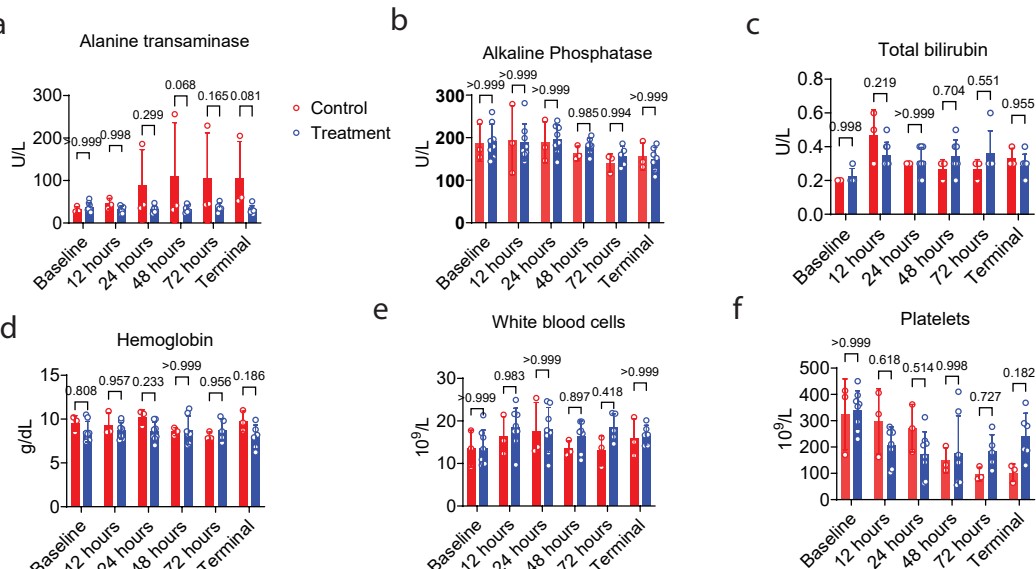

**Extended Data Fig. 2 | Changes in select clinical markers in the asphyxia efficacy study.** The values for various clinical biomarkers including alanine transaminase (**a**), alkaline phosphatase (**b**), total bilirubin (**c**), hemoglobin (**d**), white blood cells (**e**) and platelets (**f**) were among normal ranges for IVO2-treated animals and were not statistically different from the surviving animals from the control group. Data presented as mean ± SD, statistical analysis by two-way ANOVA.

# Reporting Summary

## Statistics

For all statistical analyses, confirm that the following items are present in the figure legend, table legend, main text, or Methods section.

| n/a | Confirmed | |
|---|---|---|
| ☐ | ☒ | The exact sample size ($n$) for each experimental group/condition, given as a discrete number and unit of measurement |
| ☐ | ☒ | A statement on whether measurements were taken from distinct samples or whether the same sample was measured repeatedly |
| ☐ | ☒ | The statistical test(s) used AND whether they are one- or two-sided<br>*Only common tests should be described solely by name; describe more complex techniques in the Methods section.* |
| ☐ | ☒ | A description of all covariates tested |
| ☐ | ☒ | A description of any assumptions or corrections, such as tests of normality and adjustment for multiple comparisons |
| ☐ | ☒ | A full description of the statistical parameters including central tendency (e.g. means) or other basic estimates (e.g. regression coefficient) AND variation (e.g. standard deviation) or associated estimates of uncertainty (e.g. confidence intervals) |
| ☐ | ☒ | For null hypothesis testing, the test statistic (e.g. $F$, $t$, $r$) with confidence intervals, effect sizes, degrees of freedom and $P$ value noted<br>*Give P values as exact values whenever suitable.* |
| ☒ | ☐ | For Bayesian analysis, information on the choice of priors and Markov chain Monte Carlo settings |
| ☒ | ☐ | For hierarchical and complex designs, identification of the appropriate level for tests and full reporting of outcomes |
| ☒ | ☐ | Estimates of effect sizes (e.g. Cohen's $d$, Pearson's $r$), indicating how they were calculated |

*Our web collection on statistics for biologists contains articles on many of the points above.*

## Software and code

Policy information about availability of computer code

| Data collection | Bruker Topspin was used to collected NMR data. No software was used in other data collections. |
|---|---|
| Data analysis | All data was analysed using GraphPad Prism version 10.1.2 for Windows GraphPad Software. MRI results were analysed by an open-source ITK-SNP 4.0 software application (http://www.itksnap.org/pmwiki/pmwiki.php; Penn Image Computing and Science Laboratory, University of Pennsylvania, Pennsylvania, and Scientific Computing and Imaging Institute, University of Utah, Utah). |

For manuscripts utilizing custom algorithms or software that are central to the research but not yet described in published literature, software must be made available to editors and reviewers. We strongly encourage code deposition in a community repository (e.g. GitHub). See the Nature Portfolio guidelines for submitting code & software for further information.

## Data

Policy information about availability of data

All manuscripts must include a data availability statement. This statement should provide the following information, where applicable:
- Accession codes, unique identifiers, or web links for publicly available datasets
- A description of any restrictions on data availability
- For clinical datasets or third party data, please ensure that the statement adheres to our policy

The main data supporting the results in this study are available within the paper and its Supplementary Information. Source data are provided with this paper.

Additional microscopic and ultrasound images, experimental notes, and raw (animal) patient data generated during the study are available from the corresponding authors on reasonable request.

## Research involving human participants, their data, or biological material

Policy information about studies with [human participants or human data](). See also policy information about [sex, gender (identity/presentation), and sexual orientation]() and [race, ethnicity and racism]().

| | |
|---|---|
| Reporting on sex and gender | The study did not involve human research participants. |
| Reporting on race, ethnicity, or other socially relevant groupings | – |
| Population characteristics | – |
| Recruitment | – |
| Ethics oversight | – |

Note that full information on the approval of the study protocol must also be provided in the manuscript.

## Field-specific reporting

Please select the one below that is the best fit for your research. If you are not sure, read the appropriate sections before making your selection.

☒ Life sciences ☐ Behavioural & social sciences ☐ Ecological, evolutionary & environmental sciences

For a reference copy of the document with all sections, see [nature.com/documents/nr-reporting-summary-flat.pdf](http://nature.com/documents/nr-reporting-summary-flat.pdf)

## Life sciences study design

All studies must disclose on these points even when the disclosure is negative.

| | |
|---|---|
| Sample size | On the basis of preliminary data, we expected a Swine Neurologic Deficit Score of 420 with s.d. of 100 in surviving swine in the control group. The inclusion of 3 surviving animals per group allowed us to detect a difference of 250 in the SNDS at 3 days with 90% power and an alpha of 0.05. |
| Data exclusions | No data were excluded. |
| Replication | All instruments were calibrated prior to use. |
| Randomization | Swine were randomized between groups in alternating fashion. |
| Blinding | The investigators performing experiments were not blinded to the treatment group because the difference in haemodynamics was marked and obvious to all in the room. The investigators reading the MRI and histology endpoints were blinded to treatment allocation. |

## Reporting for specific materials, systems and methods

We require information from authors about some types of materials, experimental systems and methods used in many studies. Here, indicate whether each material, system or method listed is relevant to your study. If you are not sure if a list item applies to your research, read the appropriate section before selecting a response.

### Materials & experimental systems

| n/a | Involved in the study |
|---|---|
| ☒ | ☐ Antibodies |
| ☒ | ☐ Eukaryotic cell lines |
| ☒ | ☐ Palaeontology and archaeology |
| ☐ | ☒ Animals and other organisms |
| ☒ | ☐ Clinical data |
| ☒ | ☐ Dual use research of concern |
| ☒ | ☐ Plants |

### Methods

| n/a | Involved in the study |
|---|---|
| ☒ | ☐ ChIP-seq |
| ☒ | ☐ Flow cytometry |
| ☐ | ☒ MRI-based neuroimaging |

# Animals and other research organisms

Policy information about studies involving animals; ARRIVE guidelines recommended for reporting animal research, and Sex and Gender in Research

| | |
|---|---|
| Laboratory animals | Female Yorkshire swine, <1 month; Male Sprage Dawley rats, ~500 grams. |
| Wild animals | The study did not involve wild animals. |
| Reporting on sex | Single sex was used for each study, as describe above. |
| Field-collected samples | The study did not involve samples collected from the field. |
| Ethics oversight | Boston Children's Hospital Institutional Animal Care and Use Committee. |

Note that full information on the approval of the study protocol must also be provided in the manuscript.

# Magnetic resonance imaging

## Experimental design

| | |
|---|---|
| Design type | MRI performed at a single time point at the end of the swine experiments. |
| Design specifications | One time imaging for each subject at 84 hours post-injury. |
| Behavioral performance measures | The swine neurologic deficit score was compared between groups using 2-way ANOVA. |

## Acquisition

| | |
|---|---|
| Imaging type(s) | Structural MRI, DWI, MRA. |
| Field strength | 3T |
| Sequence & imaging parameters | T1-MPRAGE: TE 2.5ms, TI 900ms, TR 1600ms, Matrix size 184x256, FoV 140mmx196mm, FA 9°, THK 0.9mm. T2-weighted TSE:TE 100ms, TR 10,400ms, Matrix size 291x512, FoV 146mmx180mm, FA 150°, THK 2mm. T2-FlAIR: TE 108ms, TR 9000ms, TI 2500ms, ETL 16, Matrix size 250x320, FoV 140mmx180mm, FA 150°,THK 2mm. Susceptibility weighted imaging: TE 20ms, TR 27ms,ETL 1, Matrix size 224x256, FoV 157mmx180mm, FA 15°, THK 1.25mm. Multistab 3D TOF MRA imaging: TE 3.69ms, TR 21ms, ETL 1, Matrix size 331x384, FoV 181mmx199mm, FA 18°, THK 0.6mm. RESOLVE-DWI: TE 75ms, TR 10490ms, ETL 0, matrix size 120x160, FoV 156mmx 220mm, FA 180°, THK 2mm. CUSP90-SMS: TE 98ms, TR 4200ms, ETL 59, Matrix size 160x160, FoV 220mmx220mm, FA 90°, THK 2mm. SVS: TE 100ms, TR 9360ms, ETL 18, Matrix size 291x512, FoV 146mmx180mm, FA 150°, THK 2mm. |
| Area of acquisition | Whole brain |
| Diffusion MRI | ☒ Used     ☐ Not used |
| Parameters | RESOLVE-DWI: TE 75ms, TR 10490ms, ETL 0, matrix size 120x160, FoV 156mmx 220mm, FA 180°, THK 2mm. CUSP90-SMS: TE 98ms, TR 4200ms, ETL 59, Matrix size 160x160, FoV 220mmx220mm, FA 90°, THK 2mm. |

## Preprocessing

| | |
|---|---|
| Preprocessing software | Scanner was a Skyra in the Boston Children's Clinical Building. Clinical protocols were used and data reviewed on Synapse 5 Viewer. |
| Normalization | These pigs were scanned on a clinical scanner using standard clinical protocols. No normalization was performed. |
| Normalization template | Not applicable |
| Noise and artifact removal | Scans were visually monitored for motions. |
| Volume censoring | Not applicable. |

## Statistical modeling & inference

| | |
|---|---|
| Model type and settings | Areas of enhancement on axial and coronal T2 and diffusion coefficient images were manually processed on a voxel-per-voxel basis and outlined (itk-SNP software application, Penn Image Computing and Science Laboratory, University of Pennsylvania, Pennsylvania, and Scientific Computing and Imaging Institute, University of Utah, Utah) by the Department of Radiology at Boston Children's Hospital, all of whom were also blinded to treatment allocation. From these values, total volumes of cranial injury were calculated using software normalized to brain volume. |
| Effect(s) tested | The total volume of injury was compared between groups by t-test. |
| Specify type of analysis: | ☒ Whole brain ☐ ROI-based ☐ Both |
| Statistic type for inference<br>(See Eklund et al. 2016) | Injury determined based on voxel-per-voxel analysis. |
| Correction | NA |

## Models & analysis

| n/a | Involved in the study |
|---|---|
| ☒ | ☐ Functional and/or effective connectivity |
| ☒ | ☐ Graph analysis |
| ☒ | ☐ Multivariate modeling or predictive analysis |

