## [Peer Review File · Nature Biomedical Engineering]

Systemically injected oxygen within rapidly dissolving microbubbles improves the outcomes of swine with severe hypoxaemia

Corresponding author: Yifeng Peng

Editorial note

This document includes relevant written communications between the manuscript's corresponding author and the editor and reviewers of the manuscript during peer review. It includes decision letters relaying any editorial points and peer-review reports, and the authors' replies to these (under 'Rebuttal' headings). The editorial decisions are signed by the manuscript's handling editor, yet the editorial team and ultimately the journal's Chief Editor share responsibility for all decisions.

Any relevant documents attached to the decision letters are referred to as **Appendix #**, and can be found appended to this document. Any information deemed confidential has been redacted or removed. Earlier versions of the manuscript are not published, yet the originally submitted version may be available as a preprint. Because of editorial edits and changes during peer review, the published title of the paper and the title mentioned in below correspondence may differ.

Correspondence

Wed 20 Dec 2023

Decision on Article nBME-23-2760

Dear Prof Peng,

Thank you again for submitting to *Nature Biomedical Engineering* your manuscript, "Injectable oxygen improves outcomes in severe hypoxemia". The manuscript has been seen by 3 experts, whose reports you will find at the end of this message.

You will see that the reviewers appreciate the work. You will see that the reviewers appreciate the work, and that they raise a number of technical criticisms that we hope you will be able to address. In particular, we would expect that a revised version of the manuscript provides:

- * Additional characterization data (including PaO₂, PV0₂ and pH), as suggested by Reviewers #1 and #2.
- * Further evidence of the rapid dissolution of the microbubbles, and of their longer-term stability.
- * Clarification of the oxygen-dosage calculations for the pigs, as queried by Reviewers #1 and #2, and comparison of the gas-infusion rates, as per the comments by Reviewer #3.
- * Clarification of the rationale for the choice of the animal model, as suggested by Reviewers #2 and #3.
- * Discussion of the translatability of the model to realistic emergency situations, as suggested by all reviewers.
- * Discussion of the limitations of the approach, as per the relevant comments of all Reviewers.

Moreover, because of sensitivities around animal experimentation in society, and given the limited scientificvalue of Supplementary Videos 6 and 7 depicting the pig, we strongly recommend that these videos be removed.

When you are ready to resubmit your manuscript, please upload the revised files, a point-by-point rebuttal to the comments from all reviewers, the reporting summary, and a cover letter that explains the main improvements included in the revision and responds to any points highlighted in this decision.

Please follow the following recommendations:

- * Clearly highlight any amendments to the text and figures to help the reviewers and editors find and understand the changes (yet keep in mind that excessive marking can hinder readability).
- * If you and your co-authors disagree with a criticism, provide the arguments to the reviewer (optionally, indicate the relevant points in the cover letter).
- * If a criticism or suggestion is not addressed, please indicate so in the rebuttal to the reviewer comments and explain the reason(s).
- * Consider including responses to any criticisms raised by more than one reviewer at the beginning of the rebuttal, in a section addressed to all reviewers.
- * The rebuttal should include the reviewer comments in point-by-point format (please note that we provide all reviewers will the reports as they appear at the end of this message).
- * Provide the rebuttal to the reviewer comments and the cover letter as separate files.

We hope that you will be able to resubmit the manuscript within 25 weeks from the receipt of this message. If this is the case, you will be protected against potential scooping. Otherwise, we will be happy to consider a revised manuscript as long as the significance of the work is not compromised by work published elsewhere or accepted for publication at *Nature Biomedical Engineering*.

We hope that you will find the referee reports helpful when revising the work. Please do not hesitate to contact me should you have any questions.

Best wishes,

Michelle

Dr Michelle Korda
Senior Editor, Nature Biomedical Engineering

Reviewer #1 (Report for the authors (Required)):

The paper by Dr. Mancebo et al. introduces a novel intravenous oxygen (IVO₂) therapy using pH-sensitive polymeric carrier microbubbles. This method encapsulates oxygen in ~5 μm spheres, potentially carrying 600ml of oxygen per liter of polymer, a significant advancement in emergent care and resuscitation.

Subsequently the authors present data on an asphyxia model that could lead to arrest and used 3 injection of a total of 140 ml of oxygen (12 ml/kg with about 12 Kg pigs) while they also delivered an addition of ~250 ml of D10W. by doing so they were able to maintain arterial saturation around 20-25% for 4-5 minutes before ventilations were started again. That was significantly higher than the control animals.

The effect of adding 15-17% of minute Oxygen requirements and maintaining oxygenation at a higher saturation level than controls (still extremely lower even for a venous blood level) led to fewer animals requiring CPR and arresting and from there all parameters were improved.

It is of note that major effect for all the findings is the difference suggesting that little more oxygen supply for just 5 minutes is beneficial to the point that at minute 13-14 no animal was receiving CPR in the IVO₂ group.

The therapy was tested using an asphyxia model in pigs, demonstrating the ability to maintain arterial oxygen saturation significantly higher than control animals for 4-5 minutes, reducing the need for CPR and improving overall outcomes. Interestingly, the survival study indicated no major toxicity concerns except for some rodent splenic pathology.

Despite these promising results, the paper raises several concerns:

1. Model Clarification: The model used is not a cardiac arrest model, but rather one simulating prevention of extreme terminal hypoxia. This distinction is crucial for understanding the therapy's application and effectiveness.
2. Data Request: Additional data like PaO₂ and PV0₂ would provide a clearer understanding of the therapy's impact under extreme conditions.
3. Volume Delivered Discrepancy: There's confusion regarding the different volumes of oxygen delivered to pigs and rodents, specifically in relation to their basic metabolic oxygen consumption.
4. Rationale for Low Oxygen Delivery: The rationale behind the decision to deliver lower oxygen volumes, especially in pigs, is unclear. This choice seems to risk uncertain outcomes and doesn't align with the potential of the therapy.

In the rodents there was a delivery of 5 ml of solution with 80% foam containing 2.5ml of oxygen x5 for 0.5 Kg rats. That amounts to 1.7ml/kg/min of BM uptake of oxygen.

The pigs on the other hand, received only 3 doses of a total of 400 ml of solution of 60% foam containing only ~15% of BM uptake of oxygen. This is a bit confusing to me. What was the rationale of only giving such a small volume in of oxygen? Why not try to give 40-50% of metabolic rate consumption. I do not understand the rationale of why you would choose such a low saturation target that would have been very uncertain of the outcome if you deliver higher oxygen volumes.

If more experiments were done with higher volumes they need to be described and or discussed in relation to limitations of side effects.

5. Experimental Design Suggestions:

* An experiment with the therapy initiated after 12 minutes of asphyxia could provide insights into its effectiveness in more realistic emergency scenarios.

* Exploring the effects of delivering higher oxygen volumes and discussing potential side effects could provide a more comprehensive understanding of the therapy's limitations and capabilities.

* Since most of the patients that would be treated by someone who has the therapy available would have professional EMS CPR and the earliest it could be delivered is after an IV line is applied, 12 minutes before therapy is the best case scenario for CPR model.

6. Practical Application Concerns: The immediate availability of the therapy within 4-5 minutes of an asphyxia event seems unrealistic for most situations, limiting its practical use.

Addressing these issues would significantly enhance the understanding and potential application of this innovative therapy. The paper holds promise, particularly in pediatric care, due to its ability to meet oxygen requirements with smaller infused volumes. However, a more detailed and critical analysis is necessary for a comprehensive evaluation.

Reviewer #2 (Report for the authors (Required)):

This is a very interesting study presenting a novel oxygen carrier which can deliver oxygen intravenously. The O₂ carrier shell immediately dissolves when given intravenously due a change to physiological pH. The study describes the design and manufacture of this product, as well as providing data on its use in a swine model of asphyxia/CPR; in addition, as studies on the safety profile of this therapy is provided. The authors

are to be congratulated on a very nice detailed study. The in vivo results are quite impressive with major changes in very clinically important endpoints in the active treatment group.

GENERAL COMMENTS

I find the study quite interesting – the approach is novel, and the results are interesting and impressive. My only major concern is that the animal model is not very clinically relevant, although it is ideal from a proof of concept perspective.

It's not clear how much O₂ is able to be delivered by how much fluid. The authors state that 400 ml of oPMB contains ~140 ml of O₂; this is pretty clear but the authors state that this is equivalent ~15% of resting O₂ consumption. The O₂ consumption of 10 kg pigs would be ~50 ml/minute, so I don't understand how this works out. So how was the 15% estimated?

The therapy requires a substantial volume of fluid to deliver a minimal amount of oxygen. For example, (if I've done the calculation correctly) the authors used a volume of fluid that would be equivalent to about 4 liters in a 70 kg human, but yet only were able to deliver less than 50% of the (normal) O₂ requirements for just a single minute. This would be hugely problematic in the clinical setting. It would only work in situations when the clinician knows exactly when there is a huge oxygen deficit in blood. In the model they use of asphyxia with a closed airway, they know exactly when to give the IV O₂. Most patients with cardiac arrest would have an open airway or partially open airway, and would suffer more from ischemia than hypoxemia. In the discussion, I think the authors should discuss these potential limitations in the clinical application of this approach.

SPECIFIC COMMENTS

1. Pg 1, para 1:

- a. The authors state that respiratory disorders disrupt O₂ diffusion in lungs leading to low O₂ saturation. Problems with O₂ diffusion are not the major factors leading to hypoxemia; issues with ventilation-perfusion inhomogeneity are much more common.
- b. The authors state that hypoxemic respiratory failure was a leading cause of death during COVID-19. Interestingly, most deaths due to ARDS are ascribed to multiple organ failure, not hypoxemia, as implied by the authors.
- c. Hypoxemia is clearly important in the situations described in this paragraph, it is likely the ischemia (which then leads to tissue hypoxia) is likely more important than the hypoxemia per se.

2. I understand that some aspects of the protocol were blinded (e.g., neuroradiological assessment), but it's not clear if all aspects of the study were blinded. Can this therapy be given in a blinded fashion?

3. Pg 4, para 1:

- a. I think that a pulmonary hemorrhage model would be different in two ways: (1) it usually would not lead to complete obstruction of all airways/alveolar regions unless extremely severe, and (2) if it was so severe as to cause massive asphyxia, it would not be quickly reversible, which is a requirement for this approach to work since only a small volume of O₂ can be given because of fluid overload.
- b. Was the 400 ml of fluid given 3 times, or was it the total volume? As well the figure states 21 ml/kg, which would not be exactly 400 ml in all animals.
- c. "At minutes 6, 8 and 10 of apnea, swine were randomized to receive IVO₂ (total 400 ml of 60% oPMBs, containing ~140 ml oxygen, which represents ~15 % of resting oxygen consumption." It's not clear to me how much oxygen is available per liter of fluid. As well, it's not clear to me what period you are referring to when you say this is equivalent to 15% of resting oxygen consumption.

4. Pg 4, para 2:

- a. Was PaO₂ measured and pH measured. Would be helpful to have PO₂ data presented
- b. The authors ascribe the increase in PaCO₂ after giving IV O₂ to preserved cellular metabolism. It may also be due to the Bohr (or maybe it's the Haldane) effect.
- c. Figure 3d: Trivial point, but I think it would be better to present these data with percent of animals receiving CPR on the Y-axis (as opposed to those NOT receiving CPR)
- d. Figure 3g: Please present earlier time points as well

5. Pg 4, para 3: Presumably the results for Figure 3i (and subsequent panes in Figure 3) were obtained only in animals that survived. As such, these data are even more impressive than they appear since the sickest animals (that died) are not included. I know this is obvious, but it's worth pointing out to the reader.

6. Safety Study: The safety profile looks very reasonable but the numbers are somewhat small and there are a few data points that look like they might be relevant (e.g., lower weight gain at 7 days, clotting time at 4 days, maximum lysis at days). This is not a major issue; just might be worth mentioning the relatively small numbers.

7. Does blood osmolarity change after therapy, and if so by how much?

8. Pg 5, last para:

- a. Replace "...expected response to hypoxia..." TO "...expected response to very severe hypoxemia ..."
- b. It's not clear which substrate you are referring to.
- c. Providing residua of the polymer shell is useful when the "patient" is volume deplete, but it may be problematic in many situations when that is not the case.

9. Pg 6, para 1:

- a. Diffusion barrier: As discussed above this is not a major issue in most cases of hypoxemia due to respiratory disease
- b. In this model, lung volume likely decreases substantially during airway occlusion.

10. Discussion:

- a. As described above, a more robust discussion of the potential problems in using this approach would be helpful
- b. What is the theoretical maximal concentration of O₂ to the required fluid using this approach?

Reviewer #3 (Report for the authors (Required)):

SUMMARY:

The authors describe a LmD polymeric microbubble (PMB) gas carrier that can be used to deliver oxygen intravenously. This PMB dissolves at physiologic pH into excretable molecular constituents, which allows for rapid delivery of oxygen while avoiding apparent toxicity and vascular obstruction, unlike other gas carriers. The authors use a rat model to demonstrate that there is no evidence of adverse hemodynamic effects of vascular obstruction following PMB administration. In a swine model of acute hypoxemic respiratory failure and subsequent cardiac arrest, PMB administration improved SaO₂, time to ROSC, survival, and neurologic outcomes as compared with control. Additionally, safety studies performed in rats showed renal and hepatic clearance (though with low levels of LmD found in the spleen, liver, and kidneys after 7 days), and no laboratory evidence of end organ, metabolic, or hematologic dysfunction.

This is a compelling area of research with a clear unmet need. The data presented will be of interest to a broad readership. There are several questions/points that would support strengthening the current manuscript and these have been delineated below.

Major:

1. How stable are the PMBs following oxygenation? How quickly does oxygen diffuse through the polymer shells and how soon after oxygenation does it have to be injected?
2. Is there any hydrogen bonding expected between different PMBs at the concentrations used to inject in vivo, and would that have any effect on the rate of dissolution or likelihood of obstruction?
3. Please comment on why spleen macrophage vacuole formation was observed in rodents but not swine.
4. Fig 2b claims that the "recovery of attenuated acoustic signals within 3 seconds illustrates the rapid dissolution and outgassing of PMBs in PBS buffer". However, ultrasound can cause cavitation of microbubbles, which may affect the observed recovery rate. To support the claim that solution pH is responsible for the observed recovery, the authors should show the acoustic attenuation versus time when PMB added to solutions of various pH (e.g. pH 3-10).
5. Would a femoral catheter be required for PMB administration? What are the theoretical limitations regarding route of administration, especially if this were to be given in a true emergent setting? What

limitations might peripheral administration or smaller catheter diameter impose? Could it be given via IO catheter?

6. The way foam concentrations are referred to throughout the text is confusing. Please make the format consistent (e.g. total volume, O₂ volume, foam percentage). Also, a table that compares the concentrations / infusion rates etc. for the various experiments would be helpful for a reader. Is there a theoretical limit for gas infusion rate based on your experiments?

7. Translational considerations. A discussion on long term stability and general setting in how the technology could be applied would help the audience understand if these advances are translatable to human application. For example chow stable are the PMBs over a range of conditions e.g. Zone IVb, or would one require instrumentation at the bedside or near to support the continual generation of PMBs.

8. Chronic dosing safety. The reported safety studies noted single dosing of the materials in rodents. Safety on repeated dosing could also be of interest to the community. Understanding the effects of repeated dosing to potentially support subjects with compromised oxygenation (e.g. severe ARDS) could expand the impact of the intervention.

Minor:

1. Fig 2C: using dotted lines for different pH values is pretty hard for me to distinguish; can they make the labeling more clear?
2. Why do the IFNP MBs used as a control in Fig 2c have such a sharp cutoff? Were these filtered by size?
3. Fig S8: size in μm , not nm. Also, why is the y axis different from 1h?
4. Fig S13: x axis needs units
5. How long is the LmD solution homogenized before HCl is added?

Mon 15 Apr 2024

Decision on Article NBME-23-2760A

Dear Prof Peng,

Thank you for your revised manuscript, "Injectable oxygen improves outcomes in severe hypoxemia", which has been seen by the original reviewers. In their reports, which you will find at the end of this message, you will see that the reviewers acknowledge the improvements to the work and that Reviewer #2 raises a few welcome and relatively minor suggestions for textual accuracy that we hope you will be able to follow, to better describe the actual findings described in the manuscript and avoid overemphasizing the implications of the work.

As before, when you are ready to resubmit your manuscript, please upload the revised files, a point-by-point rebuttal to the comments from Reviewer #2, and the reporting summary.

We look forward to receive a further revised version of the work. Please do not hesitate to contact me should you have any questions.

Best wishes,

Pep

Pep Pàmies
Chief Editor, Nature Biomedical Engineering

Reviewer #1 (Report for the authors (Required)):

The authors have answered my questions. Differences in the opinion of in hospital and out of hospital response times care very relevant and a point to debate but at this time point in the evolution of this idea are not relevant.

Although I believe that the observed overall effect is very small (as far as the total volume of oxygen being able to be delivered) and possibly difficult to implement clinically to find a meaningful effect as currently deployed, the process is very innovative.

The findings are a significant advance in the field of alternative oxygenation strategies that can affect critical care and emergency care in the years to come and therefore warrant publication. I congratulate the PI and his team for such a significant body of work.

Reviewer #2 (Report for the authors (Required)):

The authors did a very good job of responding to most of the reviewers' comments and the manuscript is greatly improved from the previous version. I have a few specific queries (see below), but my main concern is that the authors over-sell the potential utility of their approach. In reading parts of the paper, it sounds as if the authors have developed a therapy for many common forms of acute hypoxemia; this is not the case because of the large volume of fluid needed to deliver the oxygen. As stated by all the reviewers previously, the animal model of hypoxemia is very specific: acute very severe hypoxemia, fully reversible cause of hypoxemia; and knowledge of exactly when to treat. I think this is ok from a proof of concept perspective but the authors have to tone down the implied importance of their findings. I've given a number of specific examples below.

1. Abstract:

a. The abstract should give the quantitative volume of fluid needed to provide a given volume of oxygen.

Without this, the Abstract reads like their approach is useful for many subacute causes of hypoxemia

b. Sentence starting “Administration of this carrier in swine with profound hypoxemia ...”. It’s important to state that the model is an acute, short model of complete airway occlusion lasting a few minutes.

c. Same point for the last sentence of the Abstract; it’s a bit misleading to say that this is a potentially viable therapy in hypoxic disorders. The data show that it’s potentially valid in a very unique cause of hypoxemia. A broader statement about utility would require different clinically relevant model(s).

2. Second sentence of Introduction (“In hospitalized patients,...”): This sentence mixes clinical causes with physiological mechanisms, and is thus a bit unfocused. For example, hypoxemia due to lung injury is usually largely due to ventilation-perfusion mismatch.

3. 1st paragraph of Introduction: The last few sentences discuss IHCA with 15-40% being caused by respiratory insufficiency. Although the authors don’t explicitly state this, it implies that their therapeutic approach may be potentially useful for many of these cases. I don’t think this is the case. I think the introduction should address the mechanism of hypoxia that is simulated by their experimental model. I think this would provide a more realistic perspective for the reader.

4. 2nd paragraph: The authors state that transfusion of oxygenated blood is by itself not a viable way to treat severe hypoxemia because of the huge volume of blood that would have to be given. This is absolutely correct, but the approach of IVO2 also requires very large volumes of fluid (albeit not of red blood cells). Again, I think this sets the stage for the reader to come away thinking that the current approach is potentially more useful than it may in fact be.

5. Section entitled “Instrumentation and mode description”:

a. It’s not clear to me whether the animals in fact had cardiac arrest of severe hypotension based on the statement: “Animals experiencing cardiac arrest (systolic blood pressure <40 mmHg) ...” What is your definition of “cardiac arrest”?

b. The treated group had an n=8, while the control group had an n=10. Why is the sample size different between groups? Are all animals reported in this study, or did you do experiments in some animals that you are not reporting? If the latter, please give details of which animals were not included and why their data was not used.

6. Section entitled “Effect of oPMBs on resuscitation metrics”: The authors state that at 6 minutes of asphyxia/apnea, a similar number of animals experienced cardiac arrest. In reviewing fig. S21, it looks like just prior to 6 minutes about double the number of animals in the treatment group were free of cardiac arrest. Can you provide more detailed data on the physiological behavior of the animals in both groups just before 6 minutes. What was their BP, how many had complete cardiac arrest, etc. Given the small number of animals, it’s possible the Control group were somewhat sicker to start before treatment.

7. Discussion:

a. I think the first paragraph of the discussion there should be an explicit statement of how much fluid is required to provide a given volume of oxygen, e.g., 750 ml of fluid to provide 250 ml O₂.

b. Para 2: I think it’s important to emphasize that in this model, you know exactly how much oxygen has been used up and exactly when to give it. This will not be so easy in the clinical setting. This wouldn’t necessarily be a critical issue except that you have to give so much fluid to get a small amount of oxygen infused

c. Statement “...a given dose of oxygen may also have a more pronounced effect when used in clinical settings of lung injury.” I don’t understand the rationale underlying this statement.

d. Next sentence, “...in this model of airway obstruction, oxygen tension of the pulmonary artery exceeded that of the alveolus.” Did you present evidence to support this statement?

8. In the section on limitations, the authors’ give a value of ~200 mL/min for the normal O₂ consumption. I

understand that this is just a rough figure, but a more accepted value would be about 250 mL/min.

Reviewer #3 (Report for the authors (Required)):

The authors have addressed all of the concerns previously. Specifically:

-The experiments in the new Fig 2 sufficiently addressed the concern that the authors' use of ultrasound within their measurements was causing the bubbles to prematurely burst (comment 4). These new experiments support the authors' original claim that higher pH increases the speed of bubble dissolution/outgassing.

-The accelerated stability study (Figs S8-14) demonstrated the long-term stability of the foam (comment 1).

This is a wonderful study and contribution that will be of interest to a broad audience.

Wed 15 May 2024

Decision on Article NBME-23-2760B

Dear Prof Peng,

Thank you for your revised manuscript, "Injectable oxygen improves outcomes in severe hypoxemia". Having consulted with Reviewer #2 (whose comments you will find at the end of this message), I am pleased to write that we shall be happy to publish the manuscript in *Nature Biomedical Engineering*, provided that the points specified in the attached instructions file are addressed.

When you are ready to submit the final version of your manuscript, please upload the files specified in the instructions file.

We encourage authors to take up transparent peer review. If you are eligible and opt in to transparent peer review, we will publish, as a single supplementary file, all the reviewer comments for all the versions of the manuscript, your rebuttal letters, and the editorial decision letters. **If you opt in to transparent peer review, in the attached file please tick the box 'I wish to participate in transparent peer review'; if you prefer not to, please tick 'I do NOT wish to participate in transparent peer review'**. In the interest of confidentiality, we allow redactions to the rebuttal letters and to the reviewer comments. If you are concerned about the release of confidential data, please indicate what specific information you would like to have removed; we cannot incorporate redactions for any other reasons. More information on transparent peer review is available.

Best wishes,

Pep

Pep Pàmies
Chief Editor, Nature Biomedical Engineering

P.S. Nature Portfolio journals encourage authors to share their step-by-step experimental protocols on a protocol-sharing platform of their choice. Nature Portfolio's Protocol Exchange is a free-to-use and open resource for protocols; protocols deposited in Protocol Exchange are citable and can be linked from the published article. More details can be found at www.nature.com/protocolexchange/about.

Reviewer #2 (Report for the authors (Required)):

The authors have adequately addressed my concerns. One trivial suggestion is that the authors change this statement in the Abstract "The carriers deliver 35-50 mL of oxygen per dL of foam." TO "The carriers deliver 350-500 mL of oxygen per L of foam."

Nature Biomedical Engineering is a Transformative Journal. Authors may publish their research with us through the traditional subscription access route, or make their paper immediately open access through payment of an article-processing charge. More information about publication options is available.

You may need to take specific actions to comply with funder and institutional open-access mandates. If the work described in the accepted manuscript is supported by a funder that requires immediate open access

(as outlined, for example, by Plan S) and your manuscript was originally submitted on or after January 1st 2021, then you will need to select the gold OA route. Authors selecting subscription publication will need to accept our standard licensing terms (including our self-archiving policies), and these will supersede any other terms that the author or any third party may assert apply to any version of the manuscript.

Rebuttal 1

Point-by-Point response to Reviewers' Comments

Summary of major changes:

We have extensively revised the main text, added new experimental details in the Method section in Supplementary Materials. All the text changes or additions are font blue, and all the new or modified figures are font red.

The orders of figures and (new) references were changed and organized accordingly.

We added a new **Supplementary Table-1** in Supplementary Materials, summarizing the infusion rate, gas fraction, and volumes in each animal experiment.

New Figures added: **Fig. 2a-g** (Dissolution kinetics); **fig. S8-S14** (Long-term stability); **fig. S15** (DLS measurement); **fig. S23** (changes in PaO₂ in swine efficacy experiment); **fig. S30** (changes in serum osmolarity in swine efficacy experiment)

Figures modified: **Fig. 4d** (corrected y-axis title), **Fig. 4g** (included more timepoints for MAPB), **fig. S16** (corrected legend); **fig. S21** (corrected unit).

Removed: original Fig. 2a-c, original Supplementary Videos 6 and 7.

[Editorial Comments]

Thank you again for submitting to *Nature Biomedical Engineering* your manuscript, "Injectable oxygen improves outcomes in severe hypoxemia". The manuscript has been seen by 3 experts, whose reports you will find at the end of this message.

You will see that the reviewers appreciate the work. You will see that the reviewers appreciate the work, and that they raise a number of technical criticisms that we hope you will be able to address. In particular, we would expect that a revised version of the manuscript provides:

* Additional characterization data (including PaO₂, PV0₂ and pH), as suggested by Reviewers #1 and #2.

We have added the following results of arterial oxygen tension (PaO₂) into the Supplementary Materials (**fig S23**).

Fig. S23. The arterial partial pressure of oxygen (PaO₂) was significantly higher at 7, 9, 11 minutes in the IVO2-treated animals than those in the control group. Data presented as mean \pm SD; red = control, blue = oPMB-treated swine. Statistical analysis by a two-way ANOVA with Tukey's multiple comparisons post-test.

We have left this as a supplementary figure since, under conditions of severe hypoxemia the arterial oxyhemoglobin saturation (SaO₂) is a more clinically determinant parameter to follow given that it reflects arterial oxygen content much more so than PaO₂. Trends in PaO₂ may be misleading due to differences in oxygen dissociation related to acidosis and other clinical factors.[1]

Arterial pH was originally included as **fig. S20A**, now updated as **fig S29A** (below). No significant differences in blood pH were observed between treated and control groups.

Fig. S29. Changes in serum electrolytes in asphyxia efficacy study. **A**, pH. Data presented as mean \pm SD, statistical analysis by two-way ANOVA between control and treatment groups.

Regarding the systemic venous oxygen content, we were unable to measure venous oxygen tension (PvO₂) or venous oxyhemoglobin saturation (SvO₂) during cardiac arrest, as venous catheters were being used for resuscitation and experimental treatments.

*** Further evidence of the rapid dissolution of the microbubbles, and of their longer-term stability.**

We thank the Editor and Reviewers for this request. We have performed additional experiments to demonstrate the pH- dependent (rapid) dissolution kinetics and the long-term stability of PMBs. These new data are included as **Fig. 2**, and **fig. S8-14**. We added discussions in the main text and included experimental details in the Method section in the Supplementary Materials.

[Dissolution behaviors]

There are two critical aspects to the dissolution of the PMB shell: dissolution of the **gas core** and breakdown of the **shell** into its soluble components. The rapid dissolution of the gas core is essential to avoiding acute hemodynamic instability due to gas embolism and the dissolution of the shell is crucial to facilitate clearance. We have used several approaches to investigate this process as described in the new **Fig. 2**. The following text has been added to the main manuscript (page 3):

Dissolution of PMBs ex vivo

The intravenous administration of a gas requires the rapid dissolution of gas carriers to avoid vascular obstruction. Subsequently, the carrier materials must be biocompatible and rapidly cleared. As noted earlier, the LmD polymer itself exhibits pH-responsive behavior in solution due to the presence of carboxylic acid groups (pK_a ~ 4.8) and is molecularly soluble above pH 5 (**Fig. 2a**, **fig. S15**). In this work, the mechanism of dissolution of PMBs following injection hinges upon the pH-based deprotonation of carboxylic groups, which increases the solubility and hydration of polymers that compose the shell. This causes water influx in the shell, increasing surface tension and destabilizing the gas core, promoting its dissolution. The

deprotonated shell simultaneously reverts to small and soluble components. Notably, unlike lipid coated bubbles, which require a gas concentration gradient (i.e., sink) to dissolve^[24-26], PMBs dissolve at a physiologic pH within seconds even in the absence of a sink (**Supplementary Video 1**). To examine whether PMB shells fully dissolve and revert to soluble LmD constituents, we added PMBs into phosphate buffered saline (PBS) solution of varying pH under stirring and confirmed DLS size measurements 2 minutes following admixture (Earlier timepoint was not obtained by DLS due to the sampling limitation of the instrument). Between pH 9.0 and 5.0, PMB shells were all fully dissolved within 2 minutes, reverting to soluble polymers of similar sizes of LmD solutions prepared from solid states (**Fig. 2a**). Solubilized LmD polymer has a mean hydrodynamic radius less than 10 nm and an estimated MW ~12 kDa (determined by NMR in **fig. S2**), well below the MW cutoff for glomerular filtration (30 – 45 kDa)^[32,33]. In contrast, dissolution of previous IFNP MBs, which consisted of hydrophobic polymers of higher MW (>60 kDa), revert into large and insoluble nanoparticles (>100 nm) that visibly precipitate over time.

To better investigate the pH-dependent dissolution kinetics, we examined PMBs mixed with PBS while continuously applying ultrasound. Like various polymeric shelled microbubbles^[34,35 41,42], the gas core of PMBs creates acoustic backscatter and produces contrast in proportion to the presence of gas bubbles within the field of view. The decrement in contrast intensity (i.e., bubble dissolution) was shown to be pH-dependent: at pH 9.0, 7.2, and 6.5, PMBs were no longer visible within 2 to 3 seconds, while dissolution of the gas core was prolonged at pH<6 (**Fig. 2b,c**). (To note, although the LmD shell is less soluble at pHs 4.8 and 3.8, we noticed the gas core of PMBs slowly becomes fluid-filled as shown by the slow decrease in echo intensity, this is likely because the salts in PBS affected the swelling of the hydrogel-like shell). While it is known that ultrasound may contribute to loss of MBs due to inertial cavitation, **Fig. 2b** suggests that pH is the dominant factor affecting dissolution rate. To further account for acoustic destruction, we performed the same experiments while applying ultrasound only at selected terminal time points, finding similar dissolution rates (**Fig. 2d,e**).

To further examine dissolution of the shell (separate from that of the gas core), we performed UV-vis spectroscopy. In this construct, it is expected that an increase in absorbance from baseline could be caused by either undissolved gas core or a large polymeric aggregate (i.e. undissolved shell or aggregated constituents). From pH 9.0 to 6.0, UV absorbance reached baseline within 2-3 seconds, similar to the kinetics in acoustic studies, indicating both that the gas core had dissolved, and the shell reverted to its soluble constituents in that time (**Fig. 2f,g**). Between pH 5.5 and 5.0, the return to baseline was much longer than in the acoustic study, suggesting that within this pH range the gas core dissolves first, and the shell required more time to revert to soluble polymers. These findings are consistent with the pH-triggered dissolution mechanism of PMBs that is essentially an acid-base reaction, the rate of which is proportional to the concentration of hydroxyl ions in the solution and limited by diffusion. This mechanism also explains some discrepancies of dissolution kinetics seen at lower pHs. For example, in contrast to Uv-vis, DLS showed PMBs were fully dissolved at 2 minutes at pH 5.5 and 5.0, this is likely due to lack of sufficient mixing in Uv-vis experiments. However, at pH>6.0, the dissolution kinetics measured from various methods were all in good agreement. Collectively, these results validate our proposed design for the new gas carrier and established that both the gas core and the shell of PMBs rapidly dissolve at pH levels (7.5 to 6.5) that are relevant to

intravenous injection, as the blood pH rarely drops below 6.5 even in extreme instances.

Additionally, **Fig. 2** is shown here for your convenience.

Fig. 2 | LmD PMBs rapidly dissolve at physiologic pH. (a) DLS measurement of size following mixing of LmD PMBs mixed in PBS solution for 2 minutes at varying pH. LmD PMBs fully dissolve above pH 5 and revert to their soluble components with a mean size < 10 nm, similar to those of LmD solutions prepared from solid states. In contrast, the previous generation of IFNP MBs (made from more hydrophobic polymers) led to formation of much larger nanoparticles. (b) Phantom sonography of aPMBs in aerated PBS shows the pH-dependent dissolution rate, evidenced by the disappearance of contrast intensity produced by the gas core under continuous ultrasound. In the absence of a gas sink, PMBs rapidly dissolved above pH 6 within seconds. Data presented as change in contrast/bright area from baseline. Data are means, error = SEM. Continuous measurements collected as biological replicates. (To note, although the LmD shell is less soluble at pHs 4.8 and 3.8, we noticed the gas core of PMBs slowly becomes fluid-filled as shown by the slow decrease in echo intensity, this is likely because the salts in PBS affected the swelling of the hydrogel-like shell). (c) Representative images from phantom sonography study that show the dissolution profile of aPMBs at different time points at various pHs. (d, e) To account for any destructive effect that ultrasound itself has on PMBs, the experiment was repeated while only applying ultrasound at the expected dissolution time from (b), showing similar dissolution times even absent the application of continuous ultrasound. Measurements collected as biological replicates. Data presented as change in area of contrast/brightness from baseline, error = SEM, analyzed by student's t-test. (f,g) Dissolution of the shell and gas core was then studied using UV-vis absorbance spectroscopy. Continuous measurements collected as biological replicates. Similar to the characterization using ultrasound (which detects only dissolution of the gas core), UV-vis returns to baseline within seconds at pH above 6, suggesting both that the gas core has dissolved and that the shell has broken down into its constituent components. At more acidic pH, return of UV-vis absorbance to baseline took 10 minutes or longer. Contrasting this with sonographic experiments (e.g. b)

in which ultrasound scatter returned to baseline within 90-260 seconds, these findings suggest that following dissolution of gas core, the remaining shell constituents take additional time to dissolve and revert to soluble components. Data presented as change from baseline, error = SEM.

Collectively, these characterizations demonstrate that the dissolution of PMBs is pH-dependent, and both the gas core and the shell rapidly dissolve within seconds at physiologically pH levels relevant to intravenous injection.

These in vitro results also validate the use of sonography as a tool to monitor real-time PMB dissolution kinetics in vivo which as in **Fig. 3**. As in **Fig. 3g-l**, echocardiography showed that oxygen- and air-filled PMBs rapidly dissolve before reaching the left heart in healthy animals (i.e. within seconds). In contrast, the lipid based LOMs persist in the entire circulation for minutes following injection, which contributes to the significant risk of embolism and hemodynamic instability.

[Long-term stability]

Previously, the longest stability data that we had was obtained after 3-month storage of PMBs in D10 at room temperature, in which we did not observe change in foam volume or size. Longer-term studies were marred by bacterial contamination, a hard-to-avoid artifact when fabricating biomaterials in a non-GMP laboratory environment.

Thus, we addressed the Reviewers' question on the long-term stability of oxygen-filled PMBs (oPMBs) using an accelerated stability study at various temperatures over 30 days, monitoring changes in foam height and PMB size. In these studies, oPMBs were packaged either in capped syringes or plastic tubes and placed in temperature-controlled, oxygen purged incubators (pO₂ of the oPMB solution was maintained at 740 mmHg). These results are provided in new **fig. S8-14**, showing that PMBs were stable up to 45°C in glass syringes with no measurable change in foam volume or size distribution. Interestingly, there was some PMB loss at 45°C with plastic tubes, likely due to the surface-specific drying effects between plastic and the PMB shell may have compromised the shell integrity. Storage at 60°C exhibited rapid loss of PMBs (**fig. S13**), likely due to breakdown of the hydrogen bonds that stabilize the PMB shell. It is important to point out that PMBs lack the coalesce and ripening behavior seen with lipid-based oxygen microbubbles (LOMs) due to their solid shell structure. If any, the destabilization of PMBs mainly resulted from water-influx into gas core due to compromised shell structures, which led them to sediment and separate from the top foam. For instance, as shown in **fig. S8,9**, the remaining foam in the tube at 45°C did not change in size distribution, the sediment of fluid-filled PMBs at the bottom was dispersible upon shaking and can readily dissolve at physiological pHs. The following text has been added to the main manuscript (page 2-3) and also included relevant details in method section:

The long-term stability of oPMBs was evaluated via an accelerated stability test at various temperatures, as shown in **fig. S8-14**, oPMBs stored in glass syringes were stable up to 45°C up to 30 days without changes in foam volume or size distribution, while loss of oPMBs were observed at 60°C. Based on these results, we expect the shelf-life of oPMBs to be at least 6 months at room temperature and at least 1 year under refrigerated conditions.

We thank the Editor and Reviewers for requesting us to perform these additional experiments, we believe the new results have helped significantly improved this manuscript.

*** Clarification of the oxygen-dosage calculations for the pigs, as queried by Reviewers #1 and #2, and comparison of the gas-infusion rates, as per the comments by Reviewer #3.**

Thank you for this important point. We offer the following calculations and logic:

- The mean measured resting oxygen consumption (VO₂) of swine in the asphyxia experiment was 73.2 mL/min.
- We administered 140 mL of oxygen over 12 minutes of asphyxia = ~11.6 mL/minute
- 11.6 mL O₂/min provided/73.2 mL O₂/min resting VO₂ = 15.8% of resting VO₂

Parenthetically, we believe that 15% of resting VO₂ is a conservative estimate in the sense that oxygen was administered only during the last 6 minutes, such that it can also be stated that we administered ~30% of resting VO₂ during the intervention period (i.e. that consumed during the latter 6 minutes). Further, it is known that in settings of severe limitations to oxygen delivery (with hypoxemic cardiac arrest being the most extreme example of this) that VO₂ significantly diminishes, such that the proportion oxygen provided to oxygen being consumed is likely significantly higher in the setting of IHCA. To enhance the clarity of this statement, we have revised the manuscript as follows (page 4):

At minutes 6, 8 and 10 of injury, swine were randomized to receive either IVO₂ (combined total 400 ml of 35% vol O₂/vol foam oPMBs, containing ~140 ml oxygen, n=8) or an equal volume of oxygenated D10 (n=10). Given that the mean measured resting oxygen consumption (VO₂) during the baseline period in this experiment was 73.2 mL/min, this volume represents the provision of ~30% of resting oxygen consumption for the last 6 minutes of asphyxia.

To further clarify the volume of gas delivered in each experiment in both rats and swine, we have added the following table in the end of Method section in the Supplementary materials.

Supplementary Table 1. Summary of doses of PMBs used in various animal studies.

	Figure	Gas content (oxygen fraction)	Gas fraction	Gas injection rate (ml of gas/kg/min)*	Total gas volume infused	Total volume infused
Rodent hemodynamic study	Fig. 3c-f	Air (21%O ₂)	50% vol gas/vol foam	5 ml/kg/min	25 ml/kg	50 ml/kg
In vivo dissolution study by echo	Fig. 3g-l	Air (21%O ₂) and 100% O ₂	40% vol gas/vol foam	4 ml/kg/min; 8ml/kg/min; 12 ml/kg/min	4 ml/kg; 8 ml/kg; 12 ml/kg	10 ml/kg; 20 ml/kg; 30 ml/kg
Swine efficacy study	Fig. 4	100% O ₂	35% vol gas/vol foam	4 ml/kg/min	12 ml/kg	33 ml/kg
Rodent safety	Fig. 5	100% O ₂	40% vol gas/vol foam	1.3 ml/kg/min	13 ml/kg; 26 ml/kg	32 ml/kg; 64 ml/kg

*Gas injection rate was calculated as total foam volume injected/injection time (i.e., the time during which injections were actually occurring).

*** Clarification of the rationale for the choice of the animal model, as suggested by Reviewers #2 and #3.**

We apologize for our lack of clarity on the animal model, which was intended as a model of in-hospital cardiac arrest (IHCA), which occurs in more than 200,000 patients per year in the US alone.[2] IHCA has a survival rate between 25-30%, and neurologic injury is common in survivors.[3-5] In both children and adults, IHCA is most commonly caused by respiratory failure and hypoxemia, which is also an independent risk factor for mortality.[6] Survival from IHCA diminishes with each passing minute, and therefore achieving early return of spontaneous circulation (ROSC) is the target of all resuscitative efforts.

Patients suffering from IHCA are by definition hospitalized, usually have an in situ intravenous line, and are commonly in an intensive care, heavily monitored environment. Administration of resuscitative drugs (e.g. epinephrine) within the first few minutes of a resuscitation is expected, so much so that timing to first dose represents a metric of resuscitation quality. For example, in a Get With the Guidelines registry analysis of 25,095 adults from 570 hospitals with IHCA and non-shockable initial rhythm, the median time to administration of the first dose of epinephrine was 3 minutes (IQR 1-5 minutes); 82% of patients received the first dose of epinephrine within 6 minutes, more than 50% of patients within 3 minutes.[7]

We envision that intravenous oxygen could be co-administered along with epinephrine as a first-line resuscitative drug. As such, the model we describe here represents a very realistic emergency use of IVO2. The value of the experiment that we described was in part to highlight that administration of the drug early on in the resuscitation enhances return of spontaneous circulation and provides 'rescue' volumes of oxygen to the tissues, preventing downstream sequelae. While it will be additive to understand the effect of the drug when administered at later time points within a resuscitation, the timing we described here is very achievable in the vast majority of patients with IHCA.

Of note, in this series of pulseless electrical activity (PEA)/asystole patients with IHCA, the incidence of successful resuscitation was only 49%, survival to 24 hours 27%, and survival to hospital discharge 10%. Thus, the clinical outcomes in IHCA are nearly as poor as the control group in our swine study.

To better explain the rationale for our model, we have placed the following text at the beginning of the introduction (page 1):

Hypoxemia, or low blood oxygen content, can occur in the setting of lung disease or airway obstruction, which causes blood to circulate through the lungs and then to the body without being fully reoxygenated. In hospitalized patients, severe, episodic hypoxemia can result from endotracheal tube occlusion (e.g. secretions), progressive lung injury, ventilation-perfusion mismatch, and a number of other causes. Such episodes may be addressed with airway clearance, lung recruitment, increased ventilatory support, and when needed inhaled nitric oxide or extracorporeal membrane oxygenation. However, sometimes hypoxemia is temporarily so severe and refractory to these maneuvers that myocardial contractility fails, resulting in pulseless electrical activity or even asystole; this clinically manifests as cardiac arrest. Of note, 15-40% of in-hospital cardiac arrests (IHCA) are caused by respiratory insufficiency.^[1] Such patients exhibit a hypoxic insult compounded by an ischemic insult (i.e. cessation of cardiac activity and blood flow), resulting in severe ischemic injury to the brain, kidneys and other organs. Survival to hospital discharge following IHCA approximates 20%, with ~1 in 3 patients suffering from significant neurologic impairment. ^[2] It is well recognized that survival following IHCA is enhanced by maneuvers that optimize oxygen delivery during IHCA (such as high quality CPR) ^[3],

achieving early return of spontaneous circulation^[4], and rapid cannulation to ECMO during CPR^[5]. Successful resuscitation from IHCA requires the rapid identification and reversal of the underlying cause^[6], however, reversing refractory hypoxemia remains an unmet challenge.

*** Discussion of the translatability of the model to realistic emergency situations, as suggested by all reviewers.**

As described above, we believe that the injection of IVO₂ as performed in this model in the inpatient setting is quite realistic and within expected performance. We have added the following text to the Discussion to clarify the translatable potential of the model to emergency situations (page 6):

In clinical practice, we envision that IVO₂ would be available on-demand in environments caring for critically ill patients at risk for hypoxemia. Because arterial oxygen saturation is often continuously monitored by photoplethysmography in patients at risk for hypoxemia, IVO₂ could become a new treatment for refractory hypoxemia that is refractory to current standards of care, including airway clearance (e.g., suctioning), lung recruitment (e.g., hand ventilation) and the use of other critical maneuvers. Its dosing could be titrated in the same way that pre-arrest bolus doses of epinephrine are titrated to treat refractory hypotension,^[57] since both blood pressure and arterial oxygen saturation (by plethysmography) are routinely measured as vital signs in hospitalized patients. As in the swine model we described, patients being treated for in-hospital cardiac arrest (IHCA) known or presumed to be caused by hypoxemia could also be treated with IVO₂ as an emergency treatment; because such patients standardly receive IV resuscitative treatments within 3-5 minutes^[58], IVO₂ may restore early spontaneous circulation and significantly improve outcomes in such patients.

*** Discussion of the limitations of the approach, as per the relevant comments of all Reviewers.**

Thank you so much for this comment. We have added the following paragraph to the Discussion (page 6-7):

The major limitation of this technology is the volume of the fluid which must be administered to deliver the oxygen payload. In this work, the gas fraction of the foam ranged between 35-50%, which can likely be optimized further as it has with similar products.^[15,59] However, given the high rate of resting oxygen consumption (e.g. a healthy adult consumes ~200 mL/minute), oxygen provision using this technology is unlikely to provide more than several hundred milliliters of oxygen, particularly in critically ill patients with heart disease in whom even small quantities of administered volume administration may aggravate venous hypertension and pulmonary edema. Thus, IVO₂ should be considered as a strategy for the IV bolus administration of oxygen in critical situations, to reverse a critical physiology (such as a cardiac arrest or a pulmonary hypertensive crisis) or to bridge such patients to definitive treatment (such as extracorporeal membrane oxygenation support). The oxygen deficit that accrues over even a few critical minutes can be the difference between a favorable outcome and manifesting of neurologic injury for a lifetime, or be the difference between survival and death.^[5] From an experimental perspective, another important

limitation to note is that some members of the research team in the cardiac arrest experiment could not be blinded to treatment group due to the acute and rapid significant differences in physiologic response of the patient to the treatment. An additional practical challenge for blinding is that the PMB solution has a distinct white/opaque color that is easily identifiable. Although we could have taped the all the syringes and lines, it was challenging to completely mask the drug, particularly when priming the injection ports or changing the syringes during treatment, since the drippings from the syringes or residues at injection port were immediately visible. However, this limitation was balanced by actively measuring the quality of CPR (which was not different between groups), and by strictly following a treatment protocol throughout all phases of the experiment (including during the resuscitation and survival periods). Finally, the number of animals in the safety study was somewhat low, such that we were underpowered to detect minor differences between groups.

Moreover, because of sensitivities around animal experimentation in society, and given the limited scientific value of Supplementary Videos 6 and 7 depicting the pig, we strongly recommend that these videos be removed.

Thank you for this comment. We agree and have removed these two videos.

Reviewer #1 (Report for the authors (Required)):

The paper by Dr. Mancebo et al. introduces a novel intravenous oxygen (IVO2) therapy using pH-sensitive polymeric carrier microbubbles. This method encapsulates oxygen in ~5 µm spheres, potentially carrying 600ml of oxygen per liter of polymer, a significant advancement in emergent care and resuscitation.

Subsequently the authors present data on an asphyxia model that could lead to arrest and used 3 injection of a total of 140 ml of oxygen (12 ml/kg with about 12 Kg pigs) while they also delivered an addition of ~250 ml of D10W. by doing so they were able to maintain arterial saturation around 20-25% for 4-5 minutes before ventilations were started again. That was significantly higher than the control animals.

The effect of adding 15-17% of minute Oxygen requirements and maintaining oxygenation at a higher saturation level than controls (still extremely lower even for a venous blood level) led to fewer animals requiring CPR and arresting and from there all parameters were improved.

It is of note that major effect for all the findings is the difference suggesting that little more oxygen supply for just 5 minutes is beneficial to the point that at minute 13-14 no animal was receiving CPR in the IVO2 group.

The therapy was tested using an asphyxia model in pigs, demonstrating the ability to maintain arterial oxygen saturation significantly higher than control animals for 4-5 minutes, reducing the need for CPR and improving overall outcomes. Interestingly, the survival study indicated no major toxicity concerns except for some rodent splenic pathology.

We thank the Review for the comprehensive assessment.

Despite these promising results, the paper raises several concerns:

1. Model Clarification: The model used is not a cardiac arrest model, but rather one simulating prevention of extreme terminal hypoxia. This distinction is crucial for understanding the therapy's application and effectiveness.

Thank you for this important comment. We apologize for our lack of clarity on our animal model, and would like to provide greater context surrounding the injury model. Many different models of cardiac arrest have been described.[8] One model is one of coronary occlusion or injection, which induces the local myocardial ischemia and subsequent arrhythmia and cardiac arrest that is the most common cause of **out**-of-hospital cardiac arrest (OHCA). The model that we used is one of asphyxial cardiac arrest (ACA), a common cause of **in**-hospital cardiac arrest (IHCA).

There are two main variations of swine models of ACA:

1. The first model is the one used in our manuscript in which asphyxia is induced and the animal simply monitored until cardiac arrest ensues, rather than inducing electrical cardiac arrest at a specific time point (as in variation 2 below).[9,10] This model most accurately mimics the clinical insults found in ACA and IHCA, in which severe hypoxemia leads to a predictable sequence of catecholamine surge (resulting in hypertension), followed by

progressive acidosis, then circulatory failure resulting from severe oxygen deprivation and dysfunction of the myocardium and endothelium. Cardiac arrest in this scenario is most often due to global and progressive myocardial failure rather than a lethal arrhythmia, as occurs in the vast majority of clinical cases of IHCA (81% of IHCA presents with asystole or pulseless electrical activity). [11]

2. One commonly used variation on this model is asphyxia followed by the experimental induction of ventricular fibrillation (VF, a lethal arrhythmia that is one cause of cardiac arrest) at a set time by applying direct current to the myocardium. This approach is favored in experiments studying the performance of CPR itself.[12]

In this case, we chose a model of ACA that most closely mimics the sequence of insults that most closely represents clinical ACA, and the ability of injectable oxygen to interrupt the progression to ACA. We note that 100% of swine experienced asphyxial cardiac arrest (ACA), making this a model of ACA.

2. Data Request: Additional data like PaO₂ and PV0₂ would provide a clearer understanding of the therapy's impact under extreme conditions.

Thank you for this comment. We have added the following results of arterial oxygen tension (PaO₂) into the Supplementary Materials (**fig S23**).

Fig. S23. The arterial partial pressure of oxygen (PaO₂) was significantly higher at 7, 9, 11 minutes in the IVO2-treated animals than those in the control group. Data presented as mean ± SD; red = control, blue = oPMB-treated swine. Statistical analysis by a two-way ANOVA with Tukey's multiple comparisons post-test.

We have left this as a supplementary figure since, under conditions of severe hypoxemia the arterial oxyhemoglobin saturation (SaO₂) is a more clinically determinant parameter to follow given that it reflects arterial oxygen content much more so than PaO₂. Trends in PaO₂ may be misleading due to differences in oxygen dissociation related to acidosis and other clinical factors.[1]

Vis a vis systemic venous oxygen content, we were unable to measure venous oxygen tension (PvO₂) or venous oxyhemoglobin saturation (SvO₂) during cardiac arrest, as venous catheters were being used for resuscitation and experimental treatments.

3. Volume Delivered Discrepancy: There's confusion regarding the different volumes of oxygen delivered to pigs and rodents, specifically in relation to their basic metabolic oxygen consumption.

Thank you for this important point. We offer the following computation and logic:

- The mean measured resting oxygen consumption (VO₂) of swine in the asphyxia experiment was 73.2 mL/min.
- We administered 140 mL of oxygen over 12 minutes of asphyxia = ~11.6 mL/minute
- $11.6 \text{ mL-O}_2/\text{min} \text{ provided} / 73.2 \text{ mL-O}_2/\text{min} \text{ resting VO}_2 = 15.8\% \text{ of resting VO}_2$

Parenthetically, we feel that 15% of resting VO₂ is a conservative estimate in the sense that oxygen was administered only during the last 6 minutes, such that it can also be stated that we administered ~30% of resting VO₂ during the intervention period (i.e. that consumed during the latter 6 minutes). Further, it is known that in settings of severe limitations to oxygen delivery (with hypoxemic cardiac arrest being the most extreme example of this) that VO₂ significantly diminishes, such that the proportion oxygen provided to oxygen being consumed is likely significantly higher in the setting of IHCA. To enhance the clarity of this statement, we have revised the manuscript as follows:

At minutes 6, 8 and 10 of injury, swine were randomized to receive either IVO₂ (combined total 400 ml of 35% vol O₂/vol foam oPMBs, containing ~140 ml oxygen, n=8) or an equal volume of oxygenated D10 (n=10). Given that the mean measured resting oxygen consumption (VO₂) during the baseline period in this experiment was 73.2 mL/min, this volume represents the provision of ~30% of resting oxygen consumption for the last 6 minutes of asphyxia.

To further clarify the volume of gas delivered in each experiment in both rats and swine, we have added the following table in Method section in Supplementary Materials.

Supplementary Table 1. Summary of doses of PMBs used in various animal studies.

	Figure	Gas content (oxygen fraction)	Gas fraction	Gas injection rate (ml of gas/kg/min)*	Total gas volume infused	Total volume infused
Rodent hemodynamic study	Fig. 3c-f	Air (21%O ₂)	50% vol gas/vol foam	5 ml/kg/min	25 ml/kg	50 ml/kg
In vivo dissolution study by echo	Fig. 3g-l	Air (21%O ₂) and 100% O ₂	40% vol gas/vol foam	4 ml/kg/min; 8ml/kg/min; 12 ml/kg/min	4 ml/kg; 8 ml/kg; 12 ml/kg	10 ml/kg; 20 ml/kg; 30 ml/kg
Swine efficacy study	Fig. 4	100% O ₂	35% vol gas/vol foam	4 ml/kg/min	12 ml/kg	33 ml/kg
Rodent safety	Fig. 5	100% O ₂	40% vol gas/vol foam	1.3 ml/kg/min	13 ml/kg; 26 ml/kg	32 ml/kg; 64 ml/kg

*Gas injection rate was calculated as total foam volume injected/injection time (i.e., the time during which injections were actually occurring).

4. Rationale for Low Oxygen Delivery: The rationale behind the decision to deliver lower oxygen volumes, especially in pigs, is unclear. This choice seems to risk uncertain outcomes and doesn't align with the potential of the therapy. In the rodents there was a delivery of 5 ml of solution with 80% foam containing 2.5ml of oxygen x5 for 0.5 Kg rats. That amounts to 1.7ml/kg/min of BM uptake of oxygen. The pigs on the other hand, received only 3 doses of a total of 400 ml of solution of 60% foam containing only ~15% of BM uptake of oxygen. This is a bit confusing to me. What was the rationale of only giving such a small volume in of oxygen? Why not try to give 40-50% of metabolic rate consumption. I do not understand the rationale of why you would choose such a low saturation target that would have been very uncertain of the outcome if you deliver higher oxygen volumes. If more experiments were done with higher volumes they need to be described and or discussed in relation to limitations of side effects.

Thank you for this interesting and important series of questions. We have reported all of the experiments performed in this study; we did not test higher volumes of oxygen administration. While we acknowledge that we may have been able to achieve a higher arterial oxygen saturation with a greater volume of treatment, we chose the dose we used for several reasons.

The first reason is the context of its use in the setting of profound hypoxemia and cardiac arrest. Under conditions of severe limitation of oxygen delivery, it is known that oxygen consumption significantly diminishes to small fractions of a healthy baseline.[13-16] Thus, *we hypothesized a priori that provision of a small fraction of baseline oxygen consumption would be sufficient.* Our goal was not to normalize oxygen saturation but rather to preserve and restore the circulation and prevent the severe injury that results from profound hypoxemia compounded by a lack of circulation. In a clinical situation, this could be accomplished through the titration of IVO₂ to maintain a pulsatile circulation rather than to normalize saturations.

Secondly, providing more oxygen than absolutely required may have increased the risks of hyperoxia and oxygen-related toxicities. Particularly in the setting of tissue dysoxia, the provision of oxygen always raises the possibility of pathologic oxygen utilization and the formation of oxygen

radicals. In concept, the optimal dosing of oxygen would be just sufficient to relieve the 'oxygen debt' and maintain cellular architecture, but not more. In intensive care situations, this may be accomplished by either the provision of small, incremental bolus doses of predetermined volumes (as we performed in this experiment), or by titrating in just enough oxygen to maintain the circulation; as we demonstrated here, this is possible at very low oxygen saturations and fractions of baseline VO_2 .

The third reason for our choice of dosing was the potential downside of volume administration. In the absence of hypovolemia as a cause of cardiac arrest, the provision of volume may be physiologically detrimental during resuscitation (e.g. by raising venous pressure and therefore decreasing coronary perfusion pressure) or following resuscitation from cardiac arrest (e.g. by increasing pulmonary edema by increasing left atrial pressure). Therefore, we favored using the lowest possible dose that we hypothesized would be effective (based on points 1-2 above).

The final reason was practical. As intensive care clinicians, the administration of 400 mL of material over a 6-minute time span via hand injection is quite reasonable but administering 2 or 3 or 4 times that volume as part of a resuscitation may be more challenging. Thus, demonstrating the effects of this level of dosing was important.

5. Experimental Design Suggestions:

*** An experiment with the therapy initiated after 12 minutes of asphyxia could provide insights into its effectiveness in more realistic emergency scenarios.**

Thank you for this suggestion. We apologize for our lack of clarity on the animal model, which was intended as a model of in-hospital cardiac arrest (IHCA), which occurs in more than 200,000 patients per year in the US.[2] IHCA has a survival rate between 25-30%, and neurologic injury is common in survivors.[3-5] In both children and adults, IHCA is most commonly caused by respiratory failure and hypoxemia, which is also an independent risk factor for mortality.⁶ Survival from IHCA diminishes with each passing minute, and therefore achieving early return of spontaneous circulation (ROSC) is the target of all resuscitative efforts.

Patients suffering from IHCA are by definition hospitalized, usually have an in situ intravenous line, and are commonly in an intensive care, heavily monitored environment. Administration of resuscitative drugs (e.g. epinephrine) within the first few minutes of a resuscitation is expected, so much so that timing to first dose represents a metric of resuscitation quality. For example, in a Get With the Guidelines registry analysis of 25,095 adults from 570 hospitals with IHCA and non-shockable initial rhythm, the median time to administration of the first dose of epinephrine was 3 minutes (IQR 1-5 minutes); 82% of patients received the first dose of epinephrine within 6 minutes, more than 50% of patients within 3 minutes.[7]

We envision that intravenous oxygen could be co-administered along with epinephrine as a first-line resuscitative drug. As such, the model we describe here represents a very realistic emergency use of IVO₂. The value of the experiment that we describe was in part to demonstrate that administration of the drug early on in the resuscitation enhances return of the patient's native circulation and provides 'rescue' volumes of oxygen to the tissues, preventing downstream sequelae. While it will be additive to understand the effect of the drug when administered at later time points within a resuscitation, the timing we described here is very achievable in the vast majority of patients with IHCA.

Of note, in this series of PEA/asystole patients with IHCA, the incidence of successful resuscitation was only 49%, survival to 24 hours 27%, and survival to hospital discharge 10%. Thus, the clinical outcomes in IHCA are nearly as poor as the control group in our swine study.

To better explain the rationale for our model, we have placed the following text at the beginning of the introduction (page 1):

Hypoxemia, or low blood oxygen content, can occur in the setting of lung disease or airway obstruction, which causes blood to circulate through the lungs and then to the body without being fully reoxygenated. In hospitalized patients, severe, episodic hypoxemia can result from endotracheal tube occlusion (e.g. secretions), progressive lung injury, ventilation-perfusion mismatch, and a number of other causes. Such episodes may be addressed with airway clearance, lung recruitment, increased ventilatory support, and when needed inhaled nitric oxide or extracorporeal membrane oxygenation. However, sometimes hypoxemia is temporarily so severe and refractory to these maneuvers that myocardial contractility fails, resulting in pulseless electrical activity or even asystole; this clinically manifests as cardiac arrest. Of note, 15-40% of in-hospital cardiac arrests (IHCA) are caused by respiratory insufficiency.^[1] Such patients exhibit a hypoxic insult compounded by an ischemic insult (i.e. cessation of cardiac activity and blood flow), resulting in severe ischemic injury to the brain, kidneys and other organs. Survival to hospital discharge following IHCA approximates 20%, with ~1 in 3 patients suffering from significant neurologic impairment.^[2] It is well recognized that survival following IHCA is enhanced by maneuvers that optimize oxygen delivery during IHCA (such as high quality CPR)^[3], achieving early return of spontaneous circulation^[4], and rapid cannulation to ECMO during CPR^[5]. Successful resuscitation from IHCA requires the rapid identification and reversal of the underlying cause^[6], however, reversing refractory hypoxemia remains an unmet challenge.

*** Exploring the effects of delivering higher oxygen volumes and discussing potential side effects could provide a more comprehensive understanding of the therapy's limitations and capabilities.**

Thank you for this important comment. We agree with this statement and did inject larger volumes of oxygen as part of our safety assessment (described in **Fig. 3**). Specifically, this study interrogated the most important and likely adverse effect of PMBs - the occurrence of vascular obstruction with increasing infusion rates. The fact that administration of 3 times the injection rate used in swine did not significantly increase the visualization of intact bubbles in the left heart demonstrates that PMBs confer a degree of tolerance above what may be required for efficacy. Certainly, a comprehensive study of the tolerability of PMBs will be required as part of regulatory approval but are outside the scope of this report.

*** Since most of the patients that would be treated by someone who has the therapy available would have professional EMS CPR and the earliest it could be delivered is after an IV line is applied, 12 minutes before therapy is the best case scenario for CPR model.**

Thank you for this comment. It is true that in the *out-of-hospital* setting ~50% of patients receive the first dose of epinephrine within 10 minutes of EMS arrival.^[17,18] However, we envision that this intervention would be performed on (and our model was based on) *hospitalized* patients with cardiac arrest (i.e. in-hospital cardiac arrest, IHCA), many of which have a respiratory etiology. In this setting, the median time to administration of the first dose of an injectable drug is 3 minutes from the time of cardiac arrest, with 82% of patients receiving the first dose of an injectable drug within 6 minutes of cardiac arrest.^[7] The outcomes of these patients are quite poor, with a significant contribution from precipitating hypoxemia. Thus, we envision that this drug will represent an important complement to the current armamentarium of resuscitative tools in IHCA. We have clarified this in the following sentences (page 6):

In clinical practice, we envision that IVO₂ would be available on-demand in environments caring for critically ill patients at risk for hypoxemia. Because arterial oxygen saturation is often continuously monitored by photoplethysmography in patients at risk for hypoxemia, IVO₂ could become a new treatment for refractory hypoxemia that is refractory to current standards of care, including airway clearance (e.g. suctioning), lung recruitment (e.g. hand ventilation) and the use of other critical maneuvers. Its dosing could be titrated in the same way that pre-arrest bolus doses of epinephrine are titrated to treat refractory hypotension,^[57] since both blood pressure and arterial oxygen saturation (by plethysmography) are routinely measured as vital signs in hospitalized patients. As in the swine model we described, patients being treated for in-hospital cardiac arrest (IHCA) known or presumed to be caused by hypoxemia could also be treated with IVO₂ as an emergency treatment; because such patients standardly receive IV resuscitative treatments within 3-5 minutes^[58], IVO₂ may restore early spontaneous circulation and significantly improve outcomes in such patients.

6. Practical Application Concerns: The immediate availability of the therapy within 4-5 minutes of an asphyxia event seems unrealistic for most situations, limiting its practical use.

As mentioned above, we envision that this drug will primarily be used to prevent or reverse asphyxial cardiac arrest in hospitalized patients. In this setting, the median time to administration of the first dose of an injectable drug is 3 minutes from the time of cardiac arrest, with 82% of patients receiving the first dose of an injectable drug within 6 minutes of cardiac arrest.[7] Thus, we believe that the use of this therapy as tested will be quite feasible, and the timing we described here is very achievable in the vast majority of patients with IHCA.

However, we recognize the Reviewer's concern and agree that it will be valuable to examine the efficacy of the drug when administered at later points within a resuscitation, especially for patients with out-of-hospital CA. Nonetheless, we believe this is beyond the scope of the current report.

Addressing these issues would significantly enhance the understanding and potential application of this innovative therapy. The paper holds promise, particularly in pediatric care, due to its ability to meet oxygen requirements with smaller infused volumes. However, a more detailed and critical analysis is necessary for a comprehensive evaluation.

Thank you for this critical appraisal and suggestion. As above, we hope that we have provided sufficient context to clarify the intended use of this drug in the Discussion section.

Thank you again for a thorough, thoughtful, and helpful review, your suggestions and comments have been very helpful in improving this manuscript.

Reviewer #2 (Report for the authors (Required)):

This is a very interesting study presenting a novel oxygen carrier which can deliver oxygen intravenously. The O₂ carrier shell immediately dissolves when given intravenously due a change to physiological pH. The study describes the design and manufacture of this product, as well as providing data on its use in a swine model of asphyxia/CPR; in addition, as studies on the safety profile of this therapy is provided. The authors are to be congratulated on a very nice detailed study. The in vivo results are quite impressive with major changes in very clinically important endpoints in the active treatment group.

GENERAL COMMENTS

I find the study quite interesting – the approach is novel, and the results are interesting and impressive. My only major concern is that the animal model is not very clinically relevant, although it is ideal from a proof of concept perspective.

Thank you so much for this important comment, and we apologize that we did not explicitly explain the clinical relevance of the animal model. This swine model was one of in-hospital cardiac arrest (IHCA), which occurs in more than 200,000 patients per year in the US alone.[2] IHCA has a survival rate between 25-30%, and neurologic injury is common in survivors.[3-5] In both children and adults, IHCA is most commonly caused by respiratory failure and hypoxemia, which leads to myocardial and endothelial failure, and subsequent cardiac arrest.[6] Survival from IHCA diminishes with each passing minute, and therefore achieving early return of spontaneous circulation (ROSC) is the target of all resuscitative efforts.

In this setting, intravenous administration of resuscitative drugs (e.g. epinephrine) within the first few minutes of a resuscitation is expected. For example, in a Get With the Guidelines registry analysis of 25,095 adults from 570 hospitals with IHCA and non-shockable initial rhythm, the median time to administration of the first dose of epinephrine was 3 minutes (IQR 1-5 minutes); 82% of patients received the first dose of epinephrine within 6 minutes, more than 50% of patients within 3 minutes.[7] We envision that, as described in this model, intravenous oxygen could be co-administered along with epinephrine (or even alone as a measure to reverse hypoxemia and prevent cardiac arrest) as a first-line resuscitative drug. Thus, the model we describe here represents the precise emergency use of IVO₂ that we anticipate.

In part, this manuscript serves to describe (somewhat surprisingly) that administration of a very small volume of oxygen (relative to resting oxygen consumption) early on in the resuscitation enhances return of the patient's native circulation and provides 'rescue' volumes of oxygen to the tissues, preventing downstream sequelae.

It's not clear how much O₂ is able to be delivered by how much fluid. The authors state that 400 ml of oPMB contains ~140 ml of O₂; this is pretty clear but the authors state that this is equivalent ~15% of resting O₂ consumption. The O₂ consumption of 10 kg pigs would be ~50 ml/minute, so I don't understand how this works out. So how was the 15% estimated?

Thank you for this important comment. We have calculated as follows:

- The average resting oxygen consumption (VO₂) of swine in the experimental group was 73.2 mL/min.
- We administered 140 mL of oxygen over 12 minutes of asphyxia = ~11.6 mL/minute

- 11.6 mL O₂/min provided/73.2 mL O₂/min resting VO₂ = 15.8% of resting VO₂

Parenthetically, we feel that 15% of resting VO₂ is a conservative estimate in the sense that oxygen was administered only during the last 6 minutes, such that it can also be stated that we administered ~30% of resting VO₂ during the intervention period. Additionally, it is known that in settings of severe limitations to oxygen delivery (with cardiac arrest being the most extreme example of this) that VO₂ significantly diminishes, such that the proportion oxygen provided to oxygen being consumed is likely significantly higher in the setting of IHCA. To enhance the clarity of this statement, we have revised the manuscript as follows (page 4):

At minutes 6, 8 and 10 of injury, swine were randomized to receive either IVO₂ (combined total 400 ml of 35% vol O₂/vol foam oPMBs, containing ~140 ml oxygen, n=8) or an equal volume of oxygenated D10 (n=10). Given that the mean measured resting oxygen consumption (VO₂) during the baseline period in this experiment was 73.2 mL/min, this volume represents the provision of ~30% of resting oxygen consumption for the last 6 minutes of asphyxia.

The therapy requires a substantial volume of fluid to deliver a minimal amount of oxygen. For example, (if I've done the calculation correctly) the authors used a volume of fluid that would be equivalent to about 4 liters in a 70 kg human, but yet only were able to deliver less than 50% of the (normal) O₂ requirements for just a single minute.

Thank you for these comments and caveats. A few points in response:

- If the estimated resting oxygen consumption (VO₂_{resting}) of a 70 kg adult is approximately 200 mL/min, then provision of ~15% of this volume (30 mL/min) of the 35% vol gas/vol foam emulsion for 12 minutes would require:

$$\begin{aligned} & \frac{12 \text{ mins} \times 30 \text{ mL gas/min}}{0.35 \text{ mL} \frac{\text{gas}}{\text{mL}} \text{ foam}} \\ & = 1,028 \text{ mL foam} \times 0.65 \text{ mL fluid/mL foam} \\ & = 668 \text{ mL fluid phase} \end{aligned}$$

Note that because the gas phase of the foam is rapidly consumed (almost completely prior to reaching the left ventricle), the volume of injected foam contracts down to that of the fluid phase.

- Further, the gas concentration of the foam used in our swine study was not maximized. This was to facilitate the manufacture of the volumes required. However, in the safety study in rats (**Fig. 3c-f**), the gas concentration of PMBs was 50% vol gas/vol foam. In this case, the calculation above would be:

$$\begin{aligned} & \frac{12 \text{ mins} \times 30 \text{ mL gas/min}}{0.5 \text{ mL} \frac{\text{gas}}{\text{mL}} \text{ foam}} \\ & = 720 \text{ mL foam} \times 0.5 \text{ mL fluid/mL foam} \\ & = 360 \text{ mL fluid phase} \end{aligned}$$

- It is likely that this concentration could be maximized even further by optimizing the packing density of the foam, likely to the high 60% vol gas/vol foam range.
- One important aspect of our work (and surprising to us) was the demonstration that the provision of small fractions of resting VO₂ as an injection is sufficient to resuscitate the circulation, to maintain cellular integrity, and to prevent severe end organ injury. Thus, while the provision of larger fractions of resting VO₂ is intrinsically appealing out of a desire to restore normoxia to the extent possible, this may be in fact unnecessary (or even harmful due to oxygen-related toxicities). Instead, administration of the drug to a degree

sufficient to maintain the circulation would be a way to minimize volume of fluid administration and its sequelae of venous hypertension and congestion, and to also minimize oxygen-related toxicities.

This would be hugely problematic in the clinical setting. It would only work in situations when the clinician knows exactly when there is a huge oxygen deficit in blood. In the model they use of asphyxia with a closed airway, they know exactly when to give the IV O₂. Most patients with cardiac arrest would have an open airway or partially open airway, and would suffer more from ischemia than hypoxemia. In the discussion, I think the authors should discuss these potential limitations in the clinical application of this approach.

Thank you for these important points. As highlighted above, we undertook what we view as a clinically realistic model of an in-hospital cardiac arrest (IHCA) that is caused by hypoxemia. Although 50-60% of IHCA is indeed cardiac in origin (often coronary ischemia), 15-40% are caused by respiratory insufficiency, primarily hypoxemia.[6] Hypoxemia as a precipitating etiology is even more common in infants and children.[19] In these patients, although the airway is open, hypoxemia is typically due to intrapulmonary shunting, which is most commonly due to deficits in oxygen diffusion or ventilation-perfusion mismatch. Your points regarding the timing of injectable oxygen in clinical practice are excellent. We have added the following text to the Discussion (page 6-7):

In clinical practice, we envision that IVO₂ would be available on-demand in environments caring for critically ill patients at risk for hypoxemia. Because arterial oxygen saturation is often continuously monitored by photoplethysmography in patients at risk for hypoxemia, IVO₂ could become a new treatment for refractory hypoxemia that is refractory to current standards of care, including airway clearance (e.g. suctioning), lung recruitment (e.g. hand ventilation) and the use of other critical maneuvers. Its dosing could be titrated in the same way that pre-arrest bolus doses of epinephrine are titrated to treat refractory hypotension,^[57] since both blood pressure and arterial oxygen saturation (by plethysmography) are routinely measured as vital signs in hospitalized patients. As in the swine model we described, patients being treated for in-hospital cardiac arrest (IHCA) known or presumed to be caused by hypoxemia could also be treated with IVO₂ as an emergency treatment; because such patients standardly receive IV resuscitative treatments within 3-5 minutes ^[58], IVO₂ may restore early spontaneous circulation and significantly improve outcomes in such patients.

The major limitation of this technology is the volume of the fluid which must be administered to deliver the oxygen payload. In this work, the gas fraction of the foam ranged between 35-50%, which can likely be optimized further as it has with similar products.^[15,59] However, given the high rate of resting oxygen consumption (e.g. a healthy adult consumes ~200 mL/minute), oxygen provision using this technology is unlikely to provide more than several hundred milliliters of oxygen, particularly in critically ill patients with heart disease in whom even small quantities of administered volume administration may aggravate venous hypertension and pulmonary edema. Thus, IVO₂ should be considered as a strategy for the IV bolus administration of oxygen in critical situations, to reverse a critical physiology (such as a cardiac arrest or a pulmonary hypertensive crisis) or to bridge such patients to definitive treatment (such as extracorporeal membrane oxygenation support). The oxygen deficit that accrues over even a few critical minutes can be the difference between a favorable

outcome and manifesting of neurologic injury for a lifetime, or be the difference between survival and death.^[5]

SPECIFIC COMMENTS

1. Pg 1, para 1:

- a. The authors state that respiratory disorders disrupt O2 diffusion in lungs leading to low O2 saturation. Problems with O2 diffusion are not the major factors leading to hypoxemia; issues with ventilation-perfusion inhomogeneity are much more common.**
- b. The authors state that hypoxemic respiratory failure was a leading cause of death during COVID-19. Interestingly, most deaths due to ARDS are ascribed to multiple organ failure, not hypoxemia, as implied by the authors.**
- c. Hypoxemia is clearly important in the situations described in this paragraph, it is likely the ischemia (which then leads to tissue hypoxia) is likely more important than the hypoxemia per se.**

Thank you for these important points. We have heavily revised the introductory paragraph to address these points, and also to better align with the intended use of the drug. The revised text is copied below for your convenience (page 1):

Hypoxemia, or low blood oxygen content, can occur in the setting of lung disease or airway obstruction, which causes blood to circulate through the lungs and then to the body without being fully reoxygenated. In hospitalized patients, severe, episodic hypoxemia can result from endotracheal tube occlusion (e.g. secretions), progressive lung injury, ventilation-perfusion mismatch, and a number of other causes. Such episodes may be addressed with airway clearance, lung recruitment, increased ventilatory support, and when needed inhaled nitric oxide or extracorporeal membrane oxygenation. However, sometimes hypoxemia is temporarily so severe and refractory to these maneuvers that myocardial contractility fails, resulting in pulseless electrical activity or even asystole; this clinically manifests as cardiac arrest. Of note, 15-40% of in-hospital cardiac arrests (IHCA) are caused by respiratory insufficiency.^[1] Such patients exhibit a hypoxic insult compounded by an ischemic insult (i.e. cessation of cardiac activity and blood flow), resulting in severe ischemic injury to the brain, kidneys and other organs. Survival to hospital discharge following IHCA approximates 20%, with ~1 in 3 patients suffering from significant neurologic impairment.^[2] It is well recognized that survival following IHCA is enhanced by maneuvers that optimize oxygen delivery during IHCA (such as high quality CPR)^[3], achieving early return of spontaneous circulation^[4], and rapid cannulation to ECMO during CPR^[5]. Successful resuscitation from IHCA requires the rapid identification and reversal of the underlying cause^[6], however, reversing refractory hypoxemia remains an unmet challenge.

2. I understand that some aspects of the protocol were blinded (e.g., neuroradiological assessment), but it's not clear if all aspects of the study were blinded. Can this therapy be given in a blinded fashion?

We were unable to blind members of the research team who were performing the resuscitation to treatment group primarily because the response to treatment was quite obvious during the resuscitation; swine treated with injectable oxygen had almost immediate improvements in hemodynamics and oxygen saturation while the drug was being injected, while those in the control

group did not. The injectable drug was also a white substance, while control group was a clear crystalloid. Although we could have wrapped the syringes and tubing in non-transparent tape, this turned out to be practically challenging in the experiment setting. However, we felt that the integrity of the experiment was ensured by actively measuring the quality of CPR (which was not different between groups), and by strictly following a treatment protocol throughout all phases of the experiment (including during the resuscitation and survival periods). We have listed this as a limitation in the Discussion (page 7):

From an experimental perspective, another important limitation to note is that some members of the research team in the cardiac arrest experiment could not be blinded to treatment group due to the acute and rapid significant differences in physiologic response of the patients to the treatment. An additional practical challenge for blinding is that the PMB solution has a distinct white/opaque color that is easily identifiable. Although we could have taped the all the syringes and lines, it was challenging to completely mask the drug, particularly when priming the injection ports or changing the syringes during treatment, since the drippings from the syringes or residues at injection port were immediately visible. However, this limitation was balanced by actively measuring the quality of CPR (which was not different between groups), and by strictly following a treatment protocol throughout all phases of the experiment (including during the resuscitation and survival periods). Finally, the number of animals in the safety study was somewhat low, such that we were underpowered to detect minor differences between groups.

3. Pg 4, para 1:

a. I think that a pulmonary hemorrhage model would be different in two ways: (1) it usually would not lead to complete obstruction of all airways/alveolar regions unless extremely severe, and (2) if it was so severe as to cause massive asphyxia, it would not be quickly reversible, which is a requirement for this approach to work since only a small volume of O₂ can be given because of fluid overload.

Thank you for these important points. We have removed the reference to pulmonary hemorrhage.

b. Was the 400 ml of fluid given 3 times, or was it the total volume? As well the figure states 21 ml/kg, which would not be exactly 400 ml in all animals.

We apologize for the lack of clarity on this issue. The 400 mL of foam volume was the total volume administered. We have revised the text for clarity as follows (page 4):

At minutes 6, 8 and 10 of injury, swine were randomized to receive either IVO₂ (combined total 400 ml of 35% vol O₂/vol foam oPMBs, containing ~140 ml oxygen, n=8) or an equal volume of oxygenated D10 (n=10). Given that the mean measured resting oxygen consumption (VO₂) during the baseline period in this experiment was 73.2 mL/min, this volume represents the provision of ~30% of resting oxygen consumption for the last 6 minutes of asphyxia.

We note here that we have revised the calculation from 15% to 30% of resting oxygen consumption provided and clarified the timeframe as described in the calculation listed above.

c. "At minutes 6, 8 and 10 of apnea, swine were randomized to receive IVO2 (total 400 ml of 60% oPMBs, containing ~140 ml oxygen, which represents ~15 % of resting oxygen consumption." It's not clear to me how much oxygen is available per liter of fluid. As well, it's not clear to me what period you are referring to when you say this is equivalent to 15% of resting oxygen consumption.

Thank you for this important point. We have clarified as above.

4. Pg 4, para 2:

a. Was PaO₂ measured and pH measured. Would be helpful to have PO₂ data presented.

Thank you for this comment. We have added the following results of arterial oxygen tension (PaO₂) into the Supplementary Materials (**fig S23**).

Fig. S23. The arterial partial pressure of oxygen (PaO₂) was significantly higher at 7, 9, 11 minutes in the IVO2-treated animals than those in the control group. Data presented as mean ± SD; red = control, blue = oPMB-treated swine. Statistical analysis by a two-way ANOVA with Tukey's multiple comparisons post-test.

We have left this as a supplementary figure since, under conditions of severe hypoxemia the arterial oxyhemoglobin saturation (SaO₂) is a more clinically determinant parameter to follow given that it reflects arterial oxygen content much more so than PaO₂. Trends in PaO₂ may be misleading due to differences in oxygen dissociation related to acidosis and other clinical factors.[1]

Arterial pH was originally included as **fig. S20A**, now updated as **fig S29A**. No significant differences in blood pH were observed between treated and control groups.

b. The authors ascribe the increase in PaCO₂ after giving IV O₂ to preserved cellular metabolism. It may also be due to the Bohr (or maybe it's the Haldane) effect.

Thank you for this important consideration. It is true that the Haldane effect describes the increased capacity of hemoglobin to bind CO₂ (creating carbaminohemoglobin) under conditions of decreased oxyhemoglobin saturation (SO₂). Thus, it is true that in the setting of a lower SO₂,

hemoglobin exhibits a greater affinity for CO₂, lowering plasma CO₂ (PaCO₂). However, carbaminohemoglobin only accounts for ~5% of transported CO₂, making it unlikely that the small difference in SO₂ would account for this difference in PaCO₂. We have added the following caveat to the Results section (page 4):

Arterial carbon dioxide tension was higher at 7, 9, 11, and 13 minutes in IVO₂-treated swine (P<0.05, Fig. 4c), presumably due to preserved cellular metabolism and CO₂ production (though this may be partially explained by the Haldane effect as well).

c. Figure 3d: Trivial point, but I think it would be better to present these data with percent of animals receiving CPR on the Y-axis (as opposed to those NOT receiving CPR)

Thank you for this point. We have re-labeled the Y axis of this graph (now as Fig 4d) as ‘Freedom from cardiac arrest’. We find this graphical format to reflect the fraction of patients with a preserved or intact spontaneous circulation, which is what we intend to reflect.

d. Figure 3g: Please present earlier time points as well

Thank you for this important point. Per the Reviewer’s request, we have updated this figure (now updated as Fig. 4g) to include earlier time points and have re-analyzed the data using a 2-way repeated measures ANOVA with Tukey’s multiple comparison’s test. Figure 4g is depicted below for your reference.

Fig 4g. IVO₂ treatment improved mean arterial blood pressure (MABP) during resuscitation. Data presented as mean± SEM, comparison between groups by two-way ANOVA, Sidak’s multiple comparison with only significant p values shown. For all figures, measurements are biological replicates, data are means, error = SD; red = control, blue = IVO₂ treatment.

5. Pg 4, para 3: Presumably the results for Figure 3i (and subsequent panes in Figure 3) were obtained only in animals that survived. As such, these data are even more impressive than they appear since the sickest animals (that died) are not included. I know this is obvious, but it's worth pointing out to the reader.

Thank you for this comment, and yes, it is correct that Fig. 3i-s (now as **Fig 4i-s**) were collected only in surviving swine. We have added the following statement to the Discussion (page 5):

Even amongst only surviving swine, Swine Neurologic Deficit Scores were significantly lower at post-injury days 1-3 in IVO2 treated swine (**Fig. 4i**).

We have also added the following to the caption of what is now **Fig. 4**.

Note that **Fig. 4i-s** reflects data collected only in surviving swine, which omits 7 of the 10 swine in the control group who did not survive.

6. Safety Study: The safety profile looks very reasonable but the numbers are somewhat small and there are a few data points that look like they might be relevant (e.g., lower weight gain at 7 days, clotting time at 4 days, maximum lysis at days). This is not a major issue; just might be worth mentioning the relatively small numbers.

Thank you for this important point. We have added this as a limitation to the Discussion (page 7):

Finally, the number of animals in the safety study was low, such that we were underpowered to detect minor differences between groups.

7. Does blood osmolarity change after therapy, and if so by how much?

We thank the Reviewer for this comment. We performed osmolarity measurement on collected serum samples at baseline, 30 minutes, and 4 hours post-arrest. The results are shown in a new **fig. S30**.

Fig. S30. Serum osmolarity increased compared with baseline at 30 minutes equally in control and IVO2-treated groups; this is expected since a significant volume of hypertonic D10 was infused in both groups. At 4-hour post asphyxia, the blood

osmolarity returned to normal in both groups. Data presented as mean \pm SD. Statistical analysis by a two-way ANOVA test.

8. Pg 5, last para:

a. Replace "...expected response to hypoxia..." TO "...expected response to very severe hypoxemia ..."

Thank you. This change has been made.

b. It's not clear which substrate you are referring to.

Thank you. We have clarified the statement as follows (page 6):

(2) provision of oxygen for energy generation restores systemic vascular resistance (both arteriolar and venous) and myocardial function, raising perfusion pressure and blood flow;

c. Providing residua of the polymer shell is useful when the "patient" is volume deplete, but it may be problematic in many situations when that is not the case.

We agree with this consideration. The context of that statement was one of resuscitation from cardiac arrest. We have added the following caveat as a limitation in the Discussion (page 6):

However, given the high rate of resting oxygen consumption (e.g. a healthy adult consumes ~200 mL/minute), oxygen provision using this technology is unlikely to provide more than several hundred milliliters of oxygen, particularly in critically ill patients with heart disease in whom even small quantities of administered volume administration may aggravate venous hypertension and pulmonary edema.

9. Pg 6, para 1:

a. Diffusion barrier: As discussed above this is not a major issue in most cases of hypoxemia due to respiratory disease.

Thank you for this important consideration. It is true that respiratory disease is heterogenous, and the relative contributions of diffusion limitation and ventilation-perfusion (V/Q) inequality varies based on etiology.[20] However, the root issue even in V/Q inequality is venoarterial shunting, a pathology in which venous oxygenation would ameliorate, in much the same way that venovenous (VV) ECMO effectively restores normal oxygenation. We have edited this sentence as follow to reflect this important consideration (page 6):

Further, a given dose of oxygen may also have a more pronounced effect when used in clinical settings of lung injury. Normally, oxygen flows from the alveolus into the blood, but in this model of airway obstruction, oxygen tension of the pulmonary artery exceeded that of the alveolus, such that oxygen initially diffused backwards, equilibrating with the functional residual capacity of the lung (i.e., increasing the volume of distribution of the gas payload); in this sense, this was an exaggerated model of ventilation-perfusion inequality, a central pathology in patients with clinical lung disease. We expect that this phenomenon of back-diffusion would be attenuated, and therefore the dose response more pronounced, in patients with more heterogenous ventilation-perfusion inequality or with an oxygen diffusion gradient.

b. In this model, lung volume likely decreases substantially during airway occlusion.

Thank you for this important point as well. We agree that lung volume may have decreased in this model, both due to airway occlusion and due to external thoracic compressions. We believe that the rephrasing of the statement above is still accurate given this consideration.

10. Discussion:

a. As described above, a more robust discussion of the potential problems in using this approach would be helpful

Thank you for this suggestion. As above, we have added the following comments regarding potential limitations of this approach to the Discussion (page 6).

The major limitation of this technology is the volume of the fluid which must be administered to deliver the oxygen payload. In this work, the gas fraction of the foam ranged between 35-50%, which can likely be optimized further as it has with similar products.^[15,59] However, given the high rate of resting oxygen consumption (e.g. a healthy adult consumes ~200 mL/minute), oxygen provision using this technology is unlikely to provide more than several hundred milliliters of oxygen, particularly in critically ill patients with heart disease in whom even small quantities of administered volume administration may aggravate venous hypertension and pulmonary edema. Thus, IVO₂ should be considered as a strategy for the IV bolus administration of oxygen in critical situations, to reverse a critical physiology (such as a cardiac arrest or a pulmonary hypertensive crisis) or to bridge such patients to definitive treatment (such as extracorporeal membrane oxygenation support). The oxygen deficit that accrues over even a few critical minutes can be the difference between a favorable outcome and manifesting of neurologic injury for a lifetime, or be the difference between survival and death.^[5]

b. What is the theoretical maximal concentration of O₂ to the required fluid using this approach?

Thank you for this important question. As above, the fraction of gas to the volume of foam (i.e. volume percent) of the foams tested here was 35-50%. In the past, we concentrated lipid-based microbubbles to as high as 90% vol/vol, but as shown in the figure to the right (from ref 21) the viscosity of such foams was like that of whip cream and mixing following injection was poor.^[21] We believe that the most realistic maximal volume fraction of IVO₂ would be 60-70% vol/vol. For this reason, this technology – at least in its current state – will be useful for acute resuscitations and not for prolonged oxygen supplementation. We believe that the most realistic maximal volume fraction of IVO₂ would be 60-70% vol/vol. However, to achieve this higher fraction in PMBs, further

optimizations are needed through manipulating size distributions during fabrication to further maximize their packing density. For this reason, this technology – at least in its current state – will be useful for acute resuscitations and not for prolonged oxygen supplementation.

Thank you again for a thorough, thoughtful, and helpful review of this manuscript, your comments and suggestions have made this manuscript significantly better.

Reviewer #3 (Report for the authors (Required)):

SUMMARY:

The authors describe a LmD polymeric microbubble (PMB) gas carrier that can be used to deliver oxygen intravenously. This PMB dissolves at physiologic pH into excretable molecular constituents, which allows for rapid delivery of oxygen while avoiding apparent toxicity and vascular obstruction, unlike other gas carriers. The authors use a rat model to demonstrate that there is no evidence of adverse hemodynamic effects of vascular obstruction following PMB administration. In a swine model of acute hypoxemic respiratory failure and subsequent cardiac arrest, PMB administration improved SaO₂, time to ROSC, survival, and neurologic outcomes as compared with control. Additionally, safety studies performed in rats showed renal and hepatic clearance (though with low levels of LmD found in the spleen, liver, and kidneys after 7 days), and no laboratory evidence of end organ, metabolic, or hematologic dysfunction.

This is a compelling area of research with a clear unmet need. The data presented will be of interest to a broad readership. There are several questions/points that would support strengthening the current manuscript and these have been delineated below.

We thank the Reviewer for the positive assessment.

Major:

1. How stable are the PMBs following oxygenation? How quickly does oxygen diffuse through the polymer shells and how soon after oxygenation does it have to be injected?

We thank the Reviewer for these important material-related questions, including the ones from Comment 2 and 4. In response, we have performed additional experiments to demonstrate the pH- dependent dissolution kinetics and the long-term stability of PMBs. These new data are included as **Fig 2**, and **fig. S8-14**.

Dissolution behaviors

There are two critical aspects to the dissolution of the PMB shell: dissolution of the **gas core** and breakdown of the **shell** into its soluble components. The rapid dissolution of the gas core is essential to avoiding acute hemodynamic instability due to gas embolism and the dissolution of the shell is crucial to facilitate clearance. We have used several approaches to investigate this process as described in the new **Fig. 2**. The following text has been added to the main manuscript (page 3):

Dissolution of PMBs ex vivo

The intravenous administration of a gas requires the rapid dissolution of gas carriers to avoid vascular obstruction. Subsequently, the carrier materials must be biocompatible and rapidly cleared. As noted earlier, the LmD polymer itself exhibits pH-responsive behavior in solution due to the presence of carboxylic acid groups (pK_a ~ 4.8) and is molecularly soluble above pH 5 (**Fig. 2a, fig. S15**). In this work, the mechanism of dissolution of PMBs following injection hinges upon the pH-based deprotonation of carboxylic groups, which increases the solubility and hydration of polymers that compose the shell. This causes water influx in the shell, increasing surface tension and destabilizing the gas core, promoting its dissolution. The deprotonated shell simultaneously reverts to small and soluble components. Notably,

unlike lipid coated bubbles, which require a gas concentration gradient (i.e., sink) to dissolve^[24-26], PMBs dissolve at a physiologic pH within seconds even in the absence of a sink (**Supplementary Video 1**). To examine whether PMB shells fully dissolve and revert to soluble LmD constituents, we added PMBs into phosphate buffered saline (PBS) solution of varying pH under stirring and confirmed DLS size measurements 2 minutes following admixture (Earlier timepoint was not obtained by DLS due to the sampling limitation of the instrument). Between pH 9.0 and 5.0, PMB shells were all fully dissolved within 2 minutes, reverting to soluble polymers of similar sizes of LmD solutions prepared from solid states (**Fig. 2a**). Solubilized LmD polymer has a mean hydrodynamic radius less than 10 nm and an estimated MW ~12 kDa (determined by NMR in **fig. S2**), well below the MW cutoff for glomerular filtration (30 – 45 kDa)^[32,33]. In contrast, dissolution of previous IFNP MBs, which consisted of hydrophobic polymers of higher MW (>60 kDa), revert into large and insoluble nanoparticles (>100 nm) that visibly precipitate over time.

To better investigate the pH-dependent dissolution kinetics, we examined PMBs To better investigate the pH-dependent dissolution kinetics, we examined PMBs mixed with PBS while continuously applying ultrasound. Like various polymeric shelled microbubbles^[34,35 41,42], the gas core of PMBs creates acoustic backscatter and produces contrast in proportion to the presence of gas bubbles within the field of view. The decrement in contrast intensity (i.e., bubble dissolution) was shown to be pH-dependent: at pH 9.0, 7.2, and 6.5, PMBs were no longer visible within 2 to 3 seconds, while dissolution of the gas core was prolonged at pH<6 (**Fig. 2b,c**). (To note, although the LmD shell is less soluble at pHs 4.8 and 3.8, we noticed the gas core of PMBs slowly becomes fluid-filled as shown by the slow decrease in echo intensity, this is likely because the salts in PBS affected the swelling of the hydrogel-like shell). While it is known that ultrasound may contribute to loss of MBs due to inertial cavitation, **Fig. 2b** suggests that pH is the dominant factor affecting dissolution rate. To further account for acoustic destruction, we performed the same experiments while applying ultrasound only at selected terminal time points, finding similar dissolution rates (**Fig. 2d,e**).

To further examine dissolution of the shell (separate from that of the gas core), we performed UV-vis spectroscopy. In this construct, it is expected that an increase in absorbance from baseline could be caused by either undissolved gas core or a large polymeric aggregate (i.e. undissolved shell or aggregated constituents). From pH 9.0 to 6.0, UV absorbance reached baseline within 2-3 seconds, similar to the kinetics in acoustic studies, indicating both that the gas core had dissolved, and the shell reverted to its soluble constituents in that time (**Fig. 2f,g**). Between pH 5.5 and 5.0, the return to baseline was much longer than in the acoustic study, suggesting that within this pH range the gas core dissolves first, and the shell required more time to revert to soluble polymers. These findings are consistent with the pH-triggered dissolution mechanism of PMBs that is essentially an acid-base reaction, the rate of which is proportional to the concentration of hydroxyl ions in the solution and limited by diffusion. This mechanism also explains some discrepancies of dissolution kinetics seen at lower pHs. For example, in contrast to Uv-vis, DLS showed PMBs were fully dissolved at 2 minutes at pH 5.5 and 5.0, this is likely due to lack of sufficient mixing in Uv-vis experiments. However, at pH>6.0, the dissolution kinetics measured from various methods were all in good agreement. Collectively, these results validate our proposed design for the new gas carrier and established that both the gas core and the shell of PMBs rapidly dissolve at pH levels (7.5 to 6.5) that are relevant to

intravenous injection, as the blood pH rarely drops below 6.5 even in extreme instances.

Additionally, **Fig. 2** is shown here for your convenience.

Fig. 2 | LmD PMBs rapidly dissolve at physiologic pH. (a) DLS measurement of size following mixing of LmD PMBs mixed in PBS solution for 2 minutes at varying pH. LmD PMBs fully dissolve above pH 5 and revert to their soluble components with a mean size < 10 nm, similar to those of LmD solutions prepared from solid states. In contrast, the previous generation of IFNP MBs (made from more hydrophobic polymers) led to formation of much larger nanoparticles. (b) Phantom sonography of aPMBs in aerated PBS shows the pH-dependent dissolution rate, evidenced by the disappearance of contrast intensity produced by the gas core under continuous ultrasound. In the absence of a gas sink, PMBs rapidly dissolved above pH 6 within seconds. Data presented as change in contrast/bright area from baseline. Data are means, error = SEM. Continuous measurements collected as biological replicates. (To note, although the LmD shell is less soluble at pHs 4.8 and 3.8, we noticed the gas core of PMBs slowly becomes fluid-filled as shown by the slow decrease in echo intensity, this is likely because the salts in PBS affected the swelling of the hydrogel-like shell). (c) Representative images from phantom sonography study that show the dissolution profile of aPMBs at different time points at various pHs. (d, e) To account for any destructive effect that ultrasound itself has on PMBs, the experiment was repeated while only applying ultrasound at the expected dissolution time from (b), showing similar dissolution times even absent the application of continuous ultrasound. Measurements collected as biological replicates. Data presented as change in area of contrast/brightness from baseline, error = SEM, analyzed by student's t-test. (f,g) Dissolution of the shell and gas core was then studied using UV-vis absorbance spectroscopy. Continuous measurements collected as biological replicates. Similar to the characterization using ultrasound (which detects only dissolution of the gas core), UV-vis returns to baseline within seconds at pH above 6, suggesting both that the gas core has dissolved and that the shell has broken down into its constituent components. At more acidic pH, return of UV-vis absorbance to baseline took 10 minutes or longer. Contrasting this with sonographic experiments (e.g. b) in which ultrasound scatter returned to baseline within 90-260 seconds, these findings suggest that

following dissolution of gas core, the remaining shell constituents take additional time to dissolve and revert to soluble components. Data presented as change from baseline, error = SEM.

Collectively, these characterizations demonstrate that the dissolution of PMBs is pH-dependent, and both the gas core and the shell rapidly dissolve within seconds at physiologically pH levels relevant to intravenous injection.

These in vitro results also validate the use of sonography as a tool to monitor real-time PMB dissolution kinetics in vivo which as in **Fig. 3**. As in **Fig. 3g-l**, echocardiography showed that oxygen- and air-filled PMBs rapidly dissolve before reaching the left heart in healthy animals (i.e. within seconds). In contrast, the lipid based LOMs persist in the entire circulation for minutes following injection, which contributes to the significant risk of embolism and hemodynamic instability.

[Long-term stability]

Previously, the longest stability data that we had was obtained after 3-month storage of PMB in D10 at room temperature, in which we did not observe change in foam volume or size. Longer-term studies were marred by bacterial contamination, a hard-to-avoid artifact when fabricating biomaterials in a non-GMP laboratory environment.

Thus, we addressed the Reviewers' question on the long-term stability of oxygen-filled PMBs (oPMBs) using an accelerated stability study at various temperatures over 30 days, monitoring changes in foam height and PMB size. In these studies, oPMBs were packaged either in capped syringes or plastic tubes and placed in temperature-controlled, oxygen purged incubators (pO₂ of the oPMB solution was maintained at 740 mmHg). These results are provided in new **fig. S8-14**, showing that PMBs were stable up to 45°C in glass syringes with no measurable change in foam volume or size distribution. Interestingly, there was some PMB loss at 45°C with plastic tubes, likely due to the surface-specific drying effects between plastic and the PMB shell may have compromised the shell integrity. Storage at 60°C exhibited rapid loss of PMBs (**fig. S13**), likely due to breakdown of the hydrogen bonds that stabilize the PMB shell. It is important to point out that PMBs lack the coalesce and ripening behavior seen with lipid-based oxygen microbubbles (LOMs) due to their solid shell structure. If any, the destabilization of PMBs mainly resulted from water-influx into gas core due to compromised shell structures, which led them to sediment and separate from the top foam. For instance, as shown in **fig. S8,9**, the remaining foam in the tube at 45°C did not change in size distribution, the sediment of fluid-filled PMBs at the bottom was dispersible upon shaking and can readily dissolve at physiological pHs. The following text has been added to the main manuscript (page 2-3) and also included relevant details in method section:

The long-term stability of oPMBs was evaluated via an accelerated stability test at various temperatures, as shown in **fig. S8-14**, oPMBs stored in glass syringes were stable up to 45°C up to 30 days without changes in foam volume or size distribution, while loss of oPMBs were observed at 60°C. Based on these results, we expect the shelf-life of oPMBs to be at least 6 months at room temperature and at least 1 year under refrigerated conditions.

Vis a vis the Reviewer's question regarding the diffusion rate of oxygenation, we did not measure the diffusion rate of oxygen through the polymeric shell per se, but practically, the equilibration of gases across similar polymeric thin-shelled microbubbles is almost instant [22], and thus oxygenation of a bulk PMBs solution is primarily limited by mass transfer. In our experience, purging 500 ml of concentration PMB solution for an hour with adequate shaking is sufficient to

allow full oxygenation. As shown in the newly included stability studies, the oPMBs can be stored for a long period of time before infusion. During our in vitro and in vivo studies, aPMBs and oPMBs were generally stored between 3 to 5 weeks prior to use.

2. Is there any hydrogen bonding expected between different PMBs at the concentrations used to inject in vivo, and would that have any effect on the rate of dissolution or likelihood of obstruction?

We thank the Reviewer for these important questions. Although the PMB shells itself is stabilized by hydrogen bonding, we have not observed any evidence of inter-particle hydrogen bonding: PMBs do not show evidence of aggregation or change in size distribution as would be expected if that were to occur, and they are fully dispersible. This was illustrated by optical microscopy (**Fig. 1**), where the densely packed PMBs did not form aggregated clusters and are uniformly distributed. Our newly included experiments in **Fig. 2** show that dissolution of PMBs is pH-dependent and very rapid above pH 6. As shown in our hemodynamic safety study in **Fig. 3**, PMBs of various concentrations (up to 80%) infused at various rates (up to 12 ml gas/kg/min) can all rapidly dissolve in blood and did not cause any obstruction.

3. Please comment on why spleen macrophage vacuole formation was observed in rodents but not swine.

This was indeed an interesting difference and one for which we do not have a satisfying explanation. It is possible that this finding was due to differences in metabolism between species. [23,24] Furthermore, in this study swine underwent severe injury while rodents were uninjured, such that there may have been an injury-related difference in immune response. It is also worth mentioning that macrophage vacuolation is commonly associated with many PEGylated biotherapeutics, which is generally considered as a reversible, adaptive, nonadverse finding [25,26]. Nonetheless, this phenomenon will certainly need to be further explored in a future study in anticipation of FDA filing.

4. Fig 2b claims that the “recovery of attenuated acoustic signals within 3 seconds illustrates the rapid dissolution and outgassing of PMBs in PBS buffer”. However, ultrasound can cause cavitation of microbubbles, which may affect the observed recovery rate. To support the claim that solution pH is responsible for the observed recovery, the authors should show the acoustic attenuation versus time when PMB added to solutions of various pH (e.g. pH 3-10).

We fully agree with and thank the Reviewer for this important comment. Per the Reviewer's suggestions, as discussed in detail in response to Comment 1 above, we performed acoustic study using clinical ultrasound to study the dissolution kinetics of PMBs at different between pH 3.8 to 9.0 and the results (**Fig. 2b,c**) confirmed that pH is the main factor to affect the dissolution rate of PMBs, and the dissolution time above pH6 was within seconds. To further rule out the potential effect of inertial cavitation by the continuous application of ultrasound, we compared the results between continuously imaging and intermittent imaging which only applied ultrasound at a terminal time points, the results (**Fig. 2d,e**) confirmed that the use of ultrasound did not affect the dissolution time of PMBs. We confirmed these findings with a UV-vis study as in **Fig. 2f,g**, which fully agreed with the acoustic study and showed that the PMBs can rapidly dissolve above pH 6 and revert to soluble components.

Again, we thank the Reviewer for this comment, and we believe the additional experiments have substantially improved the revised manuscript.

5. Would a femoral catheter be required for PMB administration? What are the theoretical limitations regarding route of administration, especially if this were to be given in a true emergent setting? What limitations might peripheral administration or smaller catheter diameter impose? Could it be given via IO catheter?

Thank you for these important questions. There is no requirement for a central venous catheter for injection. In fact, injection into a more distal vessel would permit more time within the venous circulation for PMB dissolution to occur prior to reaching the pulmonary circulation. In this experiment, PMBs were injected in a way that maximally interrogated their dissolution under unfavorable conditions – a high injection rate, containing air thus high fraction of inert gas (in hemodynamic safety experiments), low flow (in cardiac arrest experiment) and a central injection site.

Because the hydrogel shell of PMBs can be affected by extreme hydrostatic pressure, it is likely that the rate of injection through smaller IV catheters (e.g. 22 or 24 gauge) will be limiting, just as it is with IV contrast in CT scans. [27] However, this is also the case with other fluids, particularly in pediatrics. For example, the rate at which packed red blood cells can be pushed through a 24-gauge IV catheter in a bleeding patient is limited. However, given that we were able to inject PMBs at a rate sufficient for efficacy through a 3 French catheter (1 mm outer diameter) and with very little resistance, we believe that this consideration, while very important to consider, will not limit the use of the technology.

In the past, we have injected lipid-based microbubbles (LOMs) via intraosseous line and cadaveric bone, finding no difference in size distribution and similar efficacy following injection. [28,29] This will be important to further interrogate in future studies as we move towards clinical translation.

6. The way foam concentrations are referred to throughout the text is confusing. Please make the format consistent (e.g. total volume, O2 volume, foam percentage). Also, a table that compares the concentrations / infusion rates etc. for the various experiments would be helpful for a reader. Is there a theoretical limit for gas infusion rate based on your experiments?

Thank you for this important comment. We agree with the Reviewer and have clarified this throughout the text – now, all foam concentrations are expressed as volume of gas/volume of foam throughout the text.

Per the Reviewer's suggestion, we have also added a **Supplementary Table 1** (below) in Supplementary Materials that compares the concentrations and infusion rates for all experiments.

Supplementary Table 1. Summary of doses of PMBs used in various animal studies.

	Figure	Gas content (oxygen fraction)	Gas fraction	Gas injection rate (ml of gas/kg/min)*	Total gas volume infused	Total volume infused
Rodent hemodynamic study	Fig. 3c-f	Air (21%O ₂)	50% vol gas/vol foam	5 ml/kg/min	25 ml/kg	50 ml/kg
In vivo dissolution study by echo	Fig. 3g-l	Air (21%O ₂) and 100% O ₂	40% vol gas/vol foam	4 ml/kg/min; 8ml/kg/min; 12 ml/kg/min	4 ml/kg; 8 ml/kg; 12 ml/kg	10 ml/kg; 20 ml/kg; 30 ml/kg
Swine efficacy study	Fig. 4	100% O ₂	35% vol gas/vol foam	4 ml/kg/min	12 ml/kg	33 ml/kg
Rodent safety	Fig. 5	100% O ₂	40% vol gas/vol foam	1.3 ml/kg/min	13 ml/kg; 26 ml/kg	32 ml/kg; 64 ml/kg

*Gas injection rate was calculated as total foam volume injected/injection time (i.e., the time during which injections were actually occurring).

Finally, while there is likely to be an upper limit to the infusion rate, we did not find any data on which to base an estimate of this in injection rates up to 12 mL/kg/minute. At some theoretical infusion rate, it is possible that plasma would become supersaturated and result in gas formation (as occurs in the bends), but this maximal infusion rate will need to be experimentally determined in a future toxicology study.

7. Translational considerations. A discussion on long term stability and general setting in how the technology could be applied would help the audience understand if these advances are translatable to human application. For example how stable are the PMBs over a range of conditions e.g. Zone IVb, or would one require instrumentation at the bedside or near to support the continual generation of PMBs.

We thank the Reviewer for the comments regarding the translation of the technology. As discussed earlier, per the Reviewer's suggestions, we have performed an accelerated aging study at various temperatures and found that PMBs are quite stable in glass syringes up to 45°C (though we did not include humidity as part of this test). Given these findings, we do not anticipate the need for on-site manufacturing, which would be functionally prohibitive to the intended use in cardiac arrest and hypoxic emergencies. As a product, we envision that PMBs will be stored in a gas-tight glass in a hermetically sealed bag, stored at room temperature, ready to be deployed at bedside or emergency scenarios. We have added the following paragraph to the Discussion:

In clinical practice, we envision that IVO₂ would be available on-demand in environments caring for critically ill patients at risk for hypoxemia. Because arterial oxygen saturation is often continuously monitored by photoplethysmography in patients at risk for hypoxemia, IVO₂ could become a new treatment for refractory hypoxemia that is refractory to current standards of care, including airway clearance (e.g. suctioning), lung recruitment (e.g. hand ventilation) and the use of other critical maneuvers. Its dosing could be titrated in the same way that pre-arrest bolus doses of epinephrine are titrated to treat refractory hypotension,^[57] since both blood pressure and arterial oxygen saturation (by plethysmography) are routinely measured as vital signs in hospitalized patients. As in the swine model we described, patients being

treated for in-hospital cardiac arrest (IHCA) known or presumed to be caused by hypoxemia could also be treated with IVO₂ as an emergency treatment; because such patients standardly receive IV resuscitative treatments within 3-5 minutes ^[58], IVO₂ may restore early spontaneous circulation and significantly improve outcomes in such patients.

8. Chronic dosing safety. The reported safety studies noted single dosing of the materials in rodents. Safety on repeated dosing could also be of interest to the community. Understanding the effects of repeated dosing to potentially support subjects with compromised oxygenation (e.g.severe ARDS) could expand the impact of the intervention.

Thank you for this important comment. Because we envision the initial use of IVO₂ as an emergency rescue drug, the dosing used here supports our initial intended use. We do agree with the Reviewer that additional safety studies including repeated dosing will be additive in the future, likely as part of an FDA submission.

Minor:

1. Fig 2C: using dotted lines for different pH values is pretty hard for me to distinguish; can they make the labeling more clear?

We have modified this graph as the new **Fig. 2a** with new coloring and added an enlarged copy in the Supplementary Materials as **fig. S15** to facilitate viewing.

2. Why do the IFNP MBs used as a control in Fig 2c have such a sharp cutoff? Were these filtered by size?

The size distribution of dissolved IFNP MBs as large aggregates was indeed different LmD polymers. No size filter was applied, the sharper look of the left curve is partially due to the log scale of the x-axis.

3. Fig S8: size in um, not nm. Also, why is the y axis different from 1h?

We thank the Reviewer for pointing out this mistake. This graph in this figure (now **fig. S16**) shows the size of dissolved shells of PMBs made from modified hydroxyethyl starch, instead of the size of PMBs. We have corrected the legend for this figure.

4. Fig S13: x axis needs units

We thank the Reviewer for pointing out this error. We corrected added units to the x –axis as ‘Time from asphyxial onset (minutes).

5. How long is the LmD solution homogenized before HCl is added?

The LmD solution was homogenized for 2 minutes, before HCl was added. We modified the manufacturing procedure in the Methods section in Supplementary Materials to include this information.

Thank you again for a thorough, thoughtful, and helpful review of this manuscript, we feel your comments and suggestions have definitively made this manuscript significantly better.

References cited in Response to Reviewers:

1. Ahmed, H. et al. Use of Oxyhemoglobin Saturation, Rather Than Oxygen Tension, as a Marker of Oxygenation in Cyanotic Patients. *JAMA Pediatr.* **171**, 1012–1014 (2017).
2. Benjamin, E. J. et al. Heart Disease and Stroke Statistics-2019 Update: A Report From the American Heart Association. *Circulation.* **139**, e56–e528 (2019).
3. Girotra, S., Tang, Y., Chan, P. S. & Nallamothu, B.K. Survival After In-Hospital Cardiac Arrest in Critically Ill Patients: Implications for COVID-19 Outbreak? *Circ. Cardiovasc. Qual. Outcomes.* **13**, E006837 (2020).
4. Vestergaard, L. D. et al. Quality of Cardiopulmonary Resuscitation and 5-Year Survival Following in-Hospital Cardiac Arrest. *Open Access Emerg. Med.* **13**, 553–560 (2021).
5. Chan, P. S. et al. In-Hospital Cardiac Arrest Survival in the United States during and after the Initial Novel Coronavirus Disease 2019 Pandemic Surge. *Circ. Cardiovasc. Qual. Outcomes.* **15**, E008420 (2022).
6. Allencherril, J., Lee, P. Y. K., Khan, K., Loya, A. & Pally, A. Etiologies of In-hospital cardiac arrest: A systematic review and meta-analysis. *Resuscitation* **175**, 88–95 (2022).
7. Donnino, M. W. et al. Time to administration of epinephrine and outcome after in-hospital cardiac arrest with non-shockable rhythms: retrospective analysis of large in-hospital data registry. *BMJ.* **348**, g3028 (2014).
8. Cherry, B. H., Nguyen, A. Q., Hollrah, R. A., Olivencia-Yurvati, A. H. & Mallet, R. T. Modeling Cardiac Arrest and Resuscitation in the domestic pig. *World J. Crit. Care Med.* **4**, 1-12 (2015).
9. Voelckel, W. G. et al. Comparison of Epinephrine and Vasopressin in a Pediatric Porcine Model of Asphyxial Cardiac Arrest. *Crit. Care Med.* **28**, 3777–3783 (2000).
10. López-Herce, J. et al. Hemodynamic, respiratory, and perfusion parameters during asphyxia, resuscitation, and post-resuscitation in a pediatric model of cardiac arrest. *Intensive Care Med.* **37**, 147-55 (2011).
11. Andersen, L. W., Holmberg, M. J., Berg, K. M., Donnino, M. W. & Granfeldt, A. In-Hospital Cardiac Arrest A Review. *JAMA.* **321**, 1200-1210 (2019).
12. Sutton, R. M. et al. Hemodynamic Directed CPR Improves Short-term Survival from Asphyxia-Associated Cardiac Arrest. *Resuscitation* **84**, 696–701 (2013).
13. Leach, R. M. & Treacher, D. F. The Pulmonary Physician and Critical Care. 6. Oxygen Transport: the Relation between Oxygen Delivery and Consumption. *Thorax* **47**, 971–978 (1992).
14. Langeron, O. et al. Oxygen Consumption and Delivery Relationship in Brain-dead Organ Donors. *BJA: Br. J. Anaesth.* **76**, 783–789 (1996).
15. Shoemaker, W. C., Appel, P. L. & Kram, H.B. Tissue Oxygen Debt as a Determinant of Lethal and Nonlethal Postoperative Organ Failure. *Crit. Care Med.* **16**, 1117–1120 (1988).
16. Bland, R. D., Shoemaker, W. C., Abraham, E. & Cobo, J. C. Hemodynamic and Oxygen Transport Patterns in Surviving and Nonsurviving Postoperative Patients. *Crit. Care Med.* **13**, 85–90 (1985).
17. Hansen, M. et al. Time to Epinephrine Administration and Survival from Nonshockable Out-of-Hospital Cardiac Arrest among Children and Adults. *Circulation* **137**, 2032–2040 (2018).
18. Amoako, J., Komukai, S., Izawa, J., Callaway, C. W. & Okubo, M. Evaluation of Use of Epinephrine and Time to First Dose and Outcomes in Pediatric Patients With Out-of-Hospital Cardiac Arrest Key Points. *JAMA Netw. Open* **6**, 235187 (2023).
19. Topjian, A. A. et al. Part 4: Pediatric Basic and Advanced Life Support: 2020 American Heart Association Guidelines for Cardiopulmonary Resuscitation and Emergency Cardiovascular Care. *Circulation* **142**, S469–S523 (2020).
20. Yamaguchi, K. et al. Inhomogeneities of Ventilation and the Diffusing Capacity to Perfusion in Various Chronic Lung Diseases. *Am. J. Respir. Crit. Care Med.* **156**, 86–93 (1997).
21. Kheir, J. N. et al. Oxygen Gas-Filled Microparticles Provide Intravenous Oxygen Delivery. *Sci. Transl. Med.* **4**, 140ra88 (2012).
22. Peng, Y. et al. Interfacial Nanoprecipitation toward Stable and Responsive Microbubbles and Their Use as a Resuscitative Fluid. *Angew. Chem. Int. Ed.* **57**, 1271-1276 (2018).
23. Gao, W., Johnston, J. S., Miller, D. D. & Dalton, J. T. Interspecies Differences in Pharmacokinetics and Metabolism of S-3-(4-acetyl-amino-phenoxy)-2-hydroxy-2-methyl-N-(4-nitro-3-trifluoromethylphenyl)-propionamide: the Role of N-acetyltransferase. *Drug Metab Dispos.* **34**, 254-60 (2006).

24. Moghimi, S. M. & Simberg, D. Translational Gaps in Animal Models of Human Infusion Reactions to Nanomedicines. *Nanomedicine* **13**, 973-975 (2018).
25. Irizarry Rovira, A. R. et al. Scientific and Regulatory Policy Committee Points to Consider: Histopathologic Evaluation in Safety Assessment Studies for PEGylated Pharmaceutical Products. *Toxicol. Pathol.* **46**, 616-635 (2018).
26. Schoenbrunn, A., Juelke, K., Reipert, B. M., Horling, F., & Turecek, P. L. Polyethylene glycol 20 kDa-Induced Vacuolation does not Impair Phagocytic Function of Human Monocyte-derived Macrophages. *Front Immunol.* **13**, 894411 (2022).
27. Behrendt, F. F. et al. Impact of Different Vein Catheter Sizes for Mechanical Power Injection in CT: in vitro Evaluation with Use of a Circulation Phantom. *Cardiovasc. Intervent. Radiol.* **32**, 25–31 (2009).
28. Kheir, J., Black, K., Reese, J. & McGowan, F. Lipid-Based Oxygen Microparticles Do Not Change in Size Following Intravenous or Intraosseous Injection. *Circulation* **124**, A259 (2011).
29. Kheir, J. N. et al. Administration of Injectable Oxygen Suspension Rapidly Reverses Hypoxemia via Intravenous or Intraosseous Route. *Circulation* **124**, A3 (2011).

Rebuttal 2

Point-by-point Response to Reviewers' Comments.

Reviewer #1 (Report for the authors (Required)):

The authors have answered my questions. Differences in the opinion of in hospital and out of hospital response times care very relevant and a point to debate but at this time point in the evolution of this idea are not relevant.

Although I believe that the observed overall effect is very small (as far as the total volume of oxygen being able to be delivered) and possibly difficult to implement clinically to find a meaningful effect as currently deployed, the process is very innovative.

The findings are a significant advance in the field of alternative oxygenation strategies that can affect critical care and emergency care in the years to come and therefore warrant publication. I congratulate the PI and his team for such a significant body of work.

We thank the Reviewer for these comments and for a critical and helpful review of this manuscript.

Reviewer #2 (Report for the authors (Required)):

The authors did a very good job of responding to most of the reviewers' comments and the manuscript is greatly improved from the previous version. I have a few specific queries (see below), but my main concern is that the authors over-sell the potential utility of their approach. In reading parts of the paper, it sounds as if the authors have developed a therapy for many common forms of acute hypoxemia; this is not the case because of the large volume of fluid needed to deliver the oxygen. As stated by all the reviewers previously, the animal model of hypoxemia is very specific: acute very severe hypoxemia, fully reversible cause of hypoxemia; and knowledge of exactly when to treat. I think this is ok from a proof of concept perspective but the authors have to tone down the implied importance of their findings. I've given a number of specific examples below.

We thank the Reviewer for these comments. We agree with and appreciate these important caveats offered by the Reviewer and have made the following changes according to the Reviewer's specific suggestions.

1. Abstract:

a. The abstract should give the quantitative volume of fluid needed to provide a given volume of oxygen. Without this, the Abstract reads like their approach is useful for many subacute causes of hypoxemia

We thank the Reviewer for this comment. Congruent with the data in Supplementary Methods Table 1, we have added the following statement to the Abstract:

The carriers deliver 35-50 mL of oxygen per dL of foam.

b. Sentence starting "Administration of this carrier in swine with profound hypoxemia ...". It's important to state that the model is an acute, short model of complete airway

occlusion lasting a few minutes.

We thank the Review for this comment. We have edited this sentence to read:

Administration of this carrier in swine with profound hypoxemia **due to acute, temporary (12 minute) upper airway obstruction** maintained critical oxygenation, decreased the burden of cardiac arrest, improved survival, and resulted in significantly improved neurologic and kidney functions in survivors.

c. Same point for the last sentence of the Abstract; it's a bit misleading to say that this is a potentially viable therapy in hypoxic disorders. The data show that it's potentially valid in a very unique cause of hypoxemia. A broader statement about utility would require different clinically relevant model(s).

We thank the Reviewer for this comment. We have edited this sentence as follows:

These findings illustrate the importance of maintaining a critical threshold of oxygenation and the potential of injectable oxygen as a viable therapy in **an acute, time-limited hypoxemic crisis.**

2. Second sentence of Introduction (“In hospitalized patients,...”): This sentence mixes clinical causes with physiological mechanisms, and is thus a bit unfocused. For example, hypoxemia due to lung injury is usually largely due to ventilation-perfusion mismatch.

We thank the Reviewer for this comment. We have revised this statement to highlight a few of the clinical mechanisms as follows:

In hospitalized patients, severe, episodic hypoxemia can result from endotracheal tube occlusion (e.g. secretions), progressive lung injury, ~~ventilation-perfusion mismatch,~~ and a number of other causes.

3. 1st paragraph of Introduction: The last few sentences discuss IHCA with 15-40% being caused by respiratory insufficiency. Although the authors don't explicitly state this, it implies that their therapeutic approach may be potentially useful for many of these cases. I don't think this is the case. I think the introduction should address the mechanism of hypoxia that is simulated by their experimental model. I think this would provide a more realistic perspective for the reader.

We fully appreciate the caveats that the Reviewer pointed out. We feel the purpose of this section in the Introduction should be to provide a general description of the key problems. It is intended to highlight the general problem of hypoxemia in IHCA and describe key technological challenges in the field that have motivated our and others' efforts to develop intravenous oxygen. We are not making claims about the technology in this section.

On the other hand, although we acknowledge we only tested one experimental model, we do believe that this therapeutic approach could be useful for many of these cases of IHCA, and need not be limited to cases in which hypoxemia is caused only by upper airway obstruction, for the following reasons. The model that we describe here - acute airway obstruction – is physiologically the most extreme version of ventilation-perfusion mismatch; no lung segments are ventilated, but some are perfused. In this situation, the provision of injectable oxygen raises pulmonary arterial oxygen tension, raises *alveolar* oxygen tension in non-ventilated but perfused alveoli, and thereby

raises arterial oxygen tension. This same physiology would be effective in less extreme forms of V/Q mismatch, such as patients with lung injury. Although the causes of hypoxemia are numerous, in the end they all cause intrapulmonary veno-arterial admixture (i.e. shunting). Thus, raising 'venous' (i.e. pulmonary arterial) oxygen content using this technique results in improved arterial oxygen content irrespective of the cause, in the same way that the use of venovenous extracorporeal membrane oxygenation (VV-ECMO) indiscriminately reverses hypoxemia – regardless of its mechanism.

Nonetheless, we fully agree with the Reviewer that it is important for us to acknowledge that we have only demonstrated efficacy in one specific model of asphyxial cardiac arrest, and have added the following caveat to the 2nd paragraph of Discussion:

We acknowledge that while we have demonstrated benefit in a model of asphyxial cardiac arrest, whether this benefit is replicated in other models of severe hypoxemia (e.g. lung injury) needs to be experimentally determined using different animal models.

4. 2nd paragraph: The authors state that transfusion of oxygenated blood is by itself not a viable way to treat severe hypoxemia because of the huge volume of blood that would have to be given. This is absolutely correct, but the approach of IVO2 also requires very large volumes of fluid (albeit not of red blood cells). Again, I think this sets the stage for the reader to come away thinking that the current approach is potentially more useful that it may in fact be.

We thank the Review for this important comment. We view this statement as simply setting up the problem of oxygen injection and the need for alternative material design. However, in addition to the added statement of caveat in our reply to Comment 3, we fully acknowledge and expounded upon the limitation of volume administration in the Discussion section as below, in which we made qualifications to emphasize strategic use of IVO2 only as an emergency intervention. If the Reviewer has specific suggestions for how to lay the groundwork for this in the Introduction, we would be happy to consider them. However, we feel this progression from Introduction to Discussion 'tells the story' in an appropriate way:

The major limitation of this technology is the volume of the fluid which must be administered to deliver the oxygen payload. In this work, the gas fraction of the foam ranged between 35-50%, which can likely be optimized further as it has with similar products.^[15,59] However, given the high rate of resting oxygen consumption (e.g. a healthy adult consumes ~250 mL/minute), oxygen provision using this technology is unlikely to provide more than several hundred milliliters of oxygen, particularly in critically ill patients with heart disease in whom even small quantities of administered volume administration may aggravate venous hypertension and pulmonary edema.

As recommended below, we have also explicitly acknowledged the fluid volume required for the provision for a given volume of oxygen at the beginning of the Discussion as below:

At the gas concentration of the foams described (35-50 mL oxygen/dL foam), the provision of 100 mL oxygen gas requires the co-administration of 100-185 mL of additional fluid.

5. Section entitled “Instrumentation and model description”:

a. It’s not clear to me whether the animals in fact had cardiac arrest of severe hypotension based on the statement: “Animals experiencing cardiac arrest (systolic blood pressure <40 mmHg) …” What is your definition of “cardiac arrest”?

We thank the Reviewer for this question. We apologize for our lack of clarity on this issue. Systolic blood pressure thresholds have been used to define cardiac arrest in experimental models as a surrogate of pulselessness (as in Berg, *Circulation*, 2000) [1], since the determination of pulselessness based on physical examination findings may be prone to inaccuracy and delays [2-4]. Here, we defined the onset of cardiac arrest using a systolic blood pressure <40 mmHg for 5 seconds or longer. We have clarified this in the text as follows and added a new reference:

Animals experiencing cardiac arrest (defined as systolic blood pressure <40 mmHg for 5 seconds or longer) [39] received high-quality, chest compression CPR and rhythm-directed resuscitative interventions, including medications and defibrillation according to current standards.[40]

This is also stated explicitly in the Methods section:

Cardiac arrest was defined as a systolic blood pressure (SBP) less than 40 mmHg for 5 seconds or longer.

b. The treated group had an n=8, while the control group had an n=10. Why is the sample size different between groups? Are all animals reported in this study, or did you do experiments in some animals that you are not reporting? If the latter, please give details of which animals were not included and why their data was not used.

We have reported all experiments in the study. The sample size is different between groups only because we completed a pre-specified interim analysis following completion of 18 experimental replicates and the findings met pre-determined stopping rules.

6. Section entitled “Effect of oPMBs on resuscitation metrics”: The authors state that at 6 minutes of asphyxia/apnea, a similar number of animals experienced cardiac arrest. In reviewing fig. S21, it looks like just prior to 6 minutes about double the number of animals in the treatment group were free of cardiac arrest. Can you provide more detailed data on the physiological behavior of the animals in both groups just before 6 minutes. What was their BP, how many had complete cardiac arrest, etc. Given the small number of animals, it’s possible the Control group were somewhat sicker to start before treatment.

We thank the Reviewer for this important comment. As above, the definition of cardiac arrest was a systolic blood pressure (SBP)<40 mmHg for at least 5 seconds. As shown in **Fig 4c**, swine in both groups experienced a rapid decrease in their freedom from cardiac arrest between 4 and 6 minutes. The mean arterial blood pressures (MABP) for all swine from 5 minutes prior to cardiac arrest throughout the asphyxial period are shown in **Fig 4g**, copied below for your convenience. Red = control, blue = treatment; dots are individual replicates, data presented as mean± SEM. Comparison between groups by two-way ANOVA, Sidak’s multiple comparison with only significant p values shown. As shown, MABP did not differ significantly between groups prior to the treatment period.

7. Discussion:

a. I think the first paragraph of the discussion there should be an explicit statement of how much fluid is required to provide a given volume of oxygen, e.g., 750 ml of fluid to provide 250 ml O₂.

We thank the Reviewer for this comment. Congruent with our Supplemental Table 1, we have added the following statement in the first paragraph of the Discussion:

At the gas concentration of the foams described (35-50 mL oxygen/dL foam), the provision of 100 mL oxygen gas requires the co-administration of 100-185 mL of additional fluid.

b. Para 2: I think it's important to emphasize that in this model, you know exactly how much oxygen has been used up and exactly when to give it. This will not be so easy in the clinical setting. This wouldn't necessarily be a critical issue except that you have to give so much fluid to get a small amount of oxygen infused.

We thank the Reviewer for raising this critically important comment regarding the timing and dosing of IVO₂. We acknowledge that in this experimental model, we administered a pre-defined, fixed volume of oxygen at pre-defined, fixed time points, and agree that further investigation is needed to fully understand the optimal application of this technology in various clinical settings. We have modified paragraph 3 of the Discussion to emphasize these observations:

In clinical practice, we envision that IVO₂ would be available on-demand in environments caring for critically ill patients at risk for hypoxemia. Because arterial oxygen saturation is often continuously monitored by photoplethysmography in patients at risk for hypoxemia, IVO₂ could become a new treatment for refractory hypoxemia that is refractory to current standards of care, including airway clearance (e.g., suctioning), lung recruitment (e.g., hand ventilation) and the use of other critical maneuvers. Its dosing could be titrated in the same way that pre-arrest bolus doses of epinephrine are titrated to treat refractory hypotension,^[58] since both blood pressure and arterial oxygen saturation (by plethysmography) are routinely measured as vital signs in hospitalized patients. As in the swine model we described, pre-specified doses of IVO₂ could eventually be added to the resuscitation algorithm for patients being treated for in-hospital cardiac arrest (IHCA) caused by known or presumed hypoxemia; because such patients standardly receive IV resuscitative treatments within 3-5 minutes^[59], IVO₂ may restore early spontaneous circulation and

significantly improve outcomes in such patients. **The precise dose and timing of IVO2 treatment in each of these settings merit further investigation and optimization.**

c. Statement “ ...a given dose of oxygen may also have a more pronounced effect when used in clinical settings of lung injury.” I don’t understand the rationale underlying this statement.

We thank the Reviewer for this comment. The rationale for this statement is that the phenomenon of back-diffusion from pulmonary artery into the alveolus is attenuated in settings of more severe lung injury, which restricts ‘backwards’ gas exchange between the IVO2-oxygenated pulmonary arterial blood and the alveolar gas volume. We expounded upon this statement in the sentences that follow it, as below:

Further, a given dose of oxygen may also have a more pronounced effect when used in clinical settings of lung injury. Normally, oxygen flows from the alveolus into the blood, but in this model of airway obstruction, oxygen tension of the pulmonary artery exceeded that of the alveolus, such that oxygen initially diffused backwards, equilibrating with the functional residual capacity of the lung (i.e., increasing the volume of distribution of the gas payload); in this sense, this was an exaggerated model of ventilation-perfusion inequality, a central pathology in patients with clinical lung disease. *We expect that this phenomenon of back-diffusion would be attenuated, and therefore the dose response more pronounced,* in patients with more heterogenous ventilation-perfusion inequality or with an oxygen diffusion gradient.

As mentioned earlier in our reply to Comment 3, we also added some caveats following the above texts:

We acknowledge that while we have demonstrated benefit in a model of asphyxial cardiac arrest, whether this benefit is replicated in other models of severe hypoxemia (e.g. lung injury) needs to be experimentally determined using different animal models.

d. Next sentence, “...in this model of airway obstruction, oxygen tension of the pulmonary artery exceeded that of the alveolus.” Did you present evidence to support this statement?

We thank the Reviewer for this question. In this study, we did not directly measure the oxygen tension of the pulmonary artery. However, we believe this statement to be supported by the following data in this study:

1. A majority of PMBs dissolve prior to reaching the pulmonary artery in vivo as shown in **Fig. 3**.
2. In the asphyxial cardiac arrest model, there was severe alveolar hypoxemia, given that the arterial PO₂ at 5 minutes of asphyxia was <4 mmHg , as shown in **Fig. S23**.
3. Arterial oxygen tension increases in the setting of otherwise profoundly hypoxemic alveoli during injections of oPMBs as in **Fig. 4b**.
4. Following opening of the airway, the average arterial oxygen tension at 13 minutes in the treated group was 392±94 mmHg on FiO₂ 100%. This demonstrates that alveolar function is normal in this model, based on a near-normal alveolar-arterial oxygen gradient following restoration of the airway patency.

Thus, in the setting of severe alveolar hypoxemia (at 5 minutes), the PMBs that were injected into the venous circulation dissolved prior to reaching the pulmonary circulation, raising venous oxygen tension above that of the alveolus. In the setting of a healthy alveolus, this must be the case – otherwise arterial oxygen tension would not increase.

We have also described this process of ‘back diffusion’ in prior work during the acute injection of LOMs (lipid-based oxygen microbubbles) in hypoxic ventilation. For example, in Figure 4B of (Kheir, *Sci. Transl. Med.* 2022) [5] (shown here for convenience), we found that when LOMs were injected into a rabbit ventilated with 11% inspired oxygen fraction, the expired oxygen fraction increased at the same time that the arterial oxygen saturation (by pulse oximetry) increased. This increase in exhaled oxygen content represents the enrichment of alveolar oxygen with intravenously-injected oxygen through a process of “back diffusion.” In some examples (though not the one shown here), we have even seen expired oxygen fraction exceed inspired oxygen fraction.

8. In the section on limitations, the authors’ give a value of ~200 mL/min for the normal O₂ consumption. I understand that this is just a rough figure, but a more accepted value would be about 250 mL/min.

We thank the Reviewer for this comment. We have made this amendment.

We would like to sincerely thank the Reviewer for all the helpful suggestions, we believe these comments have helped us to provide a more balanced discussion about the limitations and the implications of the reported technology.

Reviewer #3 (Report for the authors (Required)):

The authors have addressed all of the concerns previously. Specifically:

-The experiments in the new Fig 2 sufficiently addressed the concern that the authors’ use of ultrasound within their measurements was causing the bubbles to prematurely burst (comment 4). These new experiments support the authors’ original claim that higher pH increases the speed of bubble dissolution/ outgassing.

-The accelerated stability study (Figs S8-14) demonstrated the long-term stability of the foam (comment 1).

This is a wonderful study and contribution that will be of interest to a broad audience.

We thank the Reviewer for your thorough review of our work and for your helpful commentary.

[References cited]

- 1, Berg, R. A., Hilwig, R. W., Kern, K. B. & Ewy, G. A. "Bystander" chest compressions and assisted ventilation independently improve outcome from piglet asphyxial pulseless "cardiac arrest". *Circulation* **101**,1743-1748 (2000).
2. Eberle, B. et al. Checking the carotid pulse check: diagnostic accuracy of first responders in patients with and without a pulse. *Resuscitation* **33**, 107-116 (1996).
3. Dick, W. F., Eberle, B., Wisser, G. & Schneider, T. The carotid pulse check revisited: what if there is no pulse? *Crit. Care Med.* **28**, N183-185 (2000).
4. Deakin, C. D. & Low, J. L. Accuracy of the advanced trauma life support guidelines for predicting systolic blood pressure using carotid, femoral, and radial pulses: observational study. *BMJ* **321**, 673-674 (2000).
- 5, Kheir, J. N. et al. Oxygen gas-filled microparticles provide intravenous oxygen delivery. *Sci. Transl. Med.* **4**, 140ra88 (2012).